# Understanding Outer Optimizers in Local SGD: Learning Rates, Momentum, and Acceleration

**Ahmed Khaled**[*]
Princeton University
ahmed.khaled@princeton.edu

**Satyen Kale**[†]
Google Research
satyen@satyenkale.com

**Arthur Douillard**
Google DeepMind
douillard@google.com

**Chi Jin**
Princeton University
Princeton, NJ 08544
chij@princeton.edu

**Rob Fergus**
NYU, Meta
fergus@cs.nyu.edu

**Manzil Zaheer**
Google DeepMind
manzilzaheer@gmail.com

## Abstract

Modern machine learning often requires training with large batch size, distributed data, and massively parallel compute hardware (like mobile and other edge devices or distributed data centers). Communication becomes a major bottleneck in such settings but methods like Local Stochastic Gradient Descent (Local SGD) show great promise in reducing this additional communication overhead. Local SGD consists of three parts: a local optimization process, an aggregation mechanism, and an outer optimizer that uses the aggregated updates from the nodes to produce a new model. While there exists an extensive literature on understanding the impact of hyperparameters in the local optimization process, the choice of outer optimizer and its hyperparameters is less clear. We study the role of the outer optimizer in Local SGD, and prove new convergence guarantees for the algorithm. In particular, we show that tuning the outer learning rate allows us to (a) trade off between optimization error and stochastic gradient noise variance, and (b) make up for ill-tuning of the inner learning rate. Our theory suggests that the outer learning rate should sometimes be set to values greater than 1. We extend our results to settings where we use momentum in the outer optimizer, and we show a similar role for the momentum-adjusted outer learning rate. We also study acceleration in the outer optimizer and show that it improves the convergence rate as a function of the number of communication rounds, improving upon the convergence rate of prior algorithms that apply acceleration locally. Finally, we also introduce a novel data-dependent analysis of Local SGD that yields further insights on outer learning rate tuning. We conduct comprehensive experiments with standard language models and various outer optimizers to validate our theory.

## 1 Introduction

Training very large scale machine learning models requires a lot of compute. This compute is often centrally controlled by a single entity and tightly connected in a data center. Gradients are constantly synchronized, hardware failures are controlled and mitigated, and things (mostly) run smoothly. Building this training infrastructure is expensive, however, and centralized control might not be desirable for all models. This has led to a surge of interest in decentralized collaborative training of large-scale models across different, potentially poorly connected clusters (Douillard,

---

[*]Part of this work was done during an internship at Google DeepMind.
[†]Currently at Apple.

39th Conference on Neural Information Processing Systems (NeurIPS 2025).

Feng, Rusu, Chhaparia, et al., 2023; Jaghouar, Ong, and Hagemann, 2024; Jaghouar, Ong, Basra, et al., 2024). This has motivated the adoption of federated learning algorithms in training language models, chiefly for scalability and communication efficiency rather than data privacy. Efficient parallelization strategies also factored in the remarkable recent training of DeepSeek V3 and R1 on a tight budget (DeepSeek-AI, Liu, et al., 2024; DeepSeek-AI, Guo, et al., 2025).

A foundational algorithm in distributed and federated optimization is Local SGD (Wang, Charles, et al., 2021). Many popular algorithms fit in the FedOpt template (Reddi et al., 2021) (Algorithm 1), including FedAdam (Reddi et al., 2021), FedRR (Mishchenko, Khaled, and Richtárik, 2022; Malinovsky and Richtárik, 2022), DiLoCo (Douillard, Feng, Rusu, Chhaparia, et al., 2023; Jaghouar, Ong, and Hagemann, 2024) and many others. FedOpt solves the minimization problem $\min_{x \in \mathbb{R}^d} f(x)$ given access to $M$ different computational nodes and unbiased stochastic gradients of $f$. FedOpt consists of three main components: an inner update loop on every client, an aggregation of the client updates, and then an outer update step taken on the server.

---

**Algorithm 1** The FedOpt Algorithmic Template

---

1: **Input.** Update rules $\mathrm{LocalUpdate}$ and $\mathrm{OuterUpdate}$. Initial point $x_0$.
2: **for** communication rounds $r = 0, 1, \ldots, R - 1$ **do**
3:     Broadcast $x_r$ to each node $m$
4:     **for** each node $m$ in parallel **do**
5:       Set $y_{m,r,0} = x_r$.
6:       **for** local steps $h = 0, 1, \ldots, H - 1$ **do**
7:         Set $y_{m,r,h+1} = \mathrm{LocalUpdate}(y_{m,r,h}, g_{m,r,h})$ for stochastic gradient $g_{m,r,h}$ at $y_{m,r,h}$.
8:       **end for**
9:       Communicate $y_{m,r,H}$ to the server.
10:     **end for**
11:     Compute the update or "outer gradient" $\hat{\Delta}_{r,H} = \frac{1}{M} \sum_{m=1}^{M} (y_{m,r,H} - x_r)$.
12:     Update $x_{r+1} = \mathrm{OuterUpdate}(x_r, -\hat{\Delta}_{r,H})$.
13: **end for**

---

When both the local and outer update rules correspond to gradient descent (i.e. $x_{\mathrm{new}} = x_{\mathrm{old}} - \beta \Delta$ for some stepsize $\beta$ and update vector $\Delta$), the corresponding algorithm is Generalized Local SGD. If we additionally take the outer stepsize to be 1, we get Local SGD. Local SGD simply does $H$ steps of SGD on each node, and then averages the result after applying the updates. This is the most common form in which the algorithm is analyzed, as in e.g. (Stich, 2019; Khaled, Mishchenko, and Richtárik, 2020; Woodworth, Patel, Stich, et al., 2020; Koloskova et al., 2020; Glasgow, Yuan, and Ma, 2022; Patel, Glasgow, Zindari, et al., 2024). In practice, different choices of outer optimizers perform better. For example, DiLoCo/OpenDiLoCo use SGD with Nesterov Momentum as the outer optimizer (Douillard, Feng, Rusu, Chhaparia, et al., 2023). This has motivated much analysis of different outer optimizers and their impact (Reddi et al., 2021; Malinovsky, Mishchenko, and Richtárik, 2022; Jhunjhunwala, Wang, and Joshi, 2023; Sun et al., 2024). However, our theoretical understanding of the fundamental Generalized Local SGD algorithm remains limited. In particular, it is not clear why the bilevel optimization structure of the algorithm is helpful from an optimization perspective, even in the i.i.d. setting where the data distribution is the same on all the nodes. Additionally and to the best of our knowledge, we have no explicit expressions for what the ideal learning rate pair $(\eta, \gamma)$ for the inner and outer updates, respectively, should be. Empirically, outer optimizers employing Nesterov acceleration have the best performance, yet to the best of our knowledge why or how it improves convergence is not known.

**Contributions.** Our paper takes steps to address the above questions and makes the following contributions.

- We conduct a novel, tighter analysis of Generalized Local SGD (Theorem 1) that shows the outer learning rate plays a dual role. It (a) interpolates between two extreme regimes: taking many effective steps at the cost of higher variance to taking fewer steps but at reduced variance and (b) increases the algorithmic robustness to hyperparameter tuning by making up for ill-tuned inner learning rates. The latter holds even in the absence of any stochastic gradient noise.

- We extend the above analysis to cover Generalized Local SGD where the outer optimizer also uses momentum (Theorem 2) and show that this gives additional leeway in tuning $\gamma$.

- We provide a convergence analysis for Local SGD with an accelerated outer optimizer and unaccelerated inner optimizer (Theorem 3), showing that using Nesterov acceleration in the outer loop achieves better dependence on the number of communication rounds $R$ in the drift terms compared to standard Local SGD and improving upon the convergence rate of FedAc (Yuan and Ma, 2020).

- We also derive a data-dependent, high-probability guarantee for the convergence of Local SGD with GD as the outer optimizer (Theorem 4) that shows further benefits of tuning the outer stepsize in more nuanced settings.

- We additionally conduct an extensive empirical analysis for training large-scale language models with various outer optimizers (gradient descent, accelerated gradient descent, and Schedule-Free gradient descent).

We now review related work, then proceed to our main results.

## 2   Related Work

There is a rich literature on algorithms for communication-efficient distributed optimization for *federated learning* (Konečný et al., 2016), where multiple clients collaborate on solving a machine learning problem  (Wang, Charles, et al., 2021). Federated learning algorithms are designed to reduce the effect of data heterogeneity (Karimireddy et al., 2020; Wang, Charles, et al., 2021; Murata and Suzuki, 2021), ensure the data stays private (Wei et al., 2020), deal with intermittent or cyclic client availability (Eichner et al., 2019), among other issues.

As models have grown larger in size over the past few years, going from a few million parameters to billions (Brown et al., 2020), the scale of training runs has also grown to include many more devices divided across multiple computing clusters rather than a single cluster (Diskin et al., 2021; Huang, Huang, and Liu, 2022; Borzunov et al., 2022; Douillard, Feng, Rusu, Chhaparia, et al., 2023). Even within a single datacenter, training runs now involve tens of thousands of GPUs (Jiang et al., 2024). This has motivated researchers to develop and use algorithms inspired by the federated learning setting for large-scale training instead. Examples of such algorithms include DiLoCo (Douillard, Feng, Rusu, Chhaparia, et al., 2023), its open cousin OpenDiLoCo (Jaghouar, Ong, and Hagemann, 2024), DiPaCo (Douillard, Feng, Rusu, Kuncoro, et al., 2024), and others (Liu et al., 2024; Liang et al., 2024; DeepSeek-AI, Liu, et al., 2024). Federated learning methods thus have found use in pretraining and fine-tuning language models (Jaghouar, Ong, and Hagemann, 2024; Yang et al., 2025), and may prove particularly important for scaling even larger models in the future (Iacob et al., 2024; Sani et al., 2024; Rush et al., 2024). We note that the use of methods for federated learning even for i.i.d. distributed training is not new, and is perhaps being "re-discovered" as training runs grow too large to fit on single clusters. For example, Lin et al. (2020) argued that using Local SGD can be more efficient than traditional Minibatch SGD in some settings. Ortiz et al. (2021) also conducted experiments studying the trade-offs of using Local SGD in training image classification models.

The most popular algorithm in the federated optimization literature is Local SGD or Federated Averaging (Wang, Charles, et al., 2021). It is a generalization of minibatch SGD that, rather than communicating at every step of the optimization process, communicates only intermittently. Local SGD shows remarkable efficiency in many settings in practice, and therefore its convergence and generalization properties have been the subject of intense theoretical investigation over the past few years (Stich, 2019; Khaled, Mishchenko, and Richtárik, 2020; Woodworth, Patel, Stich, et al., 2020; Woodworth, Patel, and Srebro, 2020; Patel, Glasgow, Wang, et al., 2023; Glasgow, Yuan, and Ma, 2022; Gu, Lyu, Huang, et al., 2023; Patel, Glasgow, Zindari, et al., 2024). Many variants of Local SGD exist, including those that use random reshuffling instead of i.i.d. sampling locally (Yun, Rajput, and Sra, 2022; Mishchenko, Khaled, and Richtárik, 2022), adaptive methods such as Adam (Reddi et al., 2021; Wang, Lin, and Chen, 2022), and modifications to handle data heterogeneity (Karimireddy et al., 2020; Mitra et al., 2021), personalization (Hanzely et al., 2020), or additionally use gradient compression (Haddadpour et al., 2021; Safaryan, Hanzely, and Richtárik, 2021). Generalized Local SGD, where we use two stepsizes (as in Algorithm 1), is known to be important in managing the trade-off between converging quickly and converging to a mismatched point in heterogeneous distributed optimization (Woodworth, Patel, and Srebro, 2020; Charles and Konečný, 2020; Patel, Glasgow, Zindari, et al., 2024). Our focus here is on the *homogeneous* or i.i.d. data setting; Here, the most related works are (Karimireddy et al., 2020; Malinovsky, Mishchenko, and Richtárik, 2022;

Jhunjhunwala, Wang, and Joshi, 2023; Sun et al., 2024) and we discuss our work's relation to theirs in detail in the next section after reviewing some preliminaries.

## 3 Theory

In this section we conduct the study our main algorithm, Generalized Local SGD (Algorithm 1 with $\text{LocalUpdate}(y, g) = y - \eta g$ and $\text{OuterUpdate}(x, \Delta) = x - \gamma \Delta$). We first review some preliminaries, then present our main results.

### 3.1 Preliminaries

We are solving the optimization problem $\min_{x \in \mathbb{R}^d} f(x)$, where we assume $f$ satisfies the following curvature and regularity condition.

**Assumption 1.** The function $f$ is differentiable, convex, has $L$-Lipschitz gradients, and has a minimizer $x_*$.

We suppose that we can access a *stochastic first-order oracle* that given a point $x$ returns a gradient $g(x)$ that satisfies the following assumption.

**Assumption 2.** Given a point $x \in \mathbb{R}^d$, the stochastic gradients $g(x) \in \mathbb{R}^d$ are (a) unbiased in expectation $\mathbb{E}[g(x)] = \nabla f(x)$, and (b) has variance bounded as $\mathbb{E}\left[\|g(x) - \nabla f(x)\|^2\right] \leq \sigma^2$, where $\mathbb{E}[\cdot]$ denotes the expectation operator.

Our setting is distributed, but with identically distributed data: there are $M$ different nodes, but they all sample stochastic gradients from the same data distribution in an i.i.d. (independent and identically distributed) manner. We denote the inner product between two vectors $a$ and $b$ by $\langle a, b \rangle$ and by $\|\cdot\|$ the corresponding Euclidean norm. For the purpose of theoretical analysis, can write Generalized Local SGD succinctly as

$$y_{m,r,0} = x_r, \qquad g_{m,r,h} = \text{Stochastic gradient of } y_{m,r,h}$$

$$y_{m,r,h+1} = y_{m,r,h} - \eta g_{m,r,h}, \text{ for } m = 1, \ldots, M \text{ in parallel and } h = 0, 1, \ldots, H-1 \text{ in sequence.}$$

$$x_{r+1} = x_r - \gamma \eta \sum_{h=0}^{H-1} \frac{1}{M} \sum_{m=1}^{M} g_{m,r,h}. \tag{GEN-LOC-SGD}$$

To simplify our analysis, we follow (Stich, 2019) and define the virtual sequences

$$y_{r,h} \overset{\text{def}}{=} \frac{1}{M} \sum_{m=1}^{M} y_{m,r,h}, \qquad g_{r,h} \overset{\text{def}}{=} \frac{1}{M} \sum_{m=1}^{M} g_{m,r,h} \tag{1}$$

$$\overline{g}_{m,r,h} \overset{\text{def}}{=} \mathbb{E}_{r,h-1}[g_{m,r,h}] = \nabla f(y_{m,r,h}), \qquad \overline{g}_{r,h} \overset{\text{def}}{=} \mathbb{E}_{r,h-1}[g_{r,h}].$$

### 3.2 Main convergence result

Recall that we consider Algorithm 1 the particular case when $\text{LocalUpdate}(y, g) = y - \eta g$ and $\text{OuterUpdate}(x, \Delta) = x - \gamma \Delta$.

**Existing results on the convergence of Gen. Local SGD.** When the outer stepsize $\gamma = 1$, the convergence of (GEN-LOC-SGD) is very well understood, with tightly matching upper and lower bounds (Khaled, Mishchenko, and Richtárik, 2020; Woodworth, Patel, Stich, et al., 2020; Glasgow, Yuan, and Ma, 2022). In particular, the best rate for the algorithm is

$$\mathbb{E}\left[f\left(\frac{1}{RH}\sum_{r=0}^{R-1}\sum_{h=0}^{H-1} y_{r,h}\right)\right] - f(x_*) \leq \mathcal{O}\left(\frac{L\|x_0 - x_*\|^2}{RH} + \frac{\sigma\|x_0 - x_*\|}{\sqrt{MRH}} + \frac{L^{\frac{1}{3}}\sigma^{\frac{2}{3}}\|x_0 - x_*\|^{\frac{4}{3}}}{H^{\frac{1}{3}}R^{\frac{2}{3}}}\right). \tag{2}$$

The first two terms in the above convergence guarantee show that increasing the number of local steps has the same effect as increasing the number of communication rounds $R$, and are identical to the convergence guarantee of doing $RH$ steps of SGD with minibatch size $M$. Local SGD differs from ordinary minibatch SGD in the last term, which shows different scaling between $H$ and $R$,

where increasing $R$ helps more than increasing $H$. This is because increasing $H$ incurrs additional *client drift* that slows down the convergence of the algorithm in the presence of stochastic gradient noise. When the outer stepsize $\gamma$ is allowed to vary, the convergence of the algorithm is less clear. Karimireddy et al. (2020) gives the following convergence rate in the absence of data heterogeneity,

$$\mathbb{E}\left[f\left(\tfrac{1}{R}\sum_{r=0}^{R-1} x_r\right)\right] - f(x_*) \leq \mathcal{O}\left(\tfrac{L\|x_0-x_*\|^2}{R} + \tfrac{\sigma\|x_0-x_*\|}{\sqrt{MR}}\right),$$

for specially chosen $\eta$ and $\gamma$ pairs. This rate matches that of Minibatch SGD, but does not recover the convergence rate of vanilla Local SGD given by Equation (2). Jhunjhunwala, Wang, and Joshi (2023) also give a guarantee for Generalized Local SGD with a specific outer learning rate that is always at least 1 and that depends on the heterogeneity of the iterates across the different clients. Since the analysis is conducted in the heterogeneous setting, the local stepsize required to scale with $1/H$. A guarantee that applies to any outer learning rate in the nonconvex, heterogeneous setting given by (Sun et al., 2024).

The limiting factor in existing analysis is that we are forced to choose the local stepsize $\eta$ to scale as $\frac{1}{LH}$, whereas to obtain Equation (2) we sometimes need to choose $\eta$ to be much larger, on the order of $\frac{1}{L}$. If we aim to accurately characterize the convergence of (GEN-LOC-SGD), our analysis has to encompass both large and small local stepsizes $\eta$.

**New analysis.** We now present our main convergence theorem for (GEN-LOC-SGD).

**Theorem 1.** *Suppose that Assumptions 1 and 2 hold. Then the iterates generated by Generalized Local SGD run with local stepsize $\eta > 0$ and outer stepsize $\gamma > 0$ for $R$ communication rounds and with $H$ local steps per round satisfy,*

$$\mathbb{E}\left[f\left(\tfrac{1}{RH}\sum_{r=0}^{R-1}\sum_{h=0}^{H-1} y_{r,h}\right)\right] - f(x_*) \leq \mathcal{O}\left(\tfrac{\|x_0-x_*\|^2}{\eta\gamma RH} + \tfrac{\eta\sigma^2\max(\gamma,1)}{M} + L\eta^2\sigma^2 H\right), \quad (3)$$

*provided the stepsizes $\eta$ and $\gamma$ jointly satisfy $\eta L(1 + (\gamma - 1)_+ H) \leq \frac{1}{4}$ and where $(a)_+ = \max(a, 0)$.*

**Implications of Theorem 1.** Before giving a proof sketch for Theorem 1, we first discuss its implications. Observe the stepsize condition $\eta L(1 + (\gamma - 1)_+ H)$ is asymmetric in $\eta$ and $\gamma$; That is, when $\gamma \leq 1$, we are allowed to choose $\eta$ larger than $\Omega(\frac{1}{LH})$. This is crucial to obtain the rate of Equation (2). Indeed, when $\gamma = 1$, the requirement on $\eta$ reduces to $\eta L \leq \frac{1}{4}$ and we can choose $\eta$ following (Woodworth, Patel, Stich, et al., 2020) as

$$\eta = \min\left(\tfrac{1}{4L}, \sqrt{\tfrac{M\|x_0-x_*\|^2}{\sigma^2 RH}}, \left[\tfrac{\|x_0-x_*\|^2}{L\sigma^2 H^2 R}\right]^{\frac{1}{3}}\right)$$

Plugging this choice of $\eta$ yields the convergence guarantee of Equation (2). Alternatively, when $8\eta L \leq 1$, the stepsize requirement is met if we choose $\eta\gamma LH \leq \frac{1}{8}$ and we immediately get the Minibatch SGD guarantee. In particular, choose $\eta = \mathcal{O}\left(\frac{1}{RL}\right)$ and $\gamma = \mathcal{O}\left(\frac{\gamma_*}{\eta LH}\right)$, the rate then becomes

$$f(y_{\text{out}}) - f(x_*) \leq \tfrac{8L\|x_0-x_*\|^2}{\gamma_* R} + \tfrac{\sigma^2 H}{8R^2 L} + \tfrac{\gamma_*\sigma^2}{4LMH},$$

where $y_{\text{out}}$ denotes the average over all iterations and clients as in Equation (3). Then for $R$ large enough we can choose $\gamma_* = \mathcal{O}\left(\sqrt{\tfrac{LD^2\sigma^2 MH}{R\sigma^2}}\right)$ and this gives us the minibatch SGD rate

$$f(y_{\text{out}}) - f(x_*) \leq \tfrac{LD^2}{R} + \tfrac{\sigma D}{\sqrt{MRH}}.$$

This confirms the intuition that at the extremes, manipulating the stepsizes $\gamma$ and $\eta$ allows us to interpolate between minibatch SGD and (vanilla) Local SGD, as observed by (Woodworth, Patel, and Srebro, 2020). In fact, Theorem 1 allows us to go a step further and get an explicit expression for the optimal inner and outer stepsizes depending on the problem parameters. This is given by the following proposition.

**Proposition 1.** *Let $h(\eta, \gamma)$ be defined as*

$$h(\eta, \gamma) = \tfrac{D^2}{\eta\gamma RH} + L\sigma^2 H\eta^2 + \tfrac{\eta\max(\gamma,1)\sigma^2}{M}. \quad (4)$$

*Consider the optimization problem:*

$$\min_{\eta>0,\gamma>0} h(\eta, \gamma) \quad \text{subject to} \quad \eta L\left(1 + (\gamma - 1)_+ H\right) \leq \frac{1}{4}. \quad (5)$$

*The solution $(\eta^*, \gamma^*)$ is given by comparing the following two candidates.*

1. *Candidate $(\eta_A^*, \gamma_A^*)$ defined by $\gamma_A^* = 1$ and $\eta_A^* = \min(\frac{1}{4L}, \eta_A')$ where $\eta_A'$ is the unique positive root of the cubic equation*

$$2LH\sigma^2\eta^3 + \frac{\sigma^2}{M}\eta^2 - \frac{D^2}{RH} = 0.$$

2. *Candidate $(\eta_B^*, \gamma_B^*)$ for the regime $\gamma \geq 1$ with $4\eta L < 1$, where (a) the constraint is enforced with equality:*

$$\gamma_B(\eta) = 1 + \frac{1}{H}\left(\frac{1}{4L\eta} - 1\right),$$

*and (b) $\eta_B^*$ is the unique positive root of the cubic equation*

$$-\frac{4L^2 D^2(H-1)}{R} + 2L\sigma^2 H\eta\left(\eta L(H-1)+1\right)^2 + \frac{\sigma^2(H-1)}{MH}(\eta L(H-1)+1)^2 = 0.$$

*The optimal solution $(\eta^*, \gamma^*)$ is the candidate pair from $\{(\eta_A^*, \gamma_A^*), (\eta_B^*, \gamma_B^*)\}$ that yields the smaller value of $h(\eta, \gamma)$.*

The proof of the above proposition is straightforward and follows by writing the KKT conditions for the optimization problem in Equation (5). A consequence of Proposition 1 is that in the case of ill-tuning of the inner stepsize $\eta$, a large outer stepsize $\gamma$ can make up for it. For example, if $\sigma \to 0$ and $\eta LH \ll \mathcal{O}(1)$, we can make up for this by choosing $\gamma$ as $\frac{1}{\eta LH}$. Thus, we can interpret the outer learning rate $\gamma$ as having **two dual roles.** (a) It allows us to interpolate between minibatch SGD ($\gamma > 1$) and vanilla Local SGD ($\gamma = 1$), giving us the better of the two rates, and (b) it provides us some additional leeway in hyperparameter tuning by making up for ill-tuned inner learning rate $\eta$.

Our theory suggests that *in the worst case*, choices of $\gamma < 1$ are *not* useful from an optimization perspective. We should either choose $\gamma = 1$ or $\gamma > 1$. This can be seen even on quadratic objectives, for example if $f(x) = \frac{x^\top Q x}{2}$ for some positive definite matrix $Q$, then a straightforward computation gives the expected iterate after $H$ local steps and $R$ communication rounds is $\mathbb{E}[x_R] = ((1-\gamma)I + \gamma(I - \eta Q)^H)x_0$. From this, it is clear that if $\eta$ is chosen such that $(I - \eta Q)^H$ has eigenvalues smaller than 1, we should choose $\gamma \geq 1$. While if $(I - \eta Q)^H$ has any eigenvalues larger than 1, we should just choose $\gamma = 0$ (i.e. just don't apply the algorithm at all). In other words, $\gamma$ can make up for a learning rate that is too small, but not a learning rate that is too large. This observation does not exclude that $\gamma < 1$ can be useful from a *generalization* perspective, as noted for the case of a single client by Zhou et al. (2021), in the presence of data heterogeneity, as noted by Charles and Konecný (2021), or in the presence of specific stochastic gradient distributions (see Section 3.4).

**Proof sketch for Theorem 1.** We first start by expanding the update for the round iterate $x_{r+1} - x_* = x_{r+1} - x_r + x_r - x_*$ similar to (Karimireddy et al., 2020) to get,

$$\|x_{r+1} - x_*\|^2 = \|x_r - x_*\|^2 - 2\gamma\eta\sum_{h=0}^{H-1}\langle x_r - x_*, g_{r,h}\rangle + \gamma^2\eta^2\left\|\sum_{h=0}^{H-1} g_{r,h}\right\|^2$$

$$= \|x_r - x_*\|^2 - 2\gamma\eta\sum_{h=0}^{H-1}\langle x_r - y_{r,h}, g_{r,h}\rangle - 2\gamma\eta\sum_{h=0}^{H-1}\langle y_{r,h} - x_*, g_{r,h}\rangle + \gamma^2\eta^2\left\|\sum_{h=0}^{H-1} g_{r,h}\right\|^2,$$

where $g_{r,h}$ is defined as in Equation (1). Karimireddy et al. (2020) and Jhunjhunwala, Wang, and Joshi (2023) control the inner product $-\langle x_r - y_{r,h}, g_{r,h}\rangle$ by either using smoothness or Young's inequality; This would force us to bound the stray $\|y_{r,h} - x_r\|^2$ and take the local stepsize $\eta$ to be small in order to ensure convergence. Instead, we rely on bounding this quantity directly by viewing it as the *regret* in the online convex optimization sense with respect to the comparator $x_r$. Observe that the virtual sequence of averaged local iterates satisfies $y_{r,h+1} = y_{r,h} - \eta g_{r,h}$, and thus through standard regret analysis we have

$$\sum_{h=0}^{H-1} -\langle x_r - y_{r,h}, g_{r,h}\rangle = \frac{-\|y_{r,h} - x_r\|^2}{2\eta} + \frac{\eta}{2}\sum_{h=0}^{H-1}\|g_{r,h}\|^2. \tag{6}$$

The negative terms $-\|y_{r,H} - x_r\|^2$ in Equation (6) turn out to be crucial in obtaining an analysis that works for all $\eta$ and not just small $\eta$. With this change and through carefully bounding the variance terms following (Khaled, Mishchenko, and Richtárik, 2020; Woodworth, Patel, Stich, et al., 2020), we obtain the guarantee of Theorem 1. The full proof is provided in Section B.2.

**Comparison with results on related algorithms.** Malinovsky, Mishchenko, and Richtárik (2022) analyze a closely related variant of the algorithm that uses federated random reshuffling (Mishchenko,

Khaled, and Richtárik, 2022) as a base. This is a significantly different algorithm that doesn't allow for an arbitrary number of local steps $H$ and depends on $f$ posessing finite-sum structure. Nevertheless, we can still specialize (Malinovsky and Richtárik, 2022, Theorem 2) approximately to our setting, by using $H$ as the number of data points in an epoch. In our notation, their convergence guarantee reads

$$\mathbb{E}\left[f\left(\tfrac{1}{R}\sum_{r=0}^{R-1}x_r\right)\right] - f(x_*) \leq \mathcal{O}\left(\tfrac{\|x_0-x_*\|^2}{\eta\gamma HR} + \eta^2 H^2\sigma^2\right),$$

under the conditions $\eta H \leq \tfrac{1}{L}$ and $1 \leq \gamma \leq \tfrac{1}{L\eta H}$. Their theory thus also suggests that $\gamma \geq 1$ can be useful. Optimizing over $\eta$ and $\gamma$ yields the convergence rate

$$\mathbb{E}\left[f\left(\tfrac{1}{R}\sum_{r=0}^{R-1}x_r\right)\right] - f(x_*) \leq \mathcal{O}\left(\tfrac{L\|x_0-x_*\|^2}{R}\right),$$

this rate is the same as gradient descent for $R$ steps (since the finite-sum structure means that per-epoch we approximate one step of gradient descent when $\eta$ is small). A similar rate is derived in (Li, Acharya, and Richtárik, 2024; Li and Richtárik, 2024) if we have access to the proximal operator (i.e. we can do *many* local steps $H$ on a modified objective). Li, Acharya, and Richtárik (2024) in particular show that an outer learning rate greater than 1 can be particularly useful for improving the convergence of FedProx (Li, Sahu, et al., 2020) in the heterogeneous setting when the smoothness constant varies significantly between different clients.

**Analysis with momentum.** Our analysis suggests that values of $\gamma > 1$ are potentially very useful, but in practice such values are rarely used. One reason this might be the case is because the momentum effectively acts as a stepsize multiplier, i.e. in the presence of momentum parameter $\mu$ the effective outer stepsize becomes $\tfrac{\gamma}{1-\mu}$. Our next theorem establishes this rigorously.

**Theorem 2.** *Suppose that Assumptions 1 and 2 hold. Suppose that the outer update is gradient descent with momentum,* $\mathrm{OuterUpdate}(x_r, -\Delta_{r,H}) = x_r + \gamma\Delta_{r,H} + \mu(x_r - x_{r-1})$ *with momentum parameter* $\mu \in [0,1)$ *and the local update is gradient descent* $\mathrm{LocalUpdate}(y,g) = y - \eta g$ *in Algorithm 1. Let the step sizes* $\eta, \gamma$ *satisfy* $\eta L\left(1 + \left(\tfrac{\gamma}{1-\mu} - 1\right)_+ H\right) \leq \tfrac{1}{4}$ *and* $\tfrac{\eta\gamma\mu LH}{1-\mu} \leq \tfrac{1}{16}$. *Then after* $R$ *rounds of communication, the averaged iterate satisfies*

$$\mathbb{E}\left[f(\overline{y})\right] - f(x_*) \leq \mathcal{O}\left(\frac{(1-\mu)\|z_0-x_*\|^2}{\eta\gamma HR} + L\eta^2\sigma^2 H + \frac{\eta\sigma^2\max(\tfrac{\gamma}{1-\mu},1)}{M} + \frac{\eta\gamma\mu}{1-\mu}\frac{\sigma^2}{M}\right),$$

*where* $\overline{y}$ *is defined as the average of all local iterates across training (as in Equation (3)) and* $(a)_+ = \max(a,0)$.

The proof is provided in Section B.3. Theorem 2 shows the requirement on the outer stepsize is relaxed from a requirement on $\gamma$ to a requirement on $\tfrac{\gamma}{1-\mu}$, allowing us to reap the same benefits of $\gamma > 1$ observed earlier if we also tune $\mu$. Momentum thus changes the range of stepsizes allowed but does not fundamentally alter the uter stepsize tradeoffs. This benefit was first observed in (Sun et al., 2024) for nonconvex optimization with small local stepsize $\eta$ provided we use an additional momentum buffer. Our work gives direct theoretical support to this observation even with a single momentum buffer and allowing for large $\eta$.

### 3.3 Convergence with accelerated outer optimizer

We now consider the use of acceleration. To the best of our knowledge, the combination of an accelerated outer optimizer with an unaccelerated inner optimizer, as in e.g. DiLoCo (Douillard, Feng, Rusu, Chhaparia, et al., 2023; Jaghouar, Ong, and Hagemann, 2024), has not been analyzed in the literature before. We take steps towards addressing this gap and understanding the convergence properties of such algorithms by considering Nesterov's accelerated gradient descent (Nesterov, 2018) as the outer optimizer and (stochastic) gradient descent as the inner optimizer. The following theorem gives a convergence guarantee for this setting.

**Theorem 3.** *Suppose that Assumptions 1 and 2 hold and the stepsizes satisfy* $2L\eta \leq 1$ *and* $\gamma \leq 1$. *Suppose that the outer update is accelerated gradient descent with Nesterov momentum as follows*

$$u_{r+1} = x_r - \eta\Delta_{r,H}, \qquad z_{r+1} = z_r - \gamma_r\eta\Delta_{r,H}, \qquad x_{r+1} = (1-\tau_{r+1})u_{r+1} + \tau_{r+1}z_{r+1},$$

with parameters $\gamma_r = \frac{\gamma(r+1)}{2}$ and $\tau_r = \frac{2}{r+2}$, and the local update is gradient descent $\text{LocalUpdate}(y, g) = y - \eta g$ in Algorithm 1. Then after $R$ rounds of $H$ steps, the final iterate $u_R$ satisfies

$$\mathbb{E}\left[f(u_R)\right] - f(x_*) \leq \frac{2\|x_0 - x_*\|^2}{\gamma \eta R^2 H} + \frac{RL\eta^2\sigma^2 H}{2M} + \frac{RL^2\eta^3\sigma^2 H^2}{2} + \frac{\gamma\eta\sigma^2 R}{2M}. \tag{7}$$

To understand the implications of the above guarantee, we specialize it with a tuned pair of learning rates $(\gamma, \eta)$ below.

**Corollary 1.** *In the same setting as Theorem 3, setting $\gamma = 1$ in Equation (7), and choosing*

$$\eta = \min\left\{ \frac{1}{2L}, \left(\frac{2MD^2}{R^3 L\sigma^2 H^2}\right)^{1/3}, \left(\frac{4D^2}{3R^3 L^2\sigma^2 H^3}\right)^{1/4}, \sqrt{\frac{4MD^2}{R^3 H\sigma^2}} \right\},$$

*where $D = \|x_0 - x_*\|$ and the final iterate $u_R$ satisfies*

$$\mathbb{E}[f(u_R)] - f(x_*) \leq \mathcal{O}\left( \frac{LD^2}{R^2 H} + \frac{L^{1/3}\sigma^{2/3}D^{4/3}}{RM^{1/3}H^{1/3}} + \frac{L^{1/2}\sigma^{1/2}D^{3/2}}{R^{5/4}H^{1/4}} + \frac{\sigma D}{\sqrt{MRH}} \right). \tag{8}$$

Equation (8) shows that in the absence of noise, we obtain a rate accelerated in $R$ but not $H$. This intuitively makes sense, since we do acceleration only in the outer loop. In the presence of noise, we have in the worst-case the unimprovable $\frac{\sigma D}{\sqrt{MRH}}$ term and two additional noise terms that characterize the drift suffered by this algorithm. Notably, the drift terms have much better dependence on $R$ compared to Local SGD, as given by Equation (2). Yuan and Ma (2020) analyze FedAC, an accelerated variant of Local SGD that uses acceleration *locally* and applies simple averaging as the outer optimizer. Their algorithm enjoys the convergence rate

$$\mathbb{E}[f(x_{\text{out}})] - f(x_*) \leq \mathcal{O}\left( \frac{LD^2}{R^2 H} + \frac{L^{1/3}\sigma^{2/3}D^{4/3}}{RH^{1/3}} + \frac{L^{1/2}\sigma^{1/2}D^{3/2}}{RH^{1/4}} + \frac{\sigma D}{\sqrt{MRH}} \right).$$

Comparing with Equation (8), our algorithm enjoys better dependence on $R$ and $M$ in the denominators of the two drift terms while using momentum sequences only on the server.

## 3.4 Data-dependent convergence result

To further understand the role of the outer stepsize, we now present a data-dependent, high-probability guarantee for Generalized Local SGD in Theorem 4, compared to the rather worst-case analysis of Theorem 1. This analysis may also provide insights into practical tuning of the outer learning rate

**Theorem 4.** *Suppose that Assumptions 1 and 2 hold. Then in Algorithm 1 with outer update $x = x - \gamma\Delta$ and local update $y = y - \eta g$, if the local stepsize satisfies $\eta \leq \frac{1}{L}$ then with probability at least $1 - \delta$ the iterates generated satisfy*

$$f\left(\frac{1}{RH}\sum_{r=0}^{R-1}\sum_{h=0}^{H-1} y_{r,h}\right) - f(x_*) \leq \tilde{\mathcal{O}}\Bigg( \frac{\|x_0 - x_*\|^2}{\gamma\eta RH} + \frac{\gamma\eta}{RH}\sum_{r,h}\|g_{r,h}\|^2 + \gamma\eta\sigma^2$$

$$+ \frac{|1-\gamma|\eta}{RH}\sum_r\left(\sum_h\|g_{r,h}\|\right)^2 + \frac{\eta}{\gamma H}\left(\frac{1}{M}\max_r\sum_{m,h}\|g_{m,r,h}\|\right)^2 + \eta\sigma\sqrt{\frac{1}{MR}\sum_{m,r,h}\|g_{m,r,h}\|^2} \Bigg).$$

The proof of Theorem 4 is provided in Section B.5. Compared to Theorem 1, the guarantee we obtain here is weaker in some areas, e.g., the variance term $\gamma\eta\sigma^2$ does not benefit from increasing $M$. On the other hand, this guarantee is a high-probability and data-dependent guarantee. To the best of our knowledge, this is the first high-probability convergence guarantee for Local SGD in the literature. Theorem 4 allows us to observe another potential benefit of using $\gamma \neq 1$. To see how, let us make the simplifying assumption that $\|\hat{g}_{r,h}\| \cong G_1$ and $\|g_{m,r,h}\| \cong G_2$. Observe that by the triangle inequality we have $G_1 \leq G_2$, but in fact $G_1$ can be significantly smaller than $G_2$, particularly in the later stages of the optimization process, due to the variance reduction effect of averaging together the gradients on different nodes. Then we can rewrite the above guarantee as

$$f(\overline{y}) - f(x_*) \leq \tilde{\mathcal{O}}\left( \frac{d_0^2}{\gamma\eta RH} + \gamma\eta G_1^2 + \gamma\eta\sigma^2 + |1-\gamma|\eta HG_1^2 + \frac{\eta HG_2^2}{\gamma} + \eta\sigma\sqrt{H}G_2 \right) \tag{9}$$

The $\gamma$ that minimizes this upper bound is given by the following proposition.

**Proposition 2.** *Let $g(x) = \frac{a}{x} + bx + |1 - x|\,c$ for $a, b, c \geq 0$.*

- *if $a \geq b + c$, then $\sqrt{a/(b+c)}$ minimizes $g$,*
- *if $b - c \geq 0$ and $a \leq b - c$, then $\sqrt{a/(b-c)}$ minimizes $g$,*
- *Otherwise, $x = 1$ minimizes $g$.*

Applying this lemma to Equation (9) one can see that simple averaging is suboptimal depending on the variance and relative magnitudes of $G_1$ and $G_2$. In particular, the first condition in our setting is

$$\frac{d_0^2}{\eta R H} + \eta H G_2^2 \gtrsim \eta(G_1^2 + \sigma^2) + \eta H G_1^2,$$

where $\gtrsim$ indicates that the inequality holds up to constant factors of the terms on both sides. Since $G_2 \geq G_1$, we can simplify the above condition to $\frac{d_0^2}{\eta^2 R H} + H G_2^2 \gtrsim \sigma^2$. This condition essentially asks if the noise is large relative to the "optimization term" $\frac{d_0^2}{\eta^2 R H}$ or not. In the latter case, choosing $\gamma > 1$ is helpful, and the outer optimizer acts as a form of momentum that helps reduce the optimization term further. On the other hand, the second condition yields $\gamma < 1$ and requires that $\sigma^2 \gtrsim \frac{d_0^2}{\eta^2 R H} + H G_2^2$. This is an especially noise-dominated regime, which we may expect to observe towards the end of the training process. In this case, decaying the outer learning rate to $\gamma \ll 1$ allows the algorithm to maintain convergence despite the high noise magnitude. When the optimization term and the noise term are of the same order, then $\gamma = 1$ is the optimal choice.

# 4 Experiments

We conduct two sets of experiments: (a) solving convex optimization problems to provide the most direct verification of the predictions of our theory, and (b) training transformer based language models. Due to limitations of space, we present only highlights of the results here and most of the details and ablations are provided in the supplementary materials (Section A).

## 4.1 Convex optimization

We conduct experiments on the quadratic objective $f(x) = \frac{1}{2}\|Q(x - x_*)\|^2$, where $Q = A^\top A \in \mathbb{R}^d$ for $d = 50$ and the entries $A_{i,j}$ are all drawn from a normal distribution $A_{i,j} \sim \mathcal{N}(0,1)$ for $i = 1, \ldots, d$ and $j = 1, \ldots, d$, and $x_*$ is similarly drawn from the standard $d$-dimensional Gaussian. We use stochastic gradients of the form $g(x) = \nabla f(x) + v$, where the $v$'s are random vectors drawn from the Gaussian with mean $0$ and variance

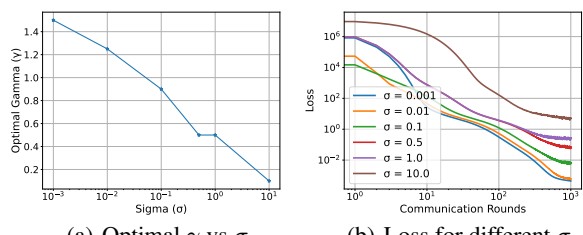

(a) Optimal $\gamma$ vs $\sigma$.     (b) Loss for different $\sigma$.

Figure 1: Effect of varying noise magnitude $\sigma$ and outer learning rate $\gamma$ for quadratic optimization.

$\sigma^2$, $v \sim \mathcal{N}(0, \sigma^2)$. We evaluate the performance of Algorithm 1 for various values of $\sigma$, $\sigma \in \{10^{-3}, 10^{-2}, 10^{-1}, 0.5, 1, 5, 10, 15, 25, 50\}$. For each $\sigma$ we perform an extensive grid search over $\gamma \in \{0.001, 0.01, 0.1, 0.5, 0.9, 1.0, 1.1, 1.25, 1.5, 2\}$ to determine the best one in terms of minimum average loss over the last ten rounds. We use $R = 1000$ rounds and $H = 50$ local steps, and fix $\eta = 0.001$ in all cases.

Figure 1(a) shows how the optimal value of $\gamma$ varies with different noise levels $\sigma$. We observe that, as $\sigma$ increases, the optimal $\gamma$ decreases from $1.0$ to $0.1$, as predicted by our analysis. Figure 1(b) also illustrates the loss trajectories for different noise levels $\sigma$ with the best $\gamma$.

## 4.2 Transformer pretraining

**Setup** Following the DiLoCo paper (Douillard, Feng, Rusu, Chhaparia, et al., 2023), we experiment using a Chinchilla decoder transformer (Hoffmann et al., 2022) on the C4 dataset (Raffel et al., 2020). The architecture hyperparameters are identical from the DiLoCo paper (Douillard, Feng, Rusu,

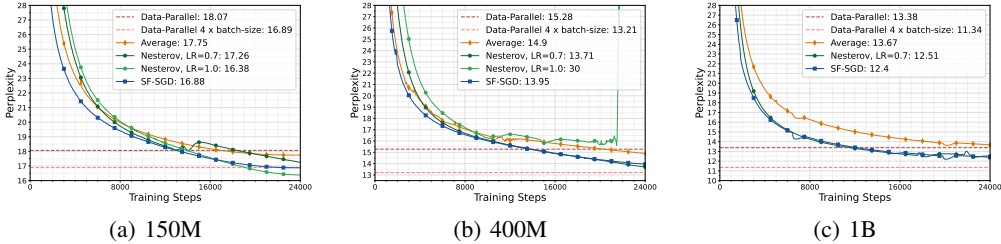

| (a) 150M | (b) 400M | (c) 1B |

Figure 2: **Scaling** distributed pretraining, at 150M, 400M, and 1B parameters. The x-axis shows the total number of training steps, including both local and communication steps. The y-axis shows the perplexity achieved by each method. Legend represents final perplexity values.

Chhaparia, et al., 2023) and are given in Section A.1.1. We fix the batch size at $512$ and the sequence length at $1024$. We experiment at different scales, from 150 million to 1 billion parameters. For all experiments, the inner optimizer is AdamW (Loshchilov and Hutter, 2019) trained with a cosine learning rate schedule defined across the total amount of steps. The inner optimizer state is never shared across replicas, and is passed from one round to the other.

**Methods** We compare three distributed methods, using different outer optimizers: SGD(lr=1) (equivalent to simple averaging of local models (McMahan et al., 2017)), Nesterov (equivalent to DiLoCo (Douillard, Feng, Rusu, Chhaparia, et al., 2023)), and ScheduleFree-SGD (SF-SGD) (Defazio et al., 2024). We use SF-SGD to substitute for outer learning rate scheduling, though it still requires tuning hyperparameters. We also include two "high-communication" data-parallel baselines: one with the global batch size as the local per-replica batch size used by the distributed methods, and one with the same batch size as the global batch size ($M \times$ the local per-replica batch size) used by the distributed methods. The latter requires either more GPUs and more thus communication, or gradient accumulation and thus more time. The latter also has an equal flops budget as the distributed methods. We tuned all our optimizers on the pretraining setting on a separate validation set . We also considered using SF-Nesterov, but it was hard to tune and unstable.

**Results** Table 1 gives the optimal hyperparameters per scale, and Figure 2 gives the perplexity curves. The perplexity was calculated on the C4 validation set. Consistent with the predictions of our theory, we found that an outer learning rate greater than $1.0$ performed best for SF-SGD and a relatively large effective outer learning rate also performed best for Nesterov; Moreover, acceleration consistently improved performance relative to the baseline Local SGD. In the supple-

| Hyperparameter | Selected | Range considered |
|---|---|---|
| Number of inner steps H | 50, 500 | 50 to 2000 |
| Peak outer LR for Nesterov | 0.7 | 0.1 to 2.0 |
| Peak outer LR for SF-SGD | 2.0 | $1e^{-4}$ to 10.0 |
| b1 for SF-SGD | 0.2 | 0.0 to 0.99 |
| Peak inner learning rate (150M) | $4e^{-4}$ | $4e^{-4}$ |
| Peak inner learning rate (400M) | $4e^{-4}$ | $4e^{-4}$ |
| Peak inner learning rate (1B) | $2e^{-4}$ | $2e^{-4}$ |

Table 1: **Optimizer hyperparameters** for the three evaluated sizes. All are based on the transformer architecture, chinchilla-style (Hoffmann et al., 2022).

mentary material, we report the effect of varying the number of local steps (Section A.1.2), the number of clients/replicas and different ways of FLOPs allocation (Section A.1.3), and gradient variance (Section A.1.6). We also include the validation results for all the main experiments we ran in Tables 3 to 5.

## 5 Conclusion and Future Work

In this paper, we studied the impact of the outer learning rate on the convergence of Local SGD through two novel convergence theorems that characterize its role in balancing a trade-off between convergence speed and stochastic gradient variance. We have also studied the impact of using momentum in the presence of an outer learning rate, and provided a new convergence analysis for using Nesterov acceleration in the outer optimizer. One limitation of our results is that we only consider the i.i.d. setting; Studying the impact of data heterogeneity is therefore a natural next step. Another avenue for future work is to investigate the role of adaptive outer optimizers in enhancing robustness to client failures and communication delays.

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

# Supplementary material

## A  Supplementary experimental details

In this section we provide the details on the language model pretraining experiments discussed in the main text.

### A.1  Language model pretraining

We study the impact of using various outer optimizers on large language model pretraining. We utilized Chinchilla-style decoder transformer architectures (Hoffmann et al., 2022) trained on the C4 dataset (Raffel et al., 2020), consistent with common practices in large-scale model training (Douillard, Feng, Rusu, Chhaparia, et al., 2023). The following subsections detail the specific hyperparameters, variations in training configurations (such as the number of inner steps and replicas/clients), and analyses of optimizer behavior, including learning rate scheduling and observed gradient cosine similarities.

#### A.1.1  Hyperparameters details

We show in Table 1 the hyperparameters considered and kept, and in Table 2 the architectural hyperparameters. We use the SentencePiece tokenizer with a sequence length of 1024 for all models. We tuned all our optimizers on a separate validation set. We also considered using the Schedule-Free Optimizer with Nesterov acceleration on top but it was hard to tune and unstable. We include the validation results for all the main experiments we ran in Tables 3 to 5.

Table 2: **Model Configuration** for the three evaluated sizes. All are based on the transformer architecture, chinchilla-style (Hoffmann et al., 2022).

| Hyperparameter | 150M | 400M | 1B |
|---|---|---|---|
| Number of layers | 12 | 12 | 24 |
| Hidden dim | 896 | 1536 | 2048 |
| Number of heads | 16 | 12 | 16 |
| K/V size | 64 | 128 | 128 |
| Vocab size | | 32,000 | |

#### A.1.2  Varying inner steps

In Figure 3, we compare the stability of different outer optimizers when varying the synchronization frequency. We experiments a different amount of inner steps, from 50, to 2000. All experiments are run in pretraining from scratch, with 150 millions (150M) parameters. We note that as the synchronization frequency decreases (number of inner/local steps increases), performance decreases. Notably, averaging (in orange), is relatively constant w.r.t the synchronization frequency: its performance stay stable from $H = 250$ to $H = 2000$. On the other hand, using Nesterov with high outer learning rate (in light green) is particularly unstable, its performance decreases by $10.7\%$, this indicates that the learning rate should be tuned alongside the synchronization frequency. On the hand, SF-SGD (in blue) has minimal degradation of performance ($4.2\%$), highlighting the *schedule-free* property when varying hyperparameters.

#### A.1.3  Varying replicas / flops budget

When increasing the number of distributed replicas, two options are possible: (a) Keeping the local per-replica batch size constant and thus increasing global batch size and flops budget, and (b) Keeping the global batch size/flops budget constant and thus reducing the local per-replica batch size.

We present in Figure 4 results of the first option with x-axis the flops budget for a single model size (150M). It is worth noting that increasing the number of replicas improves the performance of Nesterov (in green) and SF-SGD (in blue) but the gain quickly plateau. On the other hand, increasing

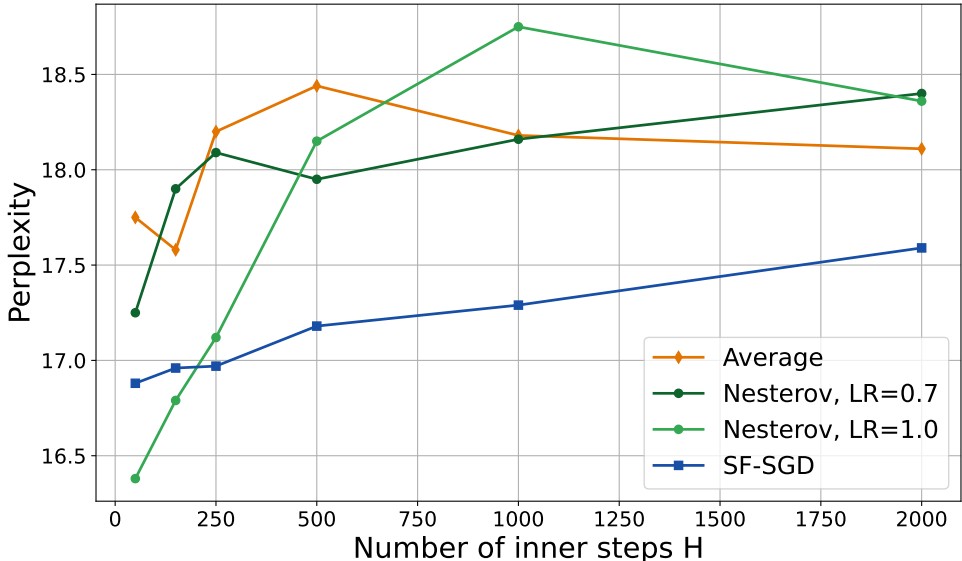

Figure 3: **Varying the communication frequency**, i.e. number of inner steps $H$, when pretraining from scratch at 150M parameters.

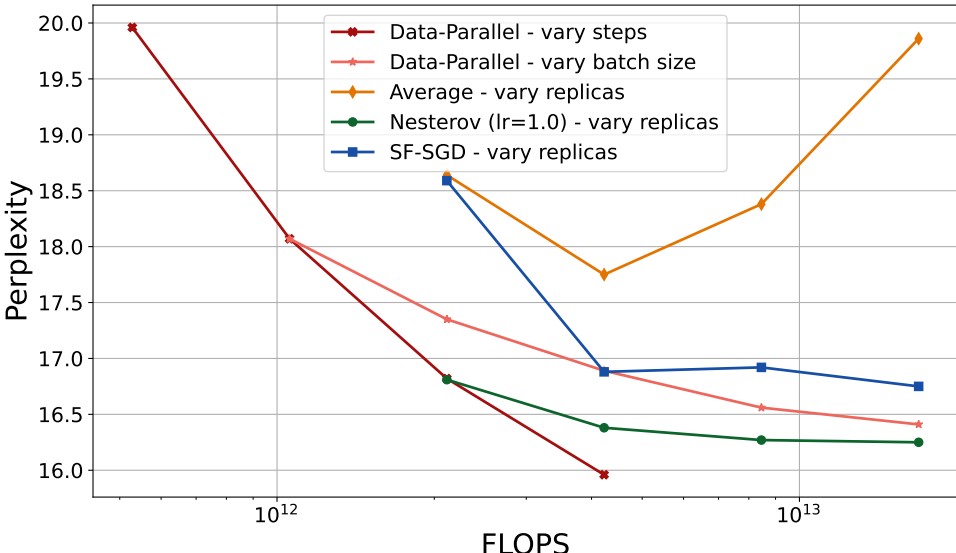

Figure 4: **Pareto front** of the flops vs perplexity, comparing various approach scaling the flops budget: increasing the number of steps, increasing the batch size in data-parallel, and increasing the number of replicas for federated learning.

the batch size for data-parallel (at the cost of more communication, because more DP replicas) or the number of steps (at the cost of longer training) still rapidly improves perplexity. Therefore, we wish to highlight here a disadvantage of federated learning methods seldom mentioned: while those methods are extremely communication-efficient, and can be made flops-efficient, their flops-efficiency disappear as the number of replicas increases.

To this problem, several hypotheses could be raised, such as the decreasing cosine similarity between outer gradients as the number of replicas increase, even when using an *i.i.d.* data split across replicas. In Figure 5, we report the average similarity across a whole training for different number of replicas. For momentum-based methods (Nesterov, SF-SGD), the similarity decreases from 30% at $M = 2$ replicas to 10% at $M = 16$ replicas. Full details across training steps can be found in the appendix.

Finally, note that we didn't investigate further the second option of keeping the global batch size/flops budget constant and thus reducing the local per-replica batch size. We found that dividing the batch

size by the number of replicas leads quickly to a local per-replica batch size that is critically low, and further reduces the flops-efficiency. More investigations should be pushed in that direction.

### A.1.4 Schedule-free but not tuning-free

The schedule-*free* method of Defazio et al., 2024 enables not doing any learning rate scheduling, greatly simplifying training configuration. However, it doesn't mean it is hyperparameters-tuning-*free*. Indeed, we found out that we had to extensively tune the initial learning rate (to 2.0), remove learning rate warm-up contrarily to what is advised, and use a particularly low $b1$ decay: 0.2, as illustrated in Figure 6.

### A.1.5 Pretraining: outer learning rate scheduling

Schedule-free SGD enables not having to manually scheduling the outer learning rate. Therefore, we wondered if we could improve the SotA federated learning baseline, DiLoCo (Nesterov outer optimizer), with an outer learning rate schedule. We investigate in Figure 7 three schedules: *constant* as in (Douillard, Feng, Rusu, Chhaparia, et al., 2023), *cosine decay*, and *linear after a plateau*. For the latter we consider a constant plateau for 10% and 25% of the total steps. For each method, we also tuned the peak outer learning rate. We don't use any warm-up in the outer optimization as we always found it to be harmful.

We find that constant outer learning rate is the best performing schedule. It's unclear how the other schedules are interacting with the inner learning rate scheduling. A possible solution, not investigated in this report, would be to increase the number of inner steps $H$ as the inner learning rate decreases (Gu, Lyu, Arora, et al., 2024).

### A.1.6 Cosine similarity between outer gradients

We display the cosine similarity between outer gradients, across scales (150M, 400M, and 1B) in Figure 8, and across replicas (for 150M, from 2 to 16 replicas) in Figure 9. The solid line represent the mean, and the shaded area the standard deviation. We normalize the x-axis as a percentage of the training in order to compare models which have done different amount of steps (e.g. 24,000 steps for 150M vs 30,000 for 400M).

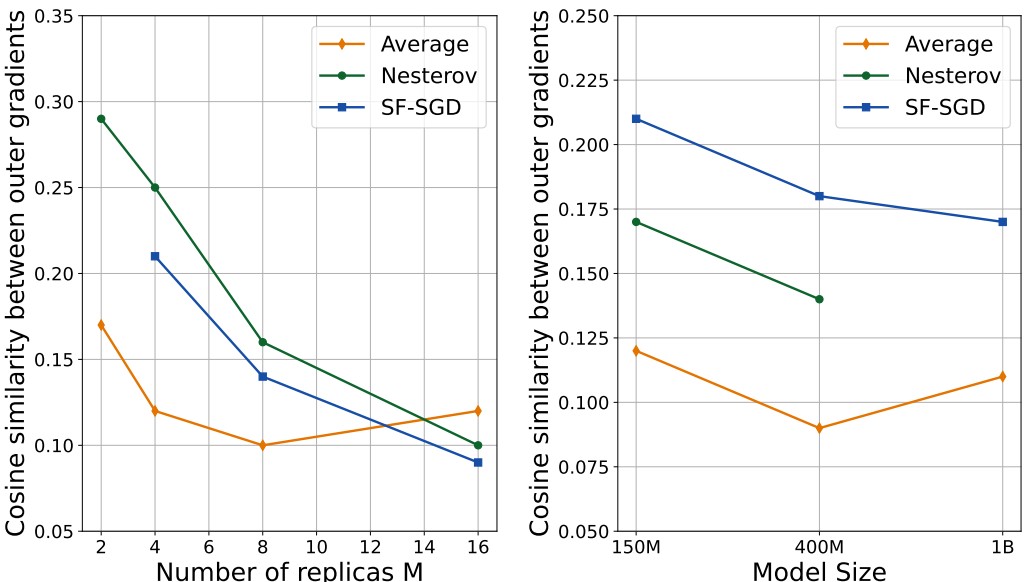

Figure 5: **Cosine similarity** between outer gradients across different number of replicas (*left*) and model scales (*right*). We average the similarity across the middle 50% of the training.

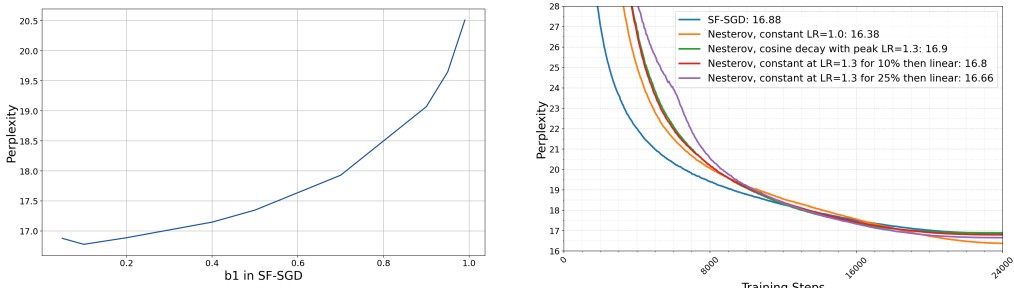

Figure 6: **Tuning b1 decay** has a major impact on performance, and its value must be very low.

Figure 7: Which outer **learning rate schedule** to use?

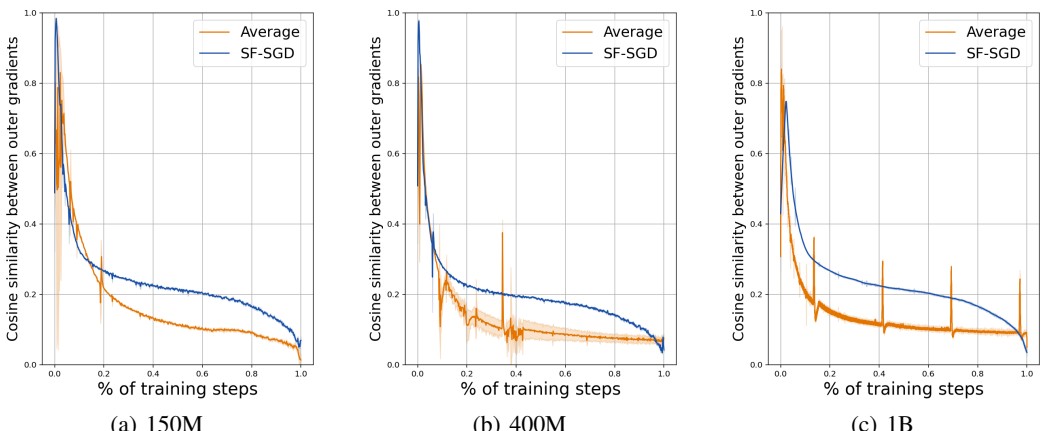

(a) 150M

(b) 400M

(c) 1B

Figure 8: **Similarity** between outer gradients across steps and scales.

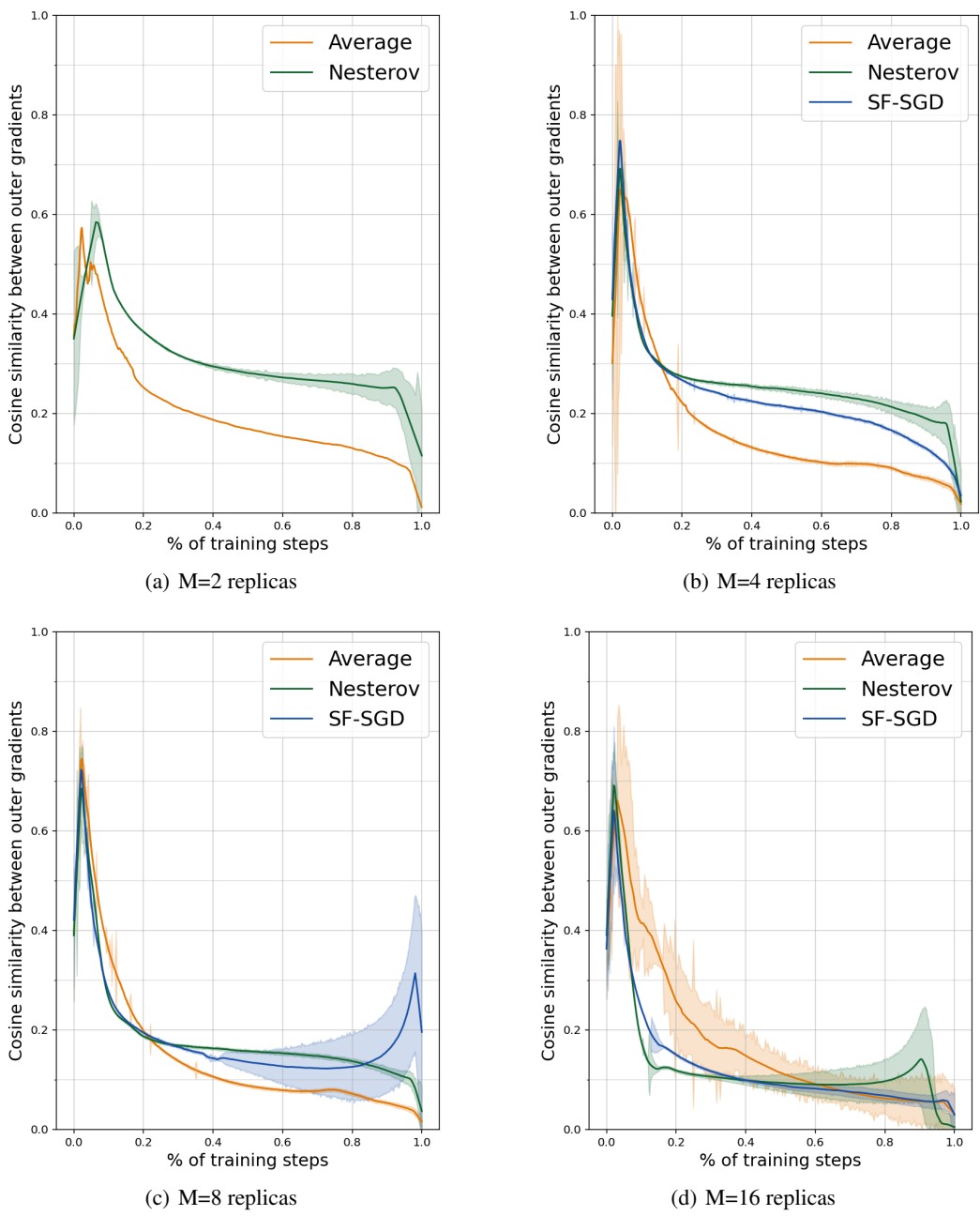

Figure 9: **Cosine similarity** between outer gradients across steps and number of replicas.

Table 3: Complete hyperparameter sweep results across model scales and configurations. All experiments use C4 validation set with sequence length 1024 and batch size 512.

| $H$ | $M$ | Algorithm | Learning Rate | Perplexity | Model Size |
|---|---|---|---|---|---|
| *Data-Parallel Baselines* | | | | | |
| - | 1 | Data-Parallel | - | 18.07 | 150M |
| - | 1 | Data-Parallel | 4x BS | 16.89 | 150M |
| - | 1 | Data-Parallel | - | 15.28 | 400M |
| - | 1 | Data-Parallel | 4x BS | 13.21 | 400M |
| - | 1 | Data-Parallel | - | 13.38 | 1B |
| - | 1 | Data-Parallel | 4x BS | 11.34 | 1B |
| *Local SGD Experiments* | | | | | |
| 50 | 4 | SGD | 1.0 | 17.75 | 150M |
| 50 | 4 | Nesterov | 0.7 | 17.25 | 150M |
| 50 | 4 | Nesterov | 1.0 | 16.38 | 150M |
| 50 | 4 | SF-SGD | 2.0 ($\beta$=0.2) | 16.88 | 150M |
| 50 | 4 | SGD | 1.0 | 14.90 | 400M |
| 50 | 4 | Nesterov | 0.7 | 13.71 | 400M |
| 50 | 4 | Nesterov | 1.0 | >30 | 400M |
| 50 | 4 | SF-SGD | 2.0 ($\beta$=0.2) | 13.95 | 400M |
| 50 | 4 | SGD | 1.0 | 13.67 | 1B |
| 50 | 4 | Nesterov | 0.7 | 12.51 | 1B |
| 50 | 4 | SF-SGD | 2.0 ($\beta$=0.2) | 12.40 | 1B |
| *Varying $H$ (Local Steps) at 150M, $M = 4$* | | | | | |
| 150 | 4 | SGD | 1.0 | 17.58 | 150M |
| 150 | 4 | Nesterov | 0.7 | 17.90 | 150M |
| 150 | 4 | Nesterov | 1.0 | 16.79 | 150M |
| 150 | 4 | SF-SGD | 2.0 ($\beta$=0.2) | 16.96 | 150M |
| 250 | 4 | SGD | 1.0 | 18.20 | 150M |
| 250 | 4 | Nesterov | 0.7 | 18.09 | 150M |
| 250 | 4 | Nesterov | 1.0 | 17.12 | 150M |
| 250 | 4 | SF-SGD | 2.0 ($\beta$=0.2) | 16.97 | 150M |
| 500 | 4 | SGD | 1.0 | 18.44 | 150M |
| 500 | 4 | Nesterov | 0.7 | 17.95 | 150M |
| 500 | 4 | Nesterov | 1.0 | 18.15 | 150M |
| 500 | 4 | SF-SGD | 2.0 ($\beta$=0.2) | 17.18 | 150M |
| 1000 | 4 | SGD | 1.0 | 18.18 | 150M |
| 1000 | 4 | Nesterov | 0.7 | 18.16 | 150M |
| 1000 | 4 | Nesterov | 1.0 | 18.75 | 150M |
| 1000 | 4 | SF-SGD | 2.0 ($\beta$=0.2) | 17.29 | 150M |
| 2000 | 4 | SGD | 1.0 | 18.11 | 150M |
| 2000 | 4 | Nesterov | 0.7 | 18.40 | 150M |
| 2000 | 4 | Nesterov | 1.0 | 18.36 | 150M |
| 2000 | 4 | SF-SGD | 2.0 ($\beta$=0.2) | 17.59 | 150M |
| *Varying $M$ (Number of Nodes) at 150M, $H = 50$* | | | | | |
| 50 | 2 | SGD | 1.0 | 18.64 | 150M |
| 50 | 2 | Nesterov | 1.0 | 16.81 | 150M |
| 50 | 2 | SF-SGD | 2.0 ($\beta$=0.2) | 17.13 | 150M |
| 50 | 8 | SGD | 1.0 | 18.38 | 150M |
| 50 | 8 | Nesterov | 1.0 | 16.27 | 150M |
| 50 | 8 | SF-SGD | 2.0 ($\beta$=0.2) | 16.92 | 150M |
| 50 | 16 | SGD | 1.0 | 19.86 | 150M |
| 50 | 16 | Nesterov | 1.0 | 16.25 | 150M |
| 50 | 16 | SF-SGD | 2.0 ($\beta$=0.2) | 16.75 | 150M |

Table 4: Additional outer learning rate sweeps for different outer optimizers. All experiments at 150M model size with $H = 50$ and $M = 4$.

| Algorithm | Learning Rate | Perplexity |
|-----------|---------------|------------|
| *SF-SGD Learning Rate Sweep ($\beta = 0.2$)* | | |
| SF-SGD | 0.1 | >30 |
| SF-SGD | 0.5 | 22.89 |
| SF-SGD | 1.0 | 19.42 |
| SF-SGD | 1.5 | 18.32 |
| SF-SGD | 2.0 | 17.98 |
| SF-SGD | 3.0 | 17.96 |
| SF-SGD | 4.0 | 18.09 |
| SF-SGD | 5.0 | 17.51 |
| *Nesterov Learning Rate Sweep (Cosine Schedule)* | | |
| Nesterov | 0.3 | 17.16 |
| Nesterov | 0.5 | 17.06 |
| Nesterov | 0.7 | 16.93 |
| Nesterov | 0.9 | 17.19 |
| Nesterov | 1.1 | 17.56 |
| *SGD Learning Rate Sweep* | | |
| SGD | 0.3 (fixed) | 21.04 |
| SGD | 0.3 (cosine) | 17.68 |
| SGD | 0.5 (cosine) | 16.63 |
| SGD | 0.7 (cosine) | 18.84 |
| SGD | 1.0 (cosine) | 19.21 |

Table 5: SF-SGD $\beta$ parameter sweep at 150M model size with $H = 50$, $M = 4$, and outer learning rate $\gamma = 2.0$.

| $\beta$ Value | Perplexity |
|---------------|------------|
| 0.0 | >30 |
| 0.05 | 16.88 |
| 0.1 | 16.78 |
| 0.2 | 16.89 |
| 0.4 | 17.15 |
| 0.5 | 17.35 |
| 0.7 | 17.93 |
| 0.9 | 19.07 |
| 0.95 | 19.65 |
| 0.99 | 20.51 |

# Theory

## B  Guarantees for Local SGD

First, we recall our setting and define some notation. We consider the problem of minimizing a function $f$ in a distributed setting with $M$ workers performing Local SGD. Let $x_r$ denote the global model parameters at the beginning of round $r$. Each worker $m$ initializes its local parameters as $y_{m,r,0} = x_r$ and performs $H$ local SGD steps according to

$$y_{m,r,h+1} = y_{m,r,h} - \eta g_{m,r,h},$$

where $g_{m,r,h} = \nabla f(y_{m,r,h}) + n_{m,r,h}$ is the stochastic gradient with noise $n_{m,r,h}$, and $\overline{g}_{m,r,h} = \nabla f(y_{m,r,h})$ is the true gradient. By Assumption 2 we have $\mathbb{E}[g_{m,r,h}] = \overline{g}_{m,r,h}$. After $H$ local steps, the global model update can be equivalently written as $x_{r+1} = x_r - \gamma\eta \sum_{h=0}^{H-1} g_{r,h}$ where $g_{r,h} = \frac{1}{M}\sum_{m=1}^{M} g_{m,r,h}$ is the average gradient across workers and $y_{r,h} = \frac{1}{M}\sum_{m=1}^{M} y_{m,r,h}$ is the average model. Note that these two last sequences are virtual sequences and not actually computed. We also define $x_{r,h} = x_r - \gamma\eta \sum_{h=0}^{H-1} g_{r,h}$ as an intermediate quantity used in the analysis. Table 6 summarizes some of the notation we use throughout this section.

Table 6: Key notation.

| Symbol | Description | Symbol | Description |
|--------|-------------|--------|-------------|
| $M$ | Number of nodes | $x_r$ | Global iterate at round $r$ |
| $H$ | Local steps per round | $y_{r,h}$ | Averaged local iterate |
| $R$ | Communication rounds | $g_{r,h}$ | Averaged stochastic gradient |
| $\eta$ | Inner learning rate | $L$ | Smoothness constant |
| $\gamma$ | Outer learning rate | $\sigma^2$ | Gradient variance bound |
| $\mu$ | Momentum parameter | $D$ | $\|x_0 - x_*\|$ initial distance to optimum |

### B.1  Algorithm-independent results

**Lemma 1.** *(Karimireddy et al., 2020, Lemma 6) Let $f$ be a convex and L-smooth function. Suppose that $\eta \leq \frac{2}{L}$, let $T_\eta(x) = x - \eta\nabla f(x)$. Then*

$$\|T_\eta(x) - T_\eta(y)\|^2 \leq \|x - y\|^2.$$

*Proof.* The proof is provided for completeness only. We have

$$\|T_\eta(x) - T_\eta(y)\|^2 = \|x - y\|^2 + \eta^2\|\nabla f(x) - \nabla f(y)\|^2 - 2\eta\langle x - y, \nabla f(x) - \nabla f(y)\rangle. \quad (10)$$

By the Baillon-Haddad theorem (Bauschke and Combettes, 2009) we have

$$\langle x - y, \nabla f(x) - \nabla f(y)\rangle \geq \frac{1}{L}\|\nabla f(x) - \nabla f(y)\|^2.$$

Using this in Equation (10) gives

$$\|T_\eta(x) - T_\eta(y)\|^2 \leq \|x - y\|^2 - \eta\left(\frac{2}{L} - \eta\right)\|\nabla f(x) - \nabla f(y)\|^2.$$

If $\eta \leq \frac{1}{L}$ then $\frac{2}{L} - \eta \geq 0$ and therefore $\|T_\eta(x) - T_\eta(y)\|^2 \leq \|x - y\|^2$. $\qquad\square$

**Lemma 2.** *Let $y_1, \ldots, y_n$ be real numbers. Then,*

$$\frac{1}{n}\sum_{k=1}^{n}|y_i| \leq \sqrt{\frac{1}{n}\sum_{k=1}^{n}y_i^2}.$$

*Proof.* This is just the arithmetic mean-root mean square inequality and we include the proof solely for completeness. Let $Y$ be a random variable that takes the value $y_i^2$ with probability $\frac{1}{n}$, and let $g(x) = \sqrt{x}$. Observe that

$$\frac{1}{n} \sum_{k=1}^{n} |y_i| = \mathbb{E}\left[g(Y)\right].$$

Since $g$ is a concave function, by Jensen's inequality we have that $\mathbb{E}\left[g(Y)\right] \leq g(\mathbb{E}\left[Y\right])$. Therefore,

$$\frac{1}{n} \sum_{k=1}^{n} |y_i| = \mathbb{E}\left[g(Y)\right] \leq g(\mathbb{E}\left[Y\right]) = \sqrt{\frac{1}{n} \sum_{k=1}^{n} y_i^2}.$$

$\square$

**Lemma 3.** *(Variance of Sum of Conditionally Independent Random Variables). Let $Z_1, \ldots, Z_n$ be random variables such that $Z_i$ satisfies*

$$\mathbb{E}_{i-1}\left[Z_i\right] = 0, \qquad \text{and,} \qquad \mathbb{E}\left[\|Z_i\|^2\right] = \sigma_i^2,$$

*where $\mathbb{E}_i\left[\cdot\right]$ denotes expectation conditional on $Z_1, Z_2, \ldots, Z_i$. Then,*

$$\mathbb{E}\left[\left\|\sum_{i=1}^{n} Z_i\right\|^2\right] = \sum_{i=1}^{n} \sigma_i^2.$$

*Proof.*

$$\mathbb{E}\left[\left\|\sum_{i=1}^{n} Z_i\right\|^2\right] = \mathbb{E}\left[\mathbb{E}_{n-1}\left[\left\|\sum_{i=1}^{n} Z_i\right\|^2\right]\right]$$

$$= \mathbb{E}\left[\mathbb{E}_{n-1}\left[\left\|\sum_{i=1}^{n-1} Z_i\right\|^2 + \|Z_n\|^2 + 2\left\langle \sum_{i=1}^{n-1} Z_i, Z_n \right\rangle\right]\right]$$

$$= \mathbb{E}\left[\mathbb{E}_{n-1}\left[\left\|\sum_{i=1}^{n-1} Z_i\right\|^2\right] + \sigma_n^2\right].$$

The cross-term $\mathbb{E}_{n-1}\left[2\left\langle \sum_{i=1}^{n-1} Z_i, Z_n \right\rangle\right]$ vanishes because $\mathbb{E}_{n-1}\left[Z_n\right] = 0$ and $\sum_{i=1}^{n-1} Z_i$ is measurable with respect to the sigma-algebra generated by $Z_1, \ldots, Z_{n-1}$. Continuing,

$$\mathbb{E}\left[\left\|\sum_{i=1}^{n} Z_i\right\|^2\right] = \mathbb{E}\left[\left\|\sum_{i=1}^{n-1} Z_i\right\|^2\right] + \sigma_n^2.$$

Recursing we get,

$$\mathbb{E}\left[\left\|\sum_{i=1}^{n} Z_i\right\|^2\right] = \sum_{i=1}^{n} \sigma_i^2.$$

This completes the proof. $\square$

**Lemma 4.** *(Ivgi, Hinder, and Carmon, 2023, Lemma 7). Let $S$ be the set of nonnegative and nondecreasing sequences. Let $y_1, y_2, \ldots$ be a sequence in $S$. Let $C_t \in \mathcal{F}_{t-1}$ for all $t = 1, 2, \ldots, T$ and let $X_t$ be a martingale difference sequence adapted to $\mathcal{F}_t$ such that $|X_t| \leq C_t$ with probability $1$ for $t = 1, 2, \ldots, T$. Then for all $\delta \in (0, 1)$ and $\hat{X}_t \in \mathcal{F}_{t-1}$ such that $\left|\hat{X}_t\right| \leq C_t$ with probability $1$, we have that with probability at least $1 - \delta - \text{Prob}\left(\exists t \leq T \mid C_t > c\right)$ that for all $c > 0$*

$$\left|\sum_{i=1}^{t} y_i X_i\right| \leq 8 y_t \sqrt{\theta_{t,\delta} \sum_{i=1}^{t} (X_i - \hat{X}_i)^2 + c^2 \theta_{t,\delta}^2},$$

*where $\theta_{t,\delta} = \log \frac{60 \log 6t}{\delta}$.*

**Lemma 5.** *Suppose we have*

$$r_{k+1} \leq (1+a)r_k - b\delta_k + c$$

*Then,*

$$\min_j \delta_j \leq \frac{r_0 e^{aK}}{bK} + \frac{c}{b}.$$

*Proof.* Let $w_{k+1} = \frac{w_k}{1+a}$. We have

$$w_{k+1}r_{k+1} \leq (1+a)w_{k+1}r_k - bw_{k+1}\delta_k + cw_{k+1}$$
$$= w_k r_k - bw_{k+1}\delta_k + cw_{k+1}.$$

Telescoping,

$$w_K r_K \leq w_0 r_0 - b\sum_{j=0}^{K-1} w_{j+1}\delta_j + c\sum_{j=0}^{K-1} w_{j+1}.$$

Rearranging,

$$\frac{1}{\sum_{j=0}^{K-1} w_{j+1}} \sum_{j=0}^{K-1} w_{j+1}\delta_j \leq \frac{w_0 r_0}{b\sum_{j=0}^{K-1} w_{j+1}} + \frac{c}{b}.$$

We have $w_k = \frac{w_{k-1}}{1+a} = \frac{w_0}{(1+a)^k}$. Therefore,

$$\sum_{j=0}^{K-1} w_{j+1} = \sum_{j=0}^{K-1} \frac{w_0}{(1+a)^{k+1}}$$
$$\geq \sum_{j=0}^{K-1} \frac{w_0}{(1+a)^K}$$
$$= \frac{w_0 K}{(1+a)^K}.$$

Therefore,

$$\frac{1}{\sum_{j=0}^{K-1} w_{j+1}} \sum_{j=0}^{K-1} w_{j+1}\delta_j \leq \frac{r_0(1+a)^K}{bK} + \frac{c}{b}.$$

Finally, it remains to use that $1 + a \leq e^a$. $\qquad\square$

## B.2  Convergence guarantees without momentum

We begin with a lemma that establishes the regret of the local optimizer. Often the regret is measured against the optimal point (like $x_*$) but here we instead utilize it against the *initial* point $y_{r,0} = x_r$.

**Lemma 6** (Regret against starting point). *For any learning rate $\eta > 0$, the inner product between the displacement from the initial average iterate and the average gradient satisfies,*

$$\sum_{h=0}^{H-1} \langle y_{r,h} - y_{r,0}, g_{r,h} \rangle \leq \frac{\eta}{2} \sum_{h=0}^{H-1} \|g_{r,h}\|^2 - \frac{1}{2\eta}\|y_{r,H} - y_{r,0}\|^2.$$

*Proof.* We begin by using that $y_{r,h+1} = y_{r,h} - \eta g_{r,h}$ and expanding the square as

$$\|y_{r,h+1} - y_{r,0}\|^2 = \|y_{r,h} - \eta g_{r,h} - y_{r,0}\|^2$$
$$= \|y_{r,h} - y_{r,0}\|^2 + \eta^2\|g_{r,h}\|^2 - 2\eta \langle y_{r,h} - y_{r,0}, g_{r,h} \rangle.$$

Rearranging to isolate the inner product term, we obtain

$$\langle y_{r,h} - y_{r,0}, g_{r,h} \rangle = \frac{\|y_{r,h} - y_{r,0}\|^2 - \|y_{r,h+1} - y_{r,0}\|^2}{2\eta} + \frac{\eta}{2}\|g_{r,h}\|^2.$$

Summing over $h$ from 0 to $H - 1$,

$$\sum_{h=0}^{H-1} \langle y_{r,h} - y_{r,0}, g_{r,h} \rangle = \sum_{h=0}^{H-1} \left( \frac{\|y_{r,h} - y_{r,0}\|^2 - \|y_{r,h+1} - y_{r,0}\|^2}{2\eta} + \frac{\eta}{2} \|g_{r,h}\|^2 \right)$$

$$= \frac{1}{2\eta} \sum_{h=0}^{H-1} (\|y_{r,h} - y_{r,0}\|^2 - \|y_{r,h+1} - y_{r,0}\|^2) + \frac{\eta}{2} \sum_{h=0}^{H-1} \|g_{r,h}\|^2.$$

The first sum telescopes

$$\sum_{h=0}^{H-1} (\|y_{r,h} - y_{r,0}\|^2 - \|y_{r,h+1} - y_{r,0}\|^2) = \|y_{r,0} - y_{r,0}\|^2 - \|y_{r,H} - y_{r,0}\|^2$$

$$= -\|y_{r,H} - y_{r,0}\|^2.$$

Therefore,

$$\sum_{h=0}^{H-1} \langle y_{r,h} - y_{r,0}, g_{r,h} \rangle = -\frac{\|y_{r,H} - y_{r,0}\|^2}{2\eta} + \frac{\eta}{2} \sum_{h=0}^{H-1} \|g_{r,h}\|^2$$

$$\leq \frac{\eta}{2} \sum_{h=0}^{H-1} \|g_{r,h}\|^2 - \frac{\|y_{r,H} - y_{r,0}\|^2}{2\eta}.$$

$\square$

**Lemma 7.** *(Local client drift bound). Suppose that Assumptions 1 and 2 hold. Then in Algorithm GEN-LOC-SGD for all $r$ and $h$, if $\eta \leq \frac{1}{L}$, then*

$$\mathbb{E} \left[ \frac{1}{M^2} \sum_{m,s=1}^{M} \|y_{m,r,h} - y_{s,r,h}\|^2 \right] \leq 2\eta^2 \sigma^2 h.$$

*Proof.* Let $\tilde{T}_\eta(y_{m,r,h}) = y_{m,r,h} - \eta g_{m,r,h}$ where $g_{m,r,h}$ is the stochastic gradient, and $T_\eta(y_{m,r,h}) = y - \eta \bar{g}_{m,r,h}$ is the corresponding expected gradient update. We have

$$y_{m,r,h+1} - y_{s,r,h+1} = \tilde{T}_\eta(y_{m,r,h}) - \tilde{T}_\eta(y_{s,r,h})$$

$$= T_\eta(y_{m,r,h}) - T_\eta(y_{s,r,h}) + \left[ \tilde{T}_\eta(y_{m,r,h}) - \tilde{T}_\eta(y_{s,r,h}) - (T_\eta(y_{m,r,h}) - T_\eta(y_{s,r,h})) \right]$$

$$= T_\eta(y_{m,r,h}) - T_\eta(y_{s,r,h}) + [\xi_{m,r,h} - \xi_{s,r,h}],$$

where $\xi_{m,r,h} = \tilde{T}_\eta(y_{m,r,h}) - T_\eta(y_{m,r,h}) = -\eta n_{m,r,h}$ is the noise term. Define $\mathcal{V}_{r,h} = \frac{1}{M^2} \sum_{m,s=1}^{M} \|y_{m,r,h} - y_{s,r,h}\|^2$. It follows that

$$\mathcal{V}_{r,h+1} = \frac{1}{M^2} \sum_{m,s=1}^{M} \|y_{m,r,h+1} - y_{s,r,h+1}\|^2$$

$$= \frac{1}{M^2} \sum_{m,s=1}^{M} \left[ \|T_\eta(y_{m,r,h}) - T_\eta(y_{s,r,h})\|^2 + \|\xi_{m,r,h} - \xi_{s,r,h}\|^2 \right.$$

$$\left. + 2 \langle T_\eta(y_{m,r,h}) - T_\eta(y_{s,r,h}), \xi_{m,r,h} - \xi_{s,r,h} \rangle \right].$$

Taking conditional expectation gives

$$\mathbb{E}_{r,h} [\mathcal{V}_{r,h+1}] = \frac{1}{M^2} \sum_{m,s=1}^{M} \left[ \|T_\eta(y_{m,r,h}) - T_\eta(y_{s,r,h})\|^2 + \mathbb{E}_h \left[ \|\xi_{m,r,h} - \xi_{s,r,h}\|^2 \right] \right].$$

Finally, using the fact that $\|T_\eta(x) - T_\eta(y)\|^2 \le \|x - y\|^2$ whenever $\eta \le \frac{2}{L}$ (Lemma 1) and Assumption 2, we get

$$\mathbb{E}_{r,h}[\mathcal{V}_{r,h+1}] \le \frac{1}{M^2} \sum_{m,s=1}^{M} \left[ \|y_{m,r,h} - y_{s,r,h}\|^2 + 2\eta^2\sigma^2 \right]$$

$$= \mathcal{V}_{r,h} + 2\eta^2\sigma^2.$$

Therefore by taking unconditional expectation and recursing from $h = 0$ where all local iterates are equal to $x_r$ (so $\mathcal{V}_{r,0} = 0$), we get $\mathbb{E}[\mathcal{V}_{r,h}] \le 2\eta^2\sigma^2 h$. □

*Proof of Theorem 1.* W begin by analyzing how the squared distance to the optimal solution changes after one round of communication. From the update rule, we have,

$$\|x_{r+1} - x_*\|^2 = \|x_r - x_*\|^2 - 2\eta\gamma \sum_{h=0}^{H-1} \langle x_r - x_*, g_{r,h} \rangle + \eta^2\gamma^2 \left\| \sum_{h=0}^{H-1} g_{r,h} \right\|^2. \qquad (11)$$

We rewrite the inner product term as

$$-\langle x_r - x_*, g_{r,h} \rangle = \langle x_* - x_r, g_{r,h} \rangle$$

$$= \langle x_* - y_{r,h}, g_{r,h} \rangle + \langle y_{r,h} - x_r, g_{r,h} \rangle.$$

Summing over all local steps we obtain

$$-\sum_{h=0}^{H-1} \langle x_r - x_*, g_{r,h} \rangle = \sum_{h=0}^{H-1} \langle x_* - y_{r,h}, g_{r,h} \rangle + \sum_{h=0}^{H-1} \langle y_{r,h} - x_r, g_{r,h} \rangle.$$

Applying Lemma 6 we get

$$-\sum_{h=0}^{H-1} \langle x_r - x_*, g_{r,h} \rangle = \sum_{h=0}^{H-1} \langle x_* - y_{r,h}, g_{r,h} \rangle - \frac{\|y_{r,H} - y_{r,0}\|^2}{2\eta} + \frac{\eta}{2} \sum_{h=0}^{H-1} \|g_{r,h}\|^2. \qquad (12)$$

Observe that since $y_{r,H} - y_{r,0} = -\eta \sum_{h=0}^{H-1} g_{r,h}$, Equation (12) becomes,

$$-\sum_{h=0}^{H-1} \langle x_r - x_*, g_{r,h} \rangle = \sum_{h=0}^{H-1} \langle x_* - y_{r,h}, g_{r,h} \rangle - \frac{\eta}{2} \left\| \sum_{h=0}^{H-1} g_{r,h} \right\|^2 + \frac{\eta}{2} \sum_{h=0}^{H-1} \|g_{r,h}\|^2.$$

Plugging this back into Equation (11),

$$\|x_{r+1} - x_*\|^2 \le \|x_r - x_*\|^2 + 2\eta\gamma \sum_{h=0}^{H-1} \langle x_* - y_{r,h}, g_{r,h} \rangle$$

$$+ \gamma\eta^2 \sum_{h=0}^{H-1} \|g_{r,h}\|^2 + \eta^2\gamma(\gamma - 1) \left\| \sum_{h=0}^{H-1} g_{r,h} \right\|^2.$$

Let us take expectation conditional on $x_1, \ldots, x_r$,

$$\mathbb{E}_r\left[\|x_{r+1} - x_*\|^2\right] \le \|x_r - x_*\|^2 + 2\eta\gamma \sum_{h=0}^{H-1} \mathbb{E}_r\left[\langle x_* - y_{r,h}, g_{r,h} \rangle\right]$$

$$+ \gamma\eta^2 \sum_{h=0}^{H-1} \mathbb{E}_r\left[\|g_{r,h}\|^2\right] + \eta^2\gamma(\gamma - 1)\mathbb{E}_r\left[\left\| \sum_{h=0}^{H-1} g_{r,h} \right\|^2\right]. \qquad (13)$$

For the squared norm of the average gradient:

$$\mathbb{E}_r\left[\|g_{r,h}\|^2\right] = \mathbb{E}_r\left[\mathbb{E}_{r,h-1}\left[\|g_{r,h}\|^2\right]\right]$$

$$= \mathbb{E}_r\left[\mathbb{E}_{r,h-1}\left[\|g_{r,h} - \bar{g}_{r,h}\|^2\right] + \|\bar{g}_{r,h}\|^2\right]$$

$$= \frac{\sigma^2}{M} + \mathbb{E}_r\left[\|\bar{g}_{r,h}\|^2\right],$$

where we use $\mathbb{E}_{r,h-1}[\cdot]$ to denote expectation conditional on the $\sigma$-algebra generated by all the stochastic gradients up to and including step $h - 1$. Substituting this into Equation (13),

$$\mathbb{E}_r\left[\|x_{r+1} - x_*\|^2\right] \leq \|x_r - x_*\|^2 + 2\eta\gamma \sum_{h=0}^{H-1} \mathbb{E}_r\left[\langle x_* - y_{r,h}, g_{r,h}\rangle\right] + \frac{\gamma\eta^2 H\sigma^2}{M}$$
$$+ \gamma\eta^2 \sum_{h=0}^{H-1} \mathbb{E}_r\left[\|\overline{g}_{r,h}\|^2\right] + \eta^2\gamma(\gamma - 1)\mathbb{E}_r\left[\left\|\sum_{h=0}^{H-1} g_{r,h}\right\|^2\right]. \tag{14}$$

Now we bound the inner product term:

$$\mathbb{E}_r\left[\langle x_* - y_{r,h}, g_{r,h}\rangle\right] = \mathbb{E}_r\left[\mathbb{E}_{h-1}\left[\langle x_* - y_{r,h}, g_{r,h}\rangle\right]\right]$$
$$= \mathbb{E}_r\left[\langle x_* - y_{r,h}, \overline{g}_{r,h}\rangle\right]$$
$$= \frac{1}{M}\sum_{m=1}^{M} \mathbb{E}_r\left[\langle x_* - y_{r,h}, \overline{g}_{m,r,h}\rangle\right]$$
$$= \frac{1}{M}\sum_{m=1}^{M} \mathbb{E}_r\left[\langle x_* - y_{m,r,h} + y_{m,r,h} - y_{r,h}, \overline{g}_{m,r,h}\rangle\right]$$
$$= \frac{1}{M}\sum_{m=1}^{M} \mathbb{E}_r\left[\langle x_* - y_{m,r,h}, \overline{g}_{m,r,h}\rangle\right] + \frac{1}{M}\sum_{m=1}^{M} \mathbb{E}_r\left[\langle y_{m,r,h} - y_{r,h}, \overline{g}_{m,r,h}\rangle\right].$$

Using Young's inequality for the second term,

$$\mathbb{E}_r\left[\langle x_* - y_{r,h}, g_{r,h}\rangle\right] \tag{15}$$
$$\leq \frac{1}{M}\sum_{m=1}^{M} \mathbb{E}_r\left[\langle x_* - y_{m,r,h}, \overline{g}_{m,r,h}\rangle\right] + \frac{1}{M}\sum_{m=1}^{M} \mathbb{E}_r\left[\frac{\|y_{m,r,h} - y_{r,h}\|^2}{2\alpha} + \frac{\alpha}{2}\|\overline{g}_{m,r,h}\|^2\right]$$
$$= \frac{1}{M}\sum_{m=1}^{M} \mathbb{E}_r\left[\langle x_* - y_{m,r,h}, \overline{g}_{m,r,h}\rangle\right] + \frac{V_{r,h}}{2\alpha} + \frac{\alpha}{2M}\sum_{m=1}^{M} \mathbb{E}_r\left[\|\overline{g}_{m,r,h}\|^2\right], \tag{16}$$

where $V_{r,h} = \frac{1}{M}\sum_{m=1}^{M} \mathbb{E}_r\left[\|y_{m,r,h} - y_{r,h}\|^2\right]$ by definition. By the convexity of $f$,

$$\langle x_* - y_{m,r,h}, \overline{g}_{m,r,h}\rangle = \langle x_* - y_{m,r,h}, \nabla f(y_{m,r,h})\rangle$$
$$\leq f(x_*) - f(y_{m,r,h})$$
$$= -\left(f(y_{m,r,h}) - f(x_*)\right). \tag{17}$$

For the variance term, when $\eta \leq \frac{1}{L}$ we use Lemma 7

$$V_{r,h} = \frac{1}{M}\sum_{m=1}^{M} \mathbb{E}_r\left[\|y_{m,r,h} - y_{r,h}\|^2\right]$$
$$\leq \frac{1}{M}\sum_{m=1}^{M} \frac{1}{M}\sum_{s=1}^{M} \mathbb{E}_r\left[\|y_{m,r,h} - y_{s,r,h}\|^2\right]$$
$$= \frac{1}{M^2}\sum_{m=1}^{M}\sum_{s=1}^{M} \mathbb{E}_r\left[\|y_{m,r,h} - y_{s,r,h}\|^2\right]$$
$$\leq 2\eta^2\sigma^2 h \leq 2\eta^2\sigma^2 H. \tag{18}$$

By smoothness,

$$\|\overline{g}_{m,r,h}\|^2 = \|\nabla f(y_{m,r,h})\|^2 \leq 2L(f(y_{m,r,h}) - f(x_*)). \tag{19}$$

Plugging Equations (17) to (19) back into Equation (16) we get

$$\mathbb{E}_r\left[\langle x_* - y_{r,h}, g_{r,h}\rangle\right] \leq \frac{-(1-\alpha L)}{M}\sum_{m=1}^{M}\left(\mathbb{E}_r\left[f(y_{m,r,h})\right] - f(x_*)\right) + \frac{\eta^2\sigma^2 H}{\alpha}. \qquad (20)$$

Substituting (20) back into our main recursion (Equation (13)),

$$\mathbb{E}_r\left[\|x_{r+1} - x_*\|^2\right] \leq \|x_r - x_*\|^2 - \frac{2\eta\gamma(1-\alpha L)}{M}\sum_{h=0}^{H-1}\sum_{m=1}^{M}\left(\mathbb{E}_r\left[f(y_{m,r,h})\right] - f(x_*)\right) + \frac{2\eta^3\gamma\sigma^2 H^2}{\alpha}$$

$$+ \frac{\gamma\eta^2 H\sigma^2}{M} + \gamma\eta^2\sum_{h=0}^{H-1}\mathbb{E}_r\left[\|\bar{g}_{r,h}\|^2\right] + \eta^2\gamma(\gamma-1)\mathbb{E}_r\left[\left\|\sum_{h=0}^{H-1}g_{r,h}\right\|^2\right]. \tag{21}$$

We now have two cases. **Case 1**. If $\gamma \geq 1$, then we have by Lemma 3 and Jensen's inequality applied to $\|\cdot\|^2$,

$$\mathbb{E}_r\left[\left\|\sum_{h=0}^{H-1}g_{r,h}\right\|^2\right] = \mathbb{E}_r\left[\left\|\sum_{h=0}^{H-1}(g_{r,h} - \mathbb{E}_r\left[g_{r,h}\right])\right\|^2\right] + \left\|\sum_{h=0}^{H-1}(\mathbb{E}_r\left[g_{r,h}\right])\right\|^2$$

$$= \mathbb{E}_r\left[\left\|\sum_{h=0}^{H-1}(g_{r,h} - \mathbb{E}_r\left[g_{r,h}\right])\right\|^2\right] + \left\|\sum_{h=0}^{H-1}(\mathbb{E}_r\left[\mathbb{E}_{r,h-1}\left[g_{r,h}\right]\right])\right\|^2$$

$$= \mathbb{E}_r\left[\left\|\sum_{h=0}^{H-1}(g_{r,h} - \mathbb{E}_r\left[g_{r,h}\right])\right\|^2\right] + \left\|\sum_{h=0}^{H-1}\mathbb{E}_r\left[\bar{g}_{r,h}\right]\right\|^2$$

$$\leq \frac{\sigma^2 H}{M} + \mathbb{E}_r\left[\left\|\sum_{h=0}^{H-1}\bar{g}_{r,h}\right\|^2\right]$$

$$\leq \frac{\sigma^2 H}{M} + H\sum_{h=0}^{H-1}\mathbb{E}_r\left[\|\bar{g}_{r,h}\|^2\right]. \tag{22}$$

Using Jensen's inequality and smoothness we have

$$\mathbb{E}_r\left[\|\bar{g}_{r,h}\|^2\right] = \mathbb{E}_r\left[\left\|\frac{1}{M}\sum_{m=1}^{M}\nabla f(y_{m,r,h})\right\|^2\right]$$

$$\leq \frac{1}{M}\sum_{m=1}^{M}\mathbb{E}_r\left[\|\nabla f(y_{m,r,h})\|^2\right]$$

$$\leq \frac{2L}{M}\sum_{m=1}^{M}\mathbb{E}_r\left[f(y_{m,r,h}) - f(x_*)\right]. \tag{23}$$

Using Equations (22) and (23) into Equation (21) we get

$$
\begin{aligned}
\mathbb{E}_r\left[\|x_{r+1} - x_*\|^2\right] &\le \|x_r - x_*\|^2 \\
&- \frac{2\eta\gamma(1-\alpha L) - 2L\gamma\eta^2(1+(\gamma-1)H)}{M} \sum_{h=0}^{H-1}\sum_{m=1}^{M}\left(\mathbb{E}_r\left[f(y_{m,r,h})\right] - f(x_*)\right) + \frac{2\eta^3\gamma\sigma^2 H^2}{\alpha} \\
&+ \frac{\gamma^2\eta^2 H\sigma^2}{M}. \\
&= \|x_r - x_*\|^2 - \frac{2\eta\gamma\left[1-\alpha L - L\eta(1+(\gamma-1)H)\right]}{M} \sum_{h=0}^{H-1}\sum_{m=1}^{M}\left(\mathbb{E}_r\left[f(y_{m,r,h})\right] - f(x_*)\right) \\
&+ \frac{2\eta^3\gamma\sigma^2 H^2}{\alpha} + \frac{\gamma^2\eta^2 H\sigma^2}{M}. \\
&= \|x_r - x_*\|^2 - 2\eta\gamma H(1-\alpha L - L\eta(1+(\gamma-1)H))\mathbb{E}_r\left[\hat{\delta}_{r+1}\right] + \frac{2\eta^3\gamma\sigma^2 H^2}{\alpha} + \frac{\eta^2\gamma^2 H\sigma^2}{M},
\end{aligned}
$$
(24)

where in the last line we defined

$$
\hat{\delta}_{r+1} = \frac{1}{MH} \sum_{h=0}^{H-1}\sum_{m=1}^{M}\left(f(y_{m,r,h}) - f(x_*)\right) \tag{25}
$$

**Case 2.** If $\gamma \le 1$, then we can simply drop the last term in Equation (21) and use Equation (19) to get

$$
\begin{aligned}
\mathbb{E}_r\left[\|x_{r+1} - x_*\|^2\right] &\le \|x_r - x_*\|^2 - \frac{2\eta\gamma(1-\alpha L - \eta L)}{M}\sum_{h=0}^{H-1}\sum_{m=1}^{M}\left(\mathbb{E}_r\left[f(y_{m,r,h})\right] - f(x_*)\right) \\
&+ \frac{2\eta^3\gamma\sigma^2 H^2}{\alpha} + \frac{\gamma\eta^2 H\sigma^2}{M} \\
&= \|x_r - x_*\|^2 - 2\eta\gamma H(1-\alpha L - \eta L)\mathbb{E}_r\left[\hat{\delta}_{r+1}\right] + \frac{2\eta^3\gamma\sigma^2 H^2}{\alpha} + \frac{\gamma\eta^2 H\sigma^2}{M},
\end{aligned}
$$
(26)

where in Equation (26) we again used the definition in Equation (25). Looking at both Equations (24) and (26) and taking the maximum we get that for *any* $\gamma$,

$$
\begin{aligned}
\mathbb{E}_r\left[\|x_{r+1} - x_*\|^2\right] &\le \|x_r - x_*\|^2 - 2\eta\gamma H(1-\alpha L - \eta L(1+(\gamma-1)_+ H))\mathbb{E}_r\left[\hat{\delta}_{r+1}\right] \\
&+ \frac{2\eta^3\gamma\sigma^2 H^2}{\alpha} + \frac{\eta^2\max\{\gamma^2,\gamma\}H\sigma^2}{M},
\end{aligned}
$$

where $(x)_+ = \max(x,0)$ is the ReLU function. Putting $\alpha = \frac{1}{2L}$ we get

$$
\begin{aligned}
\mathbb{E}_r\left[\|x_{r+1} - x_*\|^2\right] &\le \|x_r - x_*\|^2 - \eta\gamma H(1-2\eta L(1+(\gamma-1)_+ H))\mathbb{E}_r\left[\hat{\delta}_{r+1}\right] \\
&+ 4L\eta^3\gamma\sigma^2 H^2 + \frac{\eta^2\max\{\gamma^2,\gamma\}H\sigma^2}{M}.
\end{aligned}
$$

Under the requirement that the stepsizes $\eta, \gamma$ satisfy

$$
\eta L(1+(\gamma-1)_+ H) \le \frac{1}{4},
$$

we obtain our recursion

$$
\mathbb{E}_r\left[\|x_{r+1} - x_*\|^2\right] \le \|x_r - x_*\|^2 - \frac{\eta\gamma H}{2}\mathbb{E}_r\left[\hat{\delta}_{r+1}\right] + 4L\eta^3\gamma\sigma^2 H^2 + \frac{\eta^2\max\{\gamma^2,\gamma\}H\sigma^2}{M}.
$$

Taking unconditional expectations and rearranging we obtain,

$$
\mathbb{E}\left[\hat{\delta}_{r+1}\right] \le \frac{2}{\gamma\eta H}\left[\mathbb{E}\left[\|x_r - x_*\|^2\right] - \mathbb{E}\left[\|x_{r+1} - x_*\|^2\right]\right] + 8L\eta^2\sigma^2 H + \frac{2\eta\max(\gamma,1)\sigma^2}{M}.
$$

Summing up both sides as $r$ varies from $0$ to $R-1$ and dividing by $1/R$ we get

$$\frac{1}{R}\sum_{r=0}^{R-1}\mathbb{E}\left[\hat{\delta}_{r+1}\right] \le \frac{2}{\gamma\eta RH}\left[\|x_0 - x_*\|^2 - \mathbb{E}\left[\|x_R - x_*\|^2\right]\right] + 8L\eta^2\sigma^2 H + \frac{2\eta\max(\gamma,1)\sigma^2}{M}.$$

Dropping the negative term and using Jensen's inequality gives

$$\mathbb{E}\left[f\left(\frac{1}{MRH}\sum_{r=0}^{R-1}\sum_{h=0}^{H-1}\sum_{m=1}^{M}f(y_{m,r,h})\right)\right] - f(x_*) \le \frac{1}{R}\sum_{r=0}^{R-1}\mathbb{E}\left[\hat{\delta}_{r+1}\right]$$

$$\le \frac{2\|x_0 - x_*\|^2}{\gamma\eta RH} + 8L\eta^2\sigma^2 H + \frac{2\eta\max(\gamma,1)\sigma^2}{M},$$

and this is the statement of our theorem. $\qquad\square$

## B.3   Convergence guarantees with momentum

*Proof of Theorem 2.* We analyze the momentum variant of Local SGD:

$$x_{r+1} = x_r - \eta\gamma\left(\sum_{h=0}^{H-1} g_{r,h}\right) + \mu(x_r - x_{r-1}).$$

Define

$$z_r = x_r + \frac{\mu}{1-\mu}(x_r - x_{r-1}).$$

Then

$$z_{r+1} = z_r - \frac{\eta\gamma}{1-\mu}\sum_{h=0}^{H-1} g_{r,h}.$$

We have

$$\|z_{r+1} - x_*\|^2 = \|z_r - x_*\|^2 + \frac{\eta^2\gamma^2}{(1-\mu)^2}\left\|\sum_{h=0}^{H-1} g_{r,h}\right\|^2 - \frac{2\eta\gamma}{1-\mu}\sum_{h=0}^{H-1}\langle z_r - x_*, g_{r,h}\rangle$$

$$= \|z_r - x_*\|^2 + \frac{\eta^2\gamma^2}{(1-\mu)^2}\left\|\sum_{h=0}^{H-1} g_{r,h}\right\|^2 - \frac{2\eta\gamma}{1-\mu}\sum_{h=0}^{H-1}\langle x_r - x_*, g_{r,h}\rangle \qquad (27)$$

$$- \frac{2\eta\gamma\mu}{1-\mu}\sum_{h=0}^{H-1}\langle x_r - x_{r-1}, g_{r,h}\rangle.$$

Following the same proof as Theorem 1, we can bound (in expectation)

$$-\frac{2\eta\gamma}{1-\mu}\sum_{h=0}^{H-1}\mathbb{E}_r\left[\langle x_r - x_*, g_{r,h}\rangle\right] + \frac{\eta^2\gamma^2}{(1-\mu)^2}\mathbb{E}_r\left[\left\|\sum_{h=0}^{H-1} g_{r,h}\right\|^2\right] \le -\frac{\eta\gamma H}{2(1-\mu)}\mathbb{E}_r\left[\hat{\delta}_{r+1}\right]$$

$$+ 4L\eta^3\frac{\gamma}{1-\mu}\sigma^2 H^2 + \frac{\eta^2 H\sigma^2}{M}\max\left(\left(\frac{\gamma}{1-\mu}\right)^2, \frac{\gamma}{1-\mu}\right), \qquad (28)$$

because the local optimization procedure is the same– the same analysis holds line-by-line, only replacing $\gamma$ by $\frac{\gamma}{1-\mu}$, and requiring instead that

$$\eta L\left(1 + \left(\frac{\gamma}{1-\mu} - 1\right)_+ H\right) \le \frac{1}{4}. \qquad (29)$$

Using Equation (28) in Equation (27) (after taking expectation in the latter) we obtain

$$\mathbb{E}_r \left[ \|z_{r+1} - x_*\|^2 \right] \leq \|z_r - x_*\|^2 - \frac{\eta\gamma H}{2(1-\mu)} \mathbb{E}_r \left[ \hat{\delta}_{r+1} \right] + 4L\eta^3 \frac{\gamma\sigma^2 H^2}{1-\mu}$$

$$+ \frac{\eta^2 H \sigma^2}{M} \max\left( \left( \frac{\gamma}{1-\mu} \right)^2, \frac{\gamma}{1-\mu} \right) - \frac{2\eta\gamma\mu}{1-\mu} \sum_{h=0}^{H-1} \langle x_r - x_{r-1}, \overline{g}_{r,h} \rangle . \tag{30}$$

In the following, we use the shorthand $G_r \overset{\text{def}}{=} \sum_{h=0}^{H-1} g_{r,h}$. We now proceed to bound $\sum_{h=0}^{H-1} \langle x_{r-1} - x_r, g_{r,h} \rangle = \langle x_{r-1} - x_r, G_r \rangle$ without using the bounded iterates assumption. We note that by definition:

$$x_r - x_{r-1} = -\eta\gamma G_{r-1} + \mu(x_{r-1} - x_{r-2}).$$

Expanding this out recursively, we get the following formula:

$$x_r - x_{r-1} = -\eta\gamma \sum_{s=0}^{r-1} \mu^{r-1-s} G_s.$$

For our analysis, we'll bound the inner product

$$\langle x_{r-1} - x_r, G_r \rangle = \left\langle \eta\gamma \sum_{s=0}^{r-1} \mu^{r-1-s} G_s, G_r \right\rangle$$

$$= \eta\gamma \sum_{s=0}^{r-1} \mu^{r-1-s} \langle G_s, G_r \rangle$$

We will actually bound the sum of the momentum terms over $r$, i.e. $\sum_r \langle x_{r-1} - x_r, G_r \rangle$. We have

$$\sum_r \langle x_{r-1} - x_r, G_r \rangle = \frac{\eta\gamma}{\mu} \sum_r \sum_{s<r} \langle \mu^{r-s} G_s, G_r \rangle$$

$$= \frac{\eta\gamma}{2\mu} \left[ \sum_r \sum_s \left\langle \mu^{|r-s|} G_s, G_r \right\rangle - \sum_r \| G_r \|^2 \right].$$

To bound the first term above, let $A$ be the $R \times R$ matrix whose $(r,s)$th entry equals $\mu^{|r-s|}$, and let $\Gamma = [G_1|G_2|\dots|G_R]$. Then

$$\sum_r \sum_s \left\langle \mu^{|r-s|} G_s, G_r \right\rangle = \text{Tr}(\Gamma A \Gamma^\top).$$

We now apply the Gershgorin circle theorem to bound this sum, observe that largest sum of absolute values of entries in a row satisfy

$$1 + 2 \sum_{r=1}^{(R-1)/2} \mu^r = 1 + 2\mu \frac{1 - \mu^{(R-1)/2}}{1-\mu} = \frac{1 + \mu - 2\mu^{(R+1)/2}}{1-\mu} \leq \frac{1+\mu}{1-\mu}.$$

Then, we have

$$\text{Tr}(\Gamma A \Gamma^\top) \leq \frac{1+\mu}{1-\mu} \text{Tr}(\Gamma \Gamma^\top) = \frac{1+\mu}{1-\mu} \sum_r \|G_r\|^2.$$

Therefore, taking expectations we have

$$-\frac{2\eta\gamma\mu}{1-\mu} \sum_{r=0}^{R-1} \sum_{h=0}^{H-1} \mathbb{E}\left[ \langle x_r - x_{r-1}, g_{r,h} \rangle \right] = \frac{2\eta\gamma\mu}{1-\mu} \sum_{r=0}^{R-1} \mathbb{E}\left[ \langle x_{r-1} - x_r, G_r \rangle \right]$$

$$\leq \frac{2\eta\gamma\mu}{1-\mu} \frac{\eta\gamma}{1-\mu} \sum_{r=0}^{R-1} \mathbb{E}\left[ \left\| \sum_{h=0}^{H-1} g_{r,h} \right\|^2 \right]. \tag{31}$$

Using Lemma 3 we have

$$\mathbb{E}\left[\left\|\sum_{h=0}^{H-1} g_{r,h}\right\|^2\right] \leq \frac{\sigma^2 H}{M} + \mathbb{E}\left[\left\|\sum_{h=0}^{H-1} \overline{g}_{r,h}\right\|^2\right]$$

$$\leq \frac{\sigma^2 H}{M} + H \sum_{h=0}^{H-1} \mathbb{E}\left[\|\overline{g}_{r,h}\|^2\right]$$

$$\leq \frac{\sigma^2 H}{M} + 2LH^2 \mathbb{E}\left[\hat{\delta}_{r+1}\right],$$

where in the last line we used Jensen's inequality and smoothness. Using this result in Equation (31) we get

$$-\frac{2\eta\gamma\mu}{1-\mu}\sum_{r=0}^{R-1}\sum_{h=0}^{H-1}\mathbb{E}\left[\langle x_r - x_{r-1}, g_{r,h}\rangle\right] \leq \frac{\eta\gamma}{2(1-\mu)}\frac{4\eta\gamma\mu}{1-\mu}\left[\frac{\sigma^2 RH}{M} + 2LH^2 \sum_{r=0}^{R-1}\mathbb{E}\left[\hat{\delta}_{r+1}\right]\right].$$
(32)

Rearranging and summing up Equation (30) then using Equation (32) we have

$$\mathbb{E}\left[\|z_R - x_*\|^2\right] \leq \|z_0 - x_*\|^2 - \frac{\eta\gamma H}{2(1-\mu)}\left[1 - \frac{8\eta\gamma\mu LH}{1-\mu}\right]\sum_{r=0}^{R-1}\mathbb{E}\left[\hat{\delta}_{r+1}\right]$$

$$+ 4L\eta^3\frac{\gamma\sigma^2 H^2}{1-\mu}R + \frac{\eta^2 H\sigma^2}{M}\max\left(\left(\frac{\gamma}{1-\mu}\right)^2, \frac{\gamma}{1-\mu}\right)R + \frac{\eta\gamma H}{1-\mu}\frac{2\eta\gamma\mu}{1-\mu}\frac{\sigma^2 R}{M}.$$

Observe that under the condition

$$\frac{\eta\gamma\mu LH}{1-\mu} \leq \frac{1}{16}$$

the last inequality becomes

$$\mathbb{E}\left[\|z_R - x_*\|^2\right] \leq \|z_0 - x_*\|^2 - \frac{\eta\gamma H}{4(1-\mu)}\sum_{r=0}^{R-1}\mathbb{E}\left[\hat{\delta}_{r+1}\right]$$

$$+ 4L\eta^3\frac{\gamma\sigma^2 H^2}{1-\mu}R + \frac{\eta^2 H\sigma^2}{M}\max\left(\left(\frac{\gamma}{1-\mu}\right)^2, \frac{\gamma}{1-\mu}\right)R + \frac{\eta\gamma H}{1-\mu}\frac{2\eta\gamma\mu}{1-\mu}\frac{\sigma^2 R}{M}.$$

Continuing the proof and rearranging we get

$$\frac{1}{R}\sum_{r=0}^{R-1}\mathbb{E}\left[\hat{\delta}_{r+1}\right] \leq \frac{4(1-\mu)\|z_0 - x_*\|^2}{\eta\gamma HR} + 16L\eta^2\sigma^2 H + \frac{4\eta\sigma^2}{M}\max\left(\frac{\gamma}{1-\mu}, 1\right) + \frac{8\eta\gamma\mu}{1-\mu}\frac{\sigma^2}{M}.$$

It remains to use Jensen's inequality. $\qquad\square$

## B.4 Acceleration proofs

We recall the algorithm under analysis as

$$y_{m,r,0} = x_r \text{ for } m = 1, \ldots, M$$
$$y_{m,r,h+1} = y_{m,r,h} - \eta g_{m,r,h} \text{ for } h = 0, 1, \ldots, H-1$$
$$u_{r+1} = x_r - \eta \sum_{h=0}^{H-1} g_{r,h}$$
$$z_{r+1} = z_r - \gamma_r \eta \sum_{h=0}^{H-1} g_{r,h}$$
$$x_{r+1} = (1 - \tau_{r+1})u_{r+1} + \tau_{r+1}z_{r+1},$$

where $g_{r,h} = \frac{1}{M}\sum_{m=1}^{M} g_{m,r,h}$, $\gamma_r = \frac{\gamma(r+1)}{2}$, and $\tau_r = \frac{2}{r+2}$. Note that under the above, $u_{m,r,h} = y_{m,r,h}$ and $u_{r,h} = \frac{1}{M}\sum_{m=1}^{M} u_{m,r,h}$. We first derive two intermediate lemmas, then proceed to the main proof.

**Lemma 8.** *Suppose that the local stepsize $\eta$ satisfies $\eta \leq \frac{1}{2L}$. Then, for all $h \in [H-1]$ and $r$, we have*

$$Hf(u_{r+1}) \leq \frac{1}{M} \sum_{m,h<H} \left[ \mathbb{E}\left[f(y_{m,r,h})\right] - \frac{\eta}{4} \sum_{h'=h}^{H-1} \mathbb{E}\left[\|\bar{g}_{m,r,h}\|^2\right] \right] + \frac{L\eta^2\sigma^2 H^2}{2M} + \frac{\eta^3\sigma^2 H^3}{2}.$$

*Proof.* By smoothness,

$$f(u_{r,h+1}) \leq f(u_{r,h}) + \langle \nabla f(u_{r,h}), u_{r,h+1} - u_{r,h} \rangle + \frac{L}{2}\|u_{r,h+1} - u_{r,h}\|^2$$

$$= f(u_{r,h}) - \eta \langle \nabla f(u_{r,h}), g_{r,h} \rangle + \frac{L\eta^2}{2}\|g_{r,h}\|^2.$$

Taking conditional expectation we have

$$\mathbb{E}_h\left[f(u_{r,h+1})\right] \leq f(u_{r,h}) - \frac{\eta}{M}\sum_{m=1}^{M}\langle \nabla f(u_{r,h}), \nabla f(u_{m,r,h})\rangle + \frac{L\eta^2\sigma^2}{2M} + \frac{L\eta^2}{2}\left\|\frac{1}{M}\sum_{m=1}^{M}\nabla f(u_{m,r,h})\right\|^2$$

$$\leq f(u_{r,h}) - \frac{\eta}{2M}\sum_{m=1}^{M}\left[\|\nabla f(u_{r,h})\|^2 + \|\nabla f(u_{m,r,h})\|^2 - \|\nabla f(u_{r,h}) - \nabla f(u_{m,r,h})\|^2\right] + \frac{L\eta^2\sigma^2}{2M}$$

$$+ \frac{L\eta^2}{2M}\sum_{m=1}^{M}\|\nabla f(u_{m,r,h})\|^2$$

$$= f(u_{r,h}) - \frac{\eta}{2}\|\nabla f(u_{r,h})\|^2 - \frac{\eta(1-L\eta)}{2M}\sum_{m=1}^{M}\|\nabla f(u_{m,r,h})\|^2 + \frac{\eta}{2}V_{r,h} + \frac{L\eta^2\sigma^2}{2M},$$

where $V_{r,h} = \frac{1}{M}\sum_{m=1}^{M}\|\nabla f(u_{r,h}) - \nabla f(u_{m,r,h})\|^2 \leq \frac{L^2}{M}\sum_{m=1}^{M}\|u_{r,h} - u_{m,r,h}\|^2$. Taking unconditional expectation, dropping the $\|\nabla f(u_{r,h})\|^2$ term and using Lemma 7 we have

$$\mathbb{E}\left[f(u_{r,h+1})\right] \leq \mathbb{E}\left[f(u_{r,h})\right] - \frac{\eta(1-L\eta)}{2M}\sum_{m=1}^{M}\mathbb{E}\left[\|\nabla f(u_{m,r,h})\|^2\right] + \frac{L\eta^2\sigma^2}{2M} + \eta^3 L^2\sigma^2 h.$$

Observe that in the current scheme, $\bar{g}_{m,r,h} = \nabla f(u_{m,r,h})$. Suppose that $1 - L\eta \geq \frac{1}{2}$, using this and telescoping yields

$$\mathbb{E}\left[f(u_{r+1})\right] \leq \mathbb{E}\left[f(u_{r,h})\right] - \frac{\eta}{4M}\sum_{m=1}^{M}\sum_{h'=h}^{H-1}\mathbb{E}\left[\|\bar{g}_{m,r,h'}\|^2\right] + \frac{L\eta^2\sigma^2(H-h)}{2M} + \eta^3 L^2\sigma^2\sum_{h'=h}^{H-1}h'.$$

Using Jensen's inequality on $u_{r,h} = \frac{1}{M}\sum_{m=1}^{M}u_{m,r,h} = \frac{1}{M}\sum_{m=1}^{M}y_{m,r,h}$ we obtain

$$\mathbb{E}\left[f(u_{r+1})\right] \leq \frac{1}{M}\sum_{m=1}^{M}\left[f(y_{m,r,h}) - \frac{\eta}{4}\sum_{h'=h}^{H-1}\mathbb{E}\left[\|\bar{g}_{m,r,h}\|^2\right]\right] + \frac{L\eta^2\sigma^2(H-h)}{2M} + L^2\eta^3\sigma^2\sum_{h'=h}^{H-1}h'.$$

Summing up both sides as $h$ varies from 0 to $H-1$ we get

$$Hf(u_{r+1}) \leq \frac{1}{M}\sum_{m,h<H}\left[\mathbb{E}\left[f(y_{m,r,h})\right] - \frac{\eta}{4}\sum_{h'=h}^{H-1}\mathbb{E}\left[\|\bar{g}_{m,r,h}\|^2\right]\right] + \frac{L\eta^2\sigma^2 H^2}{2M} + \frac{\eta^3 L^2\sigma^2 H^3}{2}.$$

$\square$

Define $G_r = \sum_{h=0}^{H-1}g_{r,h}$ and $\bar{G}_r = \sum_{h=0}^{H-1}\bar{g}_{r,h}$. The following lemma characterizes the evolution of the momentum sequence $z_1, z_2, \ldots$.

**Lemma 9** (Momentum sequence bound). *For any $r \geq 0$, the momentum sequence satisfies:*

$$\mathbb{E}_r\left[\|z_{r+1} - x_*\|^2\right] = \|z_r - x_*\|^2 + \gamma_r^2\eta^2\mathbb{E}_r\left[\|\bar{G}_r\|^2\right] + \frac{\gamma_r^2\eta^2\sigma^2}{M} - \gamma_r\eta\langle z_r - x_*, \mathbb{E}_r\left[\bar{G}_r\right]\rangle.$$

*Proof.* Expanding the square,

$$\|z_{r+1} - x_*\|^2 = \|z_r - x_*\|^2 + \gamma_r^2 \eta^2 \|G_r\|^2 - 2\gamma_r \eta \langle z_r - x_*, G_r \rangle.$$

Taking expectations and using Lemma 3,

$$\mathbb{E}_r\left[\|z_{r+1} - x_*\|^2\right] = \|z_r - x_*\|^2 + \gamma_r^2 \eta^2 \mathbb{E}_r\left[\|\bar{G}_r\|^2\right] + \frac{\gamma_r^2 \eta^2 \sigma^2 H}{M} - \gamma_r \eta \langle z_r - x_*, \mathbb{E}_r\left[\bar{G}_r\right]\rangle.$$

$\square$

*Proof of Theorem 3.* Define the potential function

$$\Phi_r = r(r+1)H(f(u_r) - f(x_*)) + \frac{2}{\gamma\eta} \| z_r - x_* \|^2 .$$

Using Lemma 8 and Lemma 9, we have

$$\mathbb{E}_r[\Phi_{r+1}] - \Phi_r$$
$$= (r+1)(r+2)H\left(\mathbb{E}_r[f(u_{r+1})] - f(x_*)\right) - r(r+1)H\left(f(u_r) - f(x_*)\right)$$
$$\quad + \frac{2}{\gamma\eta}\left[\mathbb{E}_r[\| z_{r+1} - x_* \|^2] - \| z_r - x_* \|^2\right]$$
$$\leq (r+1)(r+2)\left[\frac{1}{M}\sum_{m,h<H}\left[(\mathbb{E}_r[f(y_{m,r,h})] - f(x_*)) - \frac{\eta}{4}\sum_{h'=h}^{H-1}\mathbb{E}_r[\| \bar{g}_{m,r,h'} \|^2]\right] + \frac{L\eta^2\sigma^2 H^2}{2M} + \frac{\eta^3 L^2\sigma^2 H^3}{2}\right]$$
$$\quad - r(r+1)H\left(f(u_r) - f(x_*)\right)$$
$$\quad + \frac{\gamma\eta(r+1)^2}{2}\mathbb{E}_r[\| \bar{G}_r \|^2] + \frac{\gamma\eta\sigma^2(r+1)^2 H}{2M} - 2(r+1)\langle z_r - x_*, \mathbb{E}_r[\bar{G}_r]\rangle$$
$$= \underbrace{\frac{1}{M}\sum_{m,h<H}\left[2(r+1)(\mathbb{E}_r[f(y_{m,r,h})] - f(x_*)) + r(r+1)(\mathbb{E}_r[f(y_{m,r,h})] - f(u_r)) - 2(r+1)\langle z_r - x_*, \mathbb{E}_r[\bar{g}_{m,r,h}]\rangle\right]}_{=:A}$$
$$\quad \underbrace{-\frac{(r+1)(r+2)\eta}{4M}\sum_{m,h<H}\sum_{h'=h}^{H-1}\mathbb{E}_r[\| \bar{g}_{m,r,h'} \|^2] + \frac{\gamma\eta(r+1)^2}{2}\mathbb{E}_r[\| \bar{G}_r \|^2]}_{=:B}$$
$$\quad + \frac{(r+1)(r+2)L\eta^2\sigma^2 H^2}{2M} + \frac{(r+1)(r+2)\eta^3 L^2\sigma^2 H^3}{2} + \frac{\gamma\eta\sigma^2(r+1)^2 H}{2M}.$$

Now, we bound the terms above separately. First, we bound $A$. Fix any $m, h < H$. We have, using convexity of $f$,

$$2(r+1)(f(y_{m,r,h}) - f(x_*)) + r(r+1)(f(y_{m,r,h}) - f(u_r)) - 2(r+1)\langle z_r - x_*, \bar{g}_{m,r,h}\rangle$$
$$\leq 2(r+1)\langle y_{m,r,h} - x_*, \bar{g}_{m,r,h}\rangle + r(r+1)\langle y_{m,r,h} - u_r, \bar{g}_{m,r,h}\rangle - 2(r+1)\langle z_r - x_*, \bar{g}_{m,r,h}\rangle$$
$$= \langle (r+1)(r+2)y_{m,r,h} - r(r+1)u_r - 2(r+1)z_r, \bar{g}_{m,r,h}\rangle$$
$$= (r+1)(r+2)\langle y_{m,r,h} - x_r, \bar{g}_{m,r,h}\rangle$$
$$= -\eta(r+1)(r+2)\sum_{h'<h}\langle g_{m,r,h'}, \bar{g}_{m,r,h}\rangle.$$

Hence,

$$2(r+1)(\mathbb{E}_r[f(y_{m,r,h})] - f(x_*)) + r(r+1)(\mathbb{E}_r[f(y_{m,r,h})] - f(u_r)) - 2(r+1)\langle z_r - x_*, \mathbb{E}_r[\bar{g}_{m,r,h}]\rangle$$
$$\leq -\eta(r+1)(r+2)\sum_{h'<h}\mathbb{E}_r[\langle \bar{g}_{m,r,h'}, \bar{g}_{m,r,h}\rangle]$$

Since $A$ equals the sum of the above over all $m, h < H$ and dividing by $M$, we get:

$$A = \frac{-\eta(r+1)(r+2)}{M} \sum_{m, h < H} \sum_{h' < h} \mathbb{E}_r[\langle \bar{g}_{m,r,h'}, \bar{g}_{m,r,h} \rangle]$$

$$= \frac{-\eta(r+1)(r+2)}{2M} \sum_m \mathbb{E}_r \left[ \| \sum_{h < H} \bar{g}_{m,r,h} \|^2 - \sum_{h < H} \| \bar{g}_{m,r,h} \|^2 \right],$$

where in the last line we used the algebraic identity that for any sequence of vectors $v_0, \ldots, v_{H-1}$,

$$\sum_{h < H} \sum_{s < h} \langle v_s, v_h \rangle = \frac{1}{2} \left[ \| \sum_{h < H} v_h \|^2 - \sum_{h < H} \| v_h \|^2 \right].$$

Next, we have

$$B = -\frac{(r+1)(r+2)\eta}{4M} \sum_{m, h < H} \sum_{h'=h}^{H-1} \mathbb{E}_r[\| \bar{g}_{m,r,h'} \|^2] + \frac{\gamma\eta(r+1)^2}{2} \mathbb{E}_r[\| \bar{G}_r \|^2]$$

$$\leq -\frac{(r+1)(r+2)\eta}{4M} \sum_m \sum_{h < H} \mathbb{E}_r[\| \bar{g}_{m,r,h} \|^2] + \frac{\gamma\eta(r+1)^2}{2M} \sum_m \mathbb{E}_r \left[ \| \sum_{h < H} \bar{g}_{m,r,h} \|^2 \right].$$

Hence, we have

$$A + B \leq \frac{-\eta(r+1)(r+2)}{2M} \sum_m \mathbb{E}_r \left[ \| \sum_{h < H} \bar{g}_{m,r,h} \|^2 - \sum_{h < H} \| \bar{g}_{m,r,h} \|^2 \right]$$

$$- \frac{(r+1)(r+2)\eta}{4M} \sum_m \sum_{h < H} \mathbb{E}_r[\| \bar{g}_{m,r,h} \|^2] + \frac{\gamma\eta(r+1)^2}{2M} \sum_m \mathbb{E}_r \left[ \| \sum_{h < H} \bar{g}_{m,r,h} \|^2 \right]$$

$$= \frac{\eta(r+1)}{2M} \sum_m \mathbb{E}_r \left[ \| \sum_{h < H} \bar{g}_{m,r,h} \|^2 \right] [\gamma(r+1) - (r+2)] \qquad\qquad \leq 0$$

$$+ \frac{\eta(r+1)(r+2)}{4M} \sum_m \sum_{h < H} \mathbb{E}_r[\| \bar{g}_{m,r,h} \|^2]$$

since $\gamma \leq 1$ implies $\gamma(r+1) - (r+2) = (r+1)(\gamma-1) - 1 \leq -1 < 0$, and the second term has a positive coefficient with a negative sign.

So overall, we have

$$\mathbb{E}_r[\Phi_{r+1}] - \Phi_r \leq \frac{(r+1)(r+2)L\eta^2\sigma^2H^2}{2M} + \frac{(r+1)(r+2)\eta^3L^2\sigma^2H^3}{2} + \frac{\gamma\eta\sigma^2(r+1)^2H}{2M}$$

$$\leq \frac{R^2L\eta^2\sigma^2H^2}{2M} + \frac{R^2\eta^3L^2\sigma^2H^3}{2} + \frac{\gamma\eta\sigma^2R^2H}{2M}.$$

Summing up from $r = 0$ to $R - 1$, and taking expectations, we get

$$\mathbb{E}[\Phi_R] - \Phi_0 \leq \frac{R^3L\eta^2\sigma^2H^2}{2M} + \frac{R^3\eta^3L^2\sigma^2H^3}{2} + \frac{\gamma\eta\sigma^2R^3H}{2M}.$$

Thus,

$$R^2H(\mathbb{E}[f(u_R)] - f(x_*))$$

$$\leq \mathbb{E}[\Phi_R] \leq \frac{2 \| x_0 - x_* \|^2}{\gamma\eta} + \frac{R^3L\eta^2\sigma^2H^2}{2M} + \frac{R^3\eta^3L^2\sigma^2H^3}{2} + \frac{\gamma\eta\sigma^2R^3H}{2M},$$

which implies that

$$\mathbb{E}[f(u_R)] - f(x_*) \leq \frac{2 \| x_0 - x_* \|^2}{\gamma\eta R^2 H} + \frac{RL\eta^2\sigma^2H}{2M} + \frac{RL^2\eta^3\sigma^2H^2}{2} + \frac{\gamma\eta\sigma^2R}{2M}.$$

$\square$

The proof of Corollary 1 is straightforward by substitution and is omitted for brevity.

## B.5 Data-dependent guarantees

**Lemma 10.** *Let $f$ be a convex and $L$-smooth function. Suppose that we run SGD on $f$ on $M$ parallel nodes as follows*

$$y_{m,r,0} = x_r,$$
$$y_{m,r,h+1} = y_{m,r,h} - \eta g_{m,r,h},$$

*where $m = 1, 2, \ldots, M$, $h = 0, 1, \ldots, H - 1$, and $g_{1,r,h}, g_{2,r,h}, \ldots, g_{M,r,h}$ are i.i.d. stochastic gradient estimates such that $\mathbb{E}_{r,h}[g_{m,r,h}] = \nabla f(y_{m,r,h})$, where $\mathbb{E}_{r,h}[\cdot]$ denotes expectation conditional on all information up to and including round $r$ and local step $h$, and $\|g_{m,r,h} - \nabla f(y_{m,r,h})\| \leq \sigma$. Define further $y_{r,h} = \frac{1}{M} \sum_{m=1}^{M} y_{m,r,h}$. Let $V_{r,h} = \frac{1}{M} \sum_{m=1}^{M} \|y_{m,r,h} - y_{r,h}\|^2$. Then for all $\eta \leq \frac{1}{L}$ we have with probability at least $1 - \delta$ that for all $h = 0, 1, \ldots, H$*

$$V_{r,h} \leq 4104 \eta^2 \sigma^2 (h+1) \theta_{h-1,\delta}^2,$$

*where $\theta_{h,\delta} = \log \frac{60 \log 6h}{\delta}$.*

*Proof.* Define

$$\Lambda_{r,h+1} = \frac{1}{M^2} \sum_{m=1}^{M} \sum_{s=1}^{M} \|y_{m,r,h+1} - y_{s,r,h+1}\|^2. \tag{33}$$

We will bound $\Lambda_{r,h}$ first, and then use it to bound $V_{r,h}$ later. We have

$$y_{m,r,h+1} - y_{s,r,h+1} = y_{m,r,h} - \eta g_{m,r,h} - [y_{s,r,h} - \eta g_{s,r,h}]$$
$$= y_{m,r,h} - \eta \nabla f(y_{m,r,h}) - \eta [g_{m,r,h} - \nabla f(y_{m,r,h})] - [y_{s,r,h} - \eta \nabla f(y_{s,r,h}) - \eta [g_{s,r,h} - \nabla f(y_{s,r,h})]]$$
$$= [y_{m,r,h} - \eta \nabla f(y_{m,r,h}) - [y_{s,r,h} - \eta \nabla f(y_{s,r,h})]] - \eta [(g_{m,r,h} - g_{s,r,h}) - [\nabla f(y_{m,r,h}) - \nabla f(y_{s,r,h})]].$$

Therefore

$$\begin{aligned}\|y_{m,r,h+1} - y_{s,r,h+1}\|^2 =&\ \| T_\eta(y_{m,r,h}) - T_\eta(y_{s,r,h}) \|^2 \\ &+ \eta^2 \| (g_{m,r,h} - g_{s,r,h}) - (\nabla f(y_{m,r,h}) - \nabla f(y_{s,r,h})) \|^2 \\ &- 2\eta \langle T_\eta(y_{m,r,h}) - T_\eta(y_{s,r,h}), (g_{m,r,h} - g_{s,r,h}) - (\nabla f(y_{m,r,h}) - \nabla f(y_{s,r,h})) \rangle \end{aligned} \tag{34}$$

We define $\rho_{m,r,h}$ as the stochastic gradient noise on node $m$ at round $r$, step $h$: $\rho_{m,r,h} = g_{m,r,h} - \nabla f(y_{m,r,h})$. Then we can write Equation (34) as

$$\begin{aligned}\|y_{m,r,h+1} - y_{s,r,h+1}\|^2 =&\ \| T_\eta(y_{m,r,h}) - T_\eta(y_{s,r,h}) \|^2 + \eta^2 \| \rho_{m,r,h} - \rho_{s,r,h} \|^2 \\ &- 2\eta \langle T_\eta(y_{m,r,h}) - T_\eta(y_{s,r,h}), \rho_{m,r,h} - \rho_{s,r,h} \rangle. \end{aligned} \tag{35}$$

We now use the inequality $\|a + b\|^2 \leq 2\|a\|^2 + 2\|b\|^2$ to get

$$\begin{aligned}\|y_{m,r,h+1} - y_{s,r,h+1}\|^2 \leq&\ \| T_\eta(y_{m,r,h}) - T_\eta(y_{s,r,h}) \|^2 + 2\eta^2 \| \rho_{m,r,h} \|^2 + 2\eta^2 \|\rho_{s,r,h}\|^2 \\ &- 2\eta \langle T_\eta(y_{m,r,h}) - T_\eta(y_{s,r,h}), \rho_{m,r,h} - \rho_{s,r,h} \rangle. \end{aligned}$$

By Lemma 1, we have

$$\begin{aligned}\|y_{m,r,h+1} - y_{s,r,h+1}\|^2 \leq&\ \|y_{m,r,h} - y_{s,r,h}\|^2 + 2\eta^2 \|\rho_{m,r,h}\|^2 + 2\eta^2 \|\rho_{s,r,h}\|^2 \\ &- 2\eta \langle T_\eta(y_{m,r,h}) - T_\eta(y_{s,r,h}), \rho_{m,r,h} - \rho_{s,r,h} \rangle. \end{aligned}$$

Now, we consider the inner product term, observe

$$\begin{aligned}&\langle T_\eta(y_{m,r,h}) - T_\eta(y_{s,r,h}), \rho_{m,r,h} - \rho_{s,r,h} \rangle \\ &= \langle T_\eta(y_{m,r,h}) - T_\eta(y_{r,h}) + T_\eta(y_{r,h}) - T_\eta(y_{s,r,h}), \rho_{m,r,h} - \rho_{s,r,h} \rangle \\ &= \langle T_\eta(y_{m,r,h}) - T_\eta(y_{r,h}), \rho_{m,r,h} - \rho_{s,r,h} \rangle + \langle T_\eta(y_{r,h}) - T_\eta(y_{s,r,h}), \rho_{m,r,h} - \rho_{s,r,h} \rangle \\ &= \langle T_\eta(y_{m,r,h}) - T_\eta(y_{r,h}), \rho_{m,r,h} - \rho_{s,r,h} \rangle + \langle -(T_\eta(y_{s,r,h}) - T_\eta(y_{r,h})), -(\rho_{s,r,h} - \rho_{m,r,h}) \rangle \\ &= \langle T_\eta(y_{m,r,h}) - T_\eta(y_{r,h}), \rho_{m,r,h} - \rho_{s,r,h} \rangle + \langle T_\eta(y_{s,r,h}) - T_\eta(y_{r,h}), \rho_{s,r,h} - \rho_{m,r,h} \rangle. \end{aligned}$$

Averaging with respect to $s$ and $m$

$$\frac{1}{M^2} \sum_{m=1}^{M} \sum_{s=1}^{M} \langle T_\eta(y_{m,r,h}) - T_\eta(y_{r,h}) + T_\eta(y_{r,h}) - T_\eta(y_{s,r,h}), \rho_{m,r,h} - \rho_{s,r,h} \rangle$$

$$= \frac{1}{M^2} \sum_{m=1}^{M} \sum_{s=1}^{M} \langle T_\eta(y_{m,r,h}) - T_\eta(y_{r,h}), \rho_{m,r,h} - \rho_{s,r,h} \rangle$$

$$+ \frac{1}{M^2} \sum_{m=1}^{M} \sum_{s=1}^{M} \langle T_\eta(y_{s,r,h}) - T_\eta(y_{r,h}), \rho_{s,r,h} - \rho_{m,r,h} \rangle$$

$$= \frac{2}{M^2} \sum_{m=1}^{M} \sum_{s=1}^{M} \langle T_\eta(y_{m,r,h}) - T_\eta(y_{r,h}), \rho_{m,r,h} - \rho_{s,r,h} \rangle. \qquad (36)$$

Averaging Equation (35) with respect to $m$ and $s$ and using Equation (36) we get

$$\frac{1}{M^2} \sum_{m=1}^{M} \sum_{s=1}^{M} \|y_{m,r,h+1} - y_{s,r,h+1}\|^2 \le \frac{1}{M^2} \sum_{m=1}^{M} \sum_{s=1}^{M} \|y_{m,r,h} - y_{s,r,h}\|^2 + \frac{4\eta^2}{M} \sum_{m=1}^{M} \|\rho_{m,r,h}\|^2$$

$$- \frac{2\eta}{M^2} \sum_{m=1}^{M} \sum_{s=1}^{M} \langle T_\eta(y_{m,r,h}) - T_\eta(y_{r,h}), \rho_{m,r,h} - \rho_{s,r,h} \rangle.$$

Using $\Lambda_{r,h}$ as defined in Equation (33) we obtain the recursion

$$\Lambda_{r,h+1} \le \Lambda_{r,h} + \frac{4\eta^2}{M} \sum_{m=1}^{M} \|\rho_{m,r,h}\|^2 - \frac{2\eta}{M^2} \sum_{m=1}^{M} \sum_{s=1}^{M} \langle T_\eta(y_{m,r,h}) - T_\eta(y_{r,h}), \rho_{m,r,h} - \rho_{s,r,h} \rangle.$$

Now observe that $\|\rho_{m,r,h}\|^2 \le \sigma^2$ by assumption, therefore

$$\Lambda_{r,h+1} \le \Lambda_{r,h} + 4\eta^2\sigma^2 - \frac{2\eta}{M^2} \sum_{m=1}^{M} \sum_{s=1}^{M} \langle T_\eta(y_{m,r,h}) - T_\eta(y_{r,h}), \rho_{m,r,h} - \rho_{s,r,h} \rangle.$$

Recursing the above inequality we get

$$\Lambda_{r,h} \le \Lambda_{r,0} + 4\eta^2\sigma^2 h - \frac{2\eta}{M^2} \sum_{k=0}^{h-1} \sum_{m=1}^{M} \sum_{s=1}^{M} \langle T_\eta(y_{m,r,k}) - T_\eta(y_{r,k}), \rho_{m,r,k} - \rho_{s,r,k} \rangle$$

$$= 4\eta^2\sigma^2 h - \frac{2\eta}{M^2} \sum_{k=0}^{h-1} \sum_{m=1}^{M} \sum_{s=1}^{M} \langle T_\eta(y_{m,r,k}) - T_\eta(y_{r,k}), \rho_{m,r,k} - \rho_{s,r,k} \rangle, \qquad (37)$$

where we used the fact that since $y_{m,r,0} = y_{s,r,0} = x_r$ for all $m, s$ then $\Lambda_{r,0} = 0$. Define

$$\mu_{r,h} = \frac{1}{M} \sum_{m=1}^{M} \|y_{m,r,h} - y_{r,h}\|, \qquad\qquad \overline{\mu}_{r,h} = \max_{k \le h} \mu_{r,k}, \quad (38)$$

$$X_{r,h} = \frac{1}{\overline{\mu}_{r,h}} \frac{1}{M^2} \sum_{m=1}^{M} \sum_{s=1}^{M} \langle T_\eta(y_{m,r,h}) - T_\eta(y_{r,h}), \rho_{m,r,h} - \rho_{s,r,h} \rangle. \qquad (39)$$

Let $\mathbb{E}_{r,h}[\cdot]$ denote the expectation conditional on all information up to and including round $r$ and local step $h$. Then,

$$\mathbb{E}_{r,h}[X_{r,h}] = 0.$$

Furthermore, we have by the triangle inequality, then our assumption on the noise followed by Lemma 1 that almost surely

$$|\langle T_\eta(y_{m,r,h}) - T_\eta(y_{r,h}), \rho_{m,r,h} - \rho_{s,r,h} \rangle| \le \|T_\eta(y_{m,r,h}) - T_\eta(y_{r,h})\| \|\rho_{m,r,h} - \rho_{s,r,h}\|$$

$$\le \|T_\eta(y_{m,r,h}) - T_\eta(y_{r,h})\| (\|\rho_{m,r,h}\| + \|\rho_{s,r,h}\|)$$

$$\le 2\sigma \|T_\eta(y_{m,r,h}) - T_\eta(y_{r,h})\|$$

$$\le 2\sigma \|y_{m,r,h} - y_{r,h}\|. \qquad (40)$$

By the definition of $X_{r,h}$ (Equation (39)), the triangle inequality, Equation (40), and the definition of $\overline{\mu}_{r,h}$ (Equation (38)) we have almost surely

$$
\begin{aligned}
|X_{r,h}| &= \frac{1}{\overline{\mu}_{r,h}} \left| \frac{1}{M^2} \sum_{m=1}^{M} \sum_{s=1}^{M} \langle T_\eta(y_{m,r,h}) - T_\eta(y_{r,h}), \rho_{m,r,h} - \rho_{s,r,h} \rangle \right| \\
&\leq \frac{1}{\overline{\mu}_{r,h}} \frac{1}{M^2} \sum_{m=1}^{M} \sum_{s=1}^{M} |\langle T_\eta(y_{m,r,h}) - T_\eta(y_{r,h}), \rho_{m,r,h} - \rho_{s,r,h} \rangle| \\
&\leq \frac{2\sigma}{\overline{\mu}_{r,h}} \frac{1}{M^2} \sum_{m=1}^{M} \sum_{s=1}^{M} \|y_{m,r,h} - y_{r,h}\| \\
&= 2\sigma \frac{\frac{1}{M} \sum_{m=1}^{M} \|y_{m,r,h} - y_{r,h}\|}{\overline{\mu}_{r,h}} \\
&\leq 2\sigma.
\end{aligned}
$$

Then by Lemma 4 with $y_h = \overline{\mu}_{r,h}$ we have with probability at least $1 - \delta$

$$
\begin{aligned}
\left| \sum_{k=0}^{h-1} \overline{\mu}_{r,k} X_{r,k} \right| &\leq 8\overline{\mu}_{r,h-1} \sqrt{\theta_{h-1,\delta} \sum_{k=0}^{h-1} X_{r,k}^2 + 4\sigma^2 \theta_{h,\delta}^2} \\
&\leq 8\overline{\mu}_{r,h-1} \sqrt{\theta_{h-1,\delta} 4h\sigma^2 + 4\sigma^2 \theta_{h,\delta}^2} \\
&\leq 16\overline{\mu}_{r,h-1} \theta_{h-1,\delta} \sigma \sqrt{h+1}. \tag{41}
\end{aligned}
$$

Observe that

$$
\sum_{k=0}^{h-1} \overline{\mu}_{r,k} X_{r,k} = \frac{1}{M^2} \sum_{k=0}^{h-1} \sum_{m=1}^{M} \sum_{s=1}^{M} \langle T_\eta(y_{m,r,k}) - T_\eta(y_{r,k}), \rho_{m,r,k} - \rho_{s,r,k} \rangle.
$$

Using this and Equation (41) to upper bound the right hand side of Equation (37) we obtain

$$
\begin{aligned}
\Lambda_{r,h} &\leq 4\eta^2 \sigma^2 h + 32\eta \overline{\mu}_{r,h-1} \theta_{h-1,\delta} \sigma \sqrt{h+1} \\
&\leq 4\eta^2 \sigma^2 h + 2\alpha (32\eta \theta_{h-1,\delta} \sigma \sqrt{h+1})^2 + \frac{\overline{\mu}_{r,h-1}^2}{2\alpha} \\
&= \eta^2 \sigma^2 (h+1) \theta_{h-1,\delta}^2 (4 + 2048\alpha) + \frac{\overline{\mu}_{r,h-1}^2}{2\alpha}, \tag{42}
\end{aligned}
$$

where we used that $2ab \leq \alpha a^2 + \frac{1}{\alpha} b^2$ in the second step. Let $\overline{\Lambda}_{r,h} = \max_{k \leq h} \Lambda_{r,k}$. Observe that the right hand side of Equation (42) is increasing in $h$, therefore

$$
\overline{\Lambda}_{r,h} \leq \eta^2 \sigma^2 (h+1) \theta_{h-1,\delta}^2 (4 + 2048\alpha) + \frac{\overline{\mu}_{r,h-1}^2}{2\alpha}. \tag{43}
$$

Observe that by the triangle inequality followed by Lemma 2

$$
\begin{aligned}
\mu_{r,h} &= \frac{1}{M} \sum_{m=1}^{M} \|y_{m,r,h} - y_{r,h}\| \\
&\leq \frac{1}{M^2} \sum_{m=1}^{M} \sum_{s=1}^{M} \|y_{m,r,h} - y_{s,r,h}\| \\
&\leq \sqrt{\frac{1}{M^2} \sum_{m=1}^{M} \sum_{s=1}^{M} \|y_{m,r,h} - y_{s,r,h}\|^2} \\
&= \sqrt{\Lambda_{r,h}}.
\end{aligned}
$$

It follows that $\overline{\mu}_{r,h} \leq \sqrt{\overline{\Lambda}_{r,h}}$. Using this in Equation (43) we get

$$\overline{\Lambda}_{r,h} \leq \eta^2\sigma^2(h+1)\theta^2_{h-1,\delta}(4+2048\alpha) + \frac{\overline{\Lambda}_{r,h-1}}{2\alpha}$$

$$\leq \eta^2\sigma^2(h+1)\theta^2_{h-1,\delta}(4+2048\alpha) + \frac{\overline{\Lambda}_{r,h}}{2\alpha}.$$

Rearranging we get

$$\left(1 - \frac{1}{2\alpha}\right)\overline{\Lambda}_{r,h} \leq \eta^2\sigma^2(h+1)\theta^2_{h-1,\delta}(4+2048\alpha)$$

Put $\alpha = 1$, then

$$\overline{\Lambda}_{r,h} \leq 4104\eta^2\sigma^2(h+1)\theta^2_{h-1,\delta}. \tag{44}$$

Now that we have our bound on $\overline{\Lambda}_{r,h}$, we can use it to bound $V_{r,h}$ as follows

$$V_{r,h} = \frac{1}{M}\sum_{m=1}^{M}\|y_{m,r,h} - y_{r,h}\|^2. \tag{45}$$

Observe that by Jensen's inequality

$$\|y_{m,r,h} - y_{r,h}\|^2 = \left\|y_{m,r,h} - \frac{1}{M}\sum_{s=1}^{M}y_{s,r,h}\right\|^2$$

$$= \left\|\frac{1}{M}(y_{m,r,h} - y_{s,r,h})\right\|^2$$

$$\leq \frac{1}{M}\sum_{s=1}^{M}\|y_{m,r,h} - y_{s,r,h}\|^2. \tag{46}$$

Combining Equations (45) and (46) we have

$$V_{r,h} \leq \frac{1}{M^2}\sum_{m=1}^{M}\sum_{s=1}^{M}\|y_{m,r,h} - y_{s,r,h}\|^2 = \Lambda_{r,h}.$$

Combining this with Equation (44) yields the lemma's statement. $\qquad\square$

**Lemma 11.** *(Per-round regret). In Algorithm 1, the iterates in a single communication round satisfy*

$$\|x_{r+1} - x_*\|^2 \leq \|x_r - x_*\|^2 + \gamma^2\eta^2\sum_{h=0}^{H-1}\|g_{r,h}\|^2 + 2\gamma\eta|1-\gamma|\zeta_2\sum_{h=0}^{H-1}\|g_{r,h}\|$$

$$+ \frac{\gamma\zeta_3 H}{\alpha} + \frac{\alpha\gamma\eta^2}{2}\frac{1}{M}\sum_{m=1}^{M}\sum_{h=0}^{H-1}\|g_{m,r,h}\|^2 - \frac{2\gamma\eta}{M}\sum_{h=0}^{H-1}\sum_{m=1}^{M}\langle y_{m,r,h} - x_*, g_{m,r,h}\rangle,$$

*where $\alpha > 0$ is arbitrary and*

$$\zeta_2 = \max_h\|y_{r,h} - y_{r,0}\|, \qquad\qquad \zeta_3 = \max_h\frac{1}{M}\sum_{m=1}^{M}\|y_{m,r,h} - y_{r,h}\|^2.$$

*Proof.* Define the virtual sequences

$$g_{r,h} = \frac{1}{M}\sum_{m=1}^{M}g_{m,r,h}, \qquad x_{r,0} = x_r, \qquad x_{r,h+1} = x_{r,h} - \gamma\eta g_{r,h}.$$

We have

$$\|x_{r,h+1} - x_*\|^2 = \|x_{r,h} - x_*\|^2 + \gamma^2\eta^2\|g_{r,h}\|^2 - 2\gamma\eta\langle x_{r,h} - x_*, g_{r,h}\rangle \tag{47}$$

The inner product term can be decomposed as

$$-\langle x_{r,h} - x_*, g_{r,h}\rangle = -\langle x_{r,h} - y_{r,h}, g_{r,h}\rangle - \langle y_{r,h} - x_*, g_{r,h}\rangle. \tag{48}$$

Observe that $x_{r,h} = x_r - \gamma\eta \sum_{s=0}^{h-1} g_{r,s}$ and $y_{r,h} = x_r - \eta \sum_{s=0}^{h-1} g_{r,s}$. Therefore,

$$
\begin{aligned}
\|x_{r,h} - y_{r,h}\| &= \left\| (\gamma - 1)\eta \sum_{s=0}^{h-1} g_{r,s} \right\| \\
&= |\gamma - 1| \, \|y_{r,h} - y_{r,0}\| \\
&\le |\gamma - 1| \, \zeta_2,
\end{aligned}
$$

where $\zeta_2 = \max_h \|y_{r,h} - y_{r,0}\|$. Using this in Equation (48)

$$-\langle x_{r,h} - y_{r,h}, g_{r,h}\rangle \le \|x_{r,h} - y_{r,h}\| \, \|g_{r,h}\| \le |1 - \gamma| \, \zeta_2 \, \|g_{r,h}\|. \tag{49}$$

Plugging Equation (49) into Equation (48) we get

$$
\begin{aligned}
-\langle x_{r,h} - x_*, g_{r,h}\rangle &\le |1 - \gamma| \, \zeta_2 \, \|g_{r,h}\| - \langle y_{r,h} - x_*, g_{r,h}\rangle \\
&= |1 - \gamma| \, \zeta_2 \, \|g_{r,h}\| - \frac{1}{M} \sum_{m=1}^{M} \langle y_{r,h} - x_*, g_{m,r,h}\rangle \\
&= |1 - \gamma| \, \zeta_2 \, \|g_{r,h}\| - \frac{1}{M} \sum_{m=1}^{M} \langle y_{r,h} - y_{m,r,h}, g_{m,r,h}\rangle - \frac{1}{M} \sum_{m=1}^{M} \langle y_{m,r,h} - x_*, g_{m,r,h}\rangle. \tag{50}
\end{aligned}
$$

For the second term in Equation (50) we have

$$
\begin{aligned}
-\frac{1}{M} \sum_{m=1}^{M} \langle y_{r,h} - y_{m,r,h}, g_{m,r,h}\rangle &\le \frac{1}{M} \sum_{m=1}^{M} \|y_{r,h} - y_{m,r,h}\| \, \|g_{m,r,h}\| \\
&\le \frac{1}{M} \sum_{m=1}^{M} \left[ \frac{\|y_{r,h} - y_{m,r,h}\|^2}{2\alpha\eta} + \frac{\alpha\eta}{2} \|g_{m,r,h}\|^2 \right] \\
&\le \frac{\zeta_3}{2\alpha\eta} + \frac{\alpha\eta}{2} \frac{1}{M} \sum_{m=1}^{M} \|g_{m,r,h}\|^2. \tag{51}
\end{aligned}
$$

Plugging Equation (51) into Equation (50) we get

$$
\begin{aligned}
-\langle x_{r,h} - x_*, g_{r,h}\rangle &\le |1 - \gamma| \, \zeta_2 \, \|g_{r,h}\| + \frac{\zeta_3}{2\alpha\eta} + \frac{\alpha\eta}{2} \frac{1}{M} \sum_{m=1}^{M} \|g_{m,r,h}\|^2 \\
&\quad - \frac{1}{M} \sum_{m=1}^{M} \langle y_{m,r,h} - x_*, g_{m,r,h}\rangle. \tag{52}
\end{aligned}
$$

Plug Equation (52) back into Equation (47) to get

$$
\begin{aligned}
\|x_{r,h+1} - x_*\|^2 &\le \|x_{r,h} - x_*\|^2 + \gamma^2\eta^2 \|g_{r,h}\|^2 + 2\gamma\eta \, |1 - \gamma| \, \zeta_2 \, \|g_{r,h}\| \\
&\quad + \frac{\gamma\zeta_3}{\alpha} + \frac{\alpha\gamma\eta^2}{M} \sum_{m=1}^{M} \|g_{m,r,h}\|^2 - \frac{2\gamma\eta}{M} \sum_{m=1}^{M} \langle y_{m,r,h} - x_*, g_{m,r,h}\rangle.
\end{aligned}
$$

Recursing we get

$$
\begin{aligned}
\|x_{r+1} - x_*\|^2 &\le \|x_r - x_*\|^2 + \gamma^2\eta^2 \sum_{h=0}^{H-1} \|g_{r,h}\|^2 + 2\gamma\eta \, |1 - \gamma| \, \zeta_2 \sum_{h=0}^{H-1} \|g_{r,h}\| \\
&\quad + \frac{\gamma\zeta_3 H}{\alpha} + \frac{\alpha\gamma\eta^2}{2} \frac{1}{M} \sum_{m=1}^{M} \sum_{h=0}^{H-1} \|g_{m,r,h}\|^2 - \frac{2\gamma\eta}{M} \sum_{h=0}^{H-1} \sum_{m=1}^{M} \langle y_{m,r,h} - x_*, g_{m,r,h}\rangle.
\end{aligned}
$$

$\square$

*Proof of Theorem 4.* Starting with the per-round recursion lemma, we have

$$\|x_{r+1} - x_*\|^2 \le \|x_r - x_*\|^2 + \gamma^2\eta^2 \sum_{h=0}^{H-1} \|g_{r,h}\|^2 + 2\gamma\eta \, |1-\gamma| \, \zeta_2 \sum_{h=0}^{H-1} \|g_{r,h}\|$$

$$+ \frac{\gamma\zeta_3 H}{\alpha} + \frac{\alpha\gamma\eta^2}{2} \frac{1}{M} \sum_{m=1}^{M} \sum_{h=0}^{H-1} \|g_{m,r,h}\|^2 - \frac{2\gamma\eta}{M} \sum_{h=0}^{H-1} \sum_{m=1}^{M} \langle y_{m,r,h} - x_*, g_{m,r,h}\rangle.$$

Observe that

$$\|y_{r,h} - y_{r,0}\| = \eta \left\| \sum_{k=0}^{h-1} g_{r,k} \right\|$$

$$\le \eta \sum_{k=0}^{h-1} \|g_{r,k}\|$$

$$\le \eta \sum_{k=0}^{H-1} \|g_{r,k}\|. \tag{53}$$

Since this holds for any $h$, we have that $\zeta_2 \le \eta \sum_{k=0}^{H-1} \| g_{r,k} \|$, where $\zeta_2$ is defined in Lemma 11. Moreover, by Lemma 10 we have that with probability $1 - \delta$ and an application of the union bound that for all $r, h$

$$\frac{1}{M} \sum_{m=1}^{M} \| y_{m,r,h} - y_{r,h} \|^2 \le 4104\iota\eta^2\sigma^2 H, \tag{54}$$

where $\iota = 2 \cdot \log \frac{60 \log 6RH}{\delta}$ and we used that $H + 1 \le 2H$. Since this bound holds for all $h$, we have

$$\zeta_3 = \max_h \frac{1}{M} \sum_{m=1}^{M} \| y_{m,r,h} - y_{r,h} \|^2 \le 4104\iota\eta^2\sigma^2 H.$$

Therefore by Equation (53) and Lemma 10

$$\|x_{r+1} - x_*\|^2 \le \|x_r - x_*\|^2 + \gamma^2\eta^2 \sum_{h=0}^{H-1} \|g_{r,h}\|^2 + 2\gamma \, |1-\gamma| \, \eta^2 \left( \sum_{h=0}^{H-1} \|g_{r,h}\| \right)^2$$

$$+ \frac{4104\gamma\eta^2\sigma^2 H^2}{\alpha}\iota + \frac{\alpha\gamma\eta^2}{2} \frac{1}{M} \sum_{m=1}^{M} \sum_{h=0}^{H-1} \|g_{m,r,h}\|^2 - \frac{2\gamma\eta}{M} \sum_{h=0}^{H-1} \sum_{m=1}^{M} \langle y_{m,r,h} - x_*, g_{m,r,h}\rangle.$$

Let $\xi_{m,r,h} = g_{m,r,h} - \nabla f(y_{m,r,h})$. Then,

$$\|x_{r+1} - x_*\|^2 \le \|x_r - x_*\|^2 + \gamma^2\eta^2 \sum_{h=0}^{H-1} \|g_{r,h}\|^2 + 2\gamma \, |1-\gamma| \, \eta^2 \left( \sum_{h=0}^{H-1} \|g_{r,h}\| \right)^2 + \frac{4104\gamma\eta^2\sigma^2 H^2}{\alpha}\iota$$

$$+ \frac{\alpha\gamma\eta^2}{2} \frac{1}{M} \sum_{m=1}^{M} \sum_{h=0}^{H-1} \|g_{m,r,h}\|^2 - \frac{2\gamma\eta}{M} \sum_{h=0}^{H-1} \sum_{m=1}^{M} \langle y_{m,r,h} - x_*, \nabla f(y_{m,r,h})\rangle$$

$$- \frac{2\gamma\eta}{M} \sum_{h=0}^{H-1} \sum_{m=1}^{M} \langle y_{m,r,h} - x_*, \xi_{m,r,h}\rangle, \tag{55}$$

where $\xi_{m,r,h} = g_{m,r,h} - \nabla f(y_{m,r,h})$. Define

$$\nu_{r,h} = \frac{1}{M} \sum_{m=1}^{M} \|y_{m,r,h} - x_*\|, \qquad \overline{\nu}_{r,h} = \max_{p \le r, s \le h} \nu_{p,s}.$$

Let

$$X_{r,h} = \frac{1}{\bar{\nu}_{r,h}} \frac{1}{M} \sum_{m=1}^{M} \langle y_{m,r,h} - x_*, \xi_{m,r,h} \rangle$$

Let $\mathcal{F}_{r,h-1}$ denote the sigma algebra generated by all randomness up to and including step $r, h-1$. Note that

$$\mathbb{E}_{\mathcal{F}_{r,h-1}}[X_{r,h}] = \frac{1}{\bar{\nu}_{r,h}} \frac{1}{M} \sum_{m=1}^{M} \mathbb{E}_{\mathcal{F}_{r,h}}[\langle y_{m,r,h} - x_*, \xi_{m,r,h} \rangle]$$

$$= \frac{1}{\bar{\nu}_{r,h}} \frac{1}{M} \sum_{m=1}^{M} \langle y_{m,r,h} - x_*, \mathbb{E}_{\mathcal{F}_{r,h}}[\xi_{m,r,h}] \rangle$$

$$= 0,$$

where we used that $\nu_{r,h}$ and $y_{m,r,h}$ are both $\mathcal{F}_{r,h-1}$-measurable and that the noise has mean zero. The edge cases $X_{r,0}$ are handled similarly. Moreover, using the assumption that $\|\xi_{m,r,h}\| \leq \sigma$ almost surely and the definition of $\bar{\nu}_{r,h}$,

$$\|X_{r,h}\| = \left\| \frac{1}{\bar{\nu}_{r,h}} \frac{1}{M} \sum_{m=1}^{M} \langle y_{m,r,h} - x_*, \xi_{m,r,h} \rangle \right\|$$

$$\leq \frac{1}{M} \sum_{m=1}^{M} \frac{\|y_{m,r,h} - x_*\| \|\xi_{m,r,h}\|}{\bar{\nu}_{r,h}}$$

$$\leq \frac{1}{M} \sum_{m=1}^{M} (1 \cdot \sigma)$$

$$= \sigma.$$

Applying Lemma 4 on $X_{r,h}$ with $y_{r,h} = \bar{\nu}_{r,h}$, $C_{r,h} = \sigma$, $\hat{X}_{r,h} = 0$ we have

$$\left| \frac{1}{M} \sum_{r=0}^{R-1} \sum_{h=0}^{H-1} \sum_{m=1}^{M} \langle y_{m,r,h} - x_*, \xi_{m,r,h} \rangle \right| \leq 16 \bar{\nu}_{R,H} \iota \sigma \sqrt{RH}, \tag{56}$$

where $\iota$ is defined as before. Using Equation (56) in Equation (55)

$$\frac{2\gamma\eta}{M} \sum_{m,r,h} \langle y_{m,r,h} - x_*, \nabla f(y_{m,r,h}) \rangle \leq \|x_0 - x_*\|^2 - \|x_R - x_*\|^2 + \gamma^2 \eta^2 \sum_{r,h} \|g_{r,h}\|^2$$

$$+ 2\gamma |1 - \gamma| \eta^2 \sum_{r=0}^{R-1} \left( \sum_{h=0}^{H-1} \|g_{r,h}\| \right)^2 + R \cdot \frac{4104 \gamma \eta^2 \sigma^2 H^2}{\alpha} \iota \tag{57}$$

$$+ \frac{\alpha \gamma \eta^2}{2} \frac{1}{M} \sum_{m,r,h} \|g_{m,r,h}\|^2 + 2\gamma\eta \left[ 16 \bar{\nu}_{R,H} \iota \sigma \sqrt{RH} \right].$$

Let

$$\Omega = \gamma^2 \eta^2 \sum_{r,h} \|g_{r,h}\|^2 + 2\gamma |1 - \gamma| \eta^2 \sum_{r=0}^{R-1} \left( \sum_{h=0}^{H-1} \|g_{r,h}\| \right)^2 + R \cdot \frac{4104 \gamma \eta^2 \sigma^2 H^2}{\alpha} \iota$$

$$+ \frac{\alpha \gamma \eta^2}{2} \frac{1}{M} \sum_{m,r,h} \|g_{m,r,h}\|^2 \tag{58}$$

Then by convexity and Equation (57) we get

$$\|x_R - x_*\|^2 \leq \|x_0 - x_*\|^2 + \Omega + 2\gamma\eta \left[ 16 \bar{\nu}_{R,H} \iota \sigma \sqrt{RH} \right] - \frac{2\gamma\eta}{M} \sum_{m,r,h} \langle y_{m,r,h} - x_*, \nabla f(y_{m,r,h}) \rangle$$

$$\leq \|x_0 - x_*\|^2 + \Omega + 2\gamma\eta \left[ 16 \bar{\nu}_{R,H} \iota \sigma \sqrt{RH} \right], \tag{59}$$

where in the second line we used that $x_*$ is the minimizer of $f$ and therefore $\langle y_{m,r,h} - x_*, \nabla f(y_{m,r,h}) \rangle \geq 0$ by convexity. It is not difficult to see that this guarantee in fact applies not just on $\|x_R - x_*\|^2$ but on any $x_r$. Let $d_r = \|x_r - x_*\|$ and $\bar{d}_r = \max_{r' \leq r} d_{r'}$. Observe

$$
\begin{aligned}
\nu_{r,h} = \frac{1}{M} \sum_{m=1}^{M} \|y_{m,r,h} - x_*\| &\leq \frac{1}{M} \sum_{m=1}^{M} \left[ \|y_{m,r,h} - y_{m,r,0}\| + \|x_r - x_*\| \right] \\
&\leq \left[ \frac{\eta}{M} \sum_{m=1}^{M} \sum_{k=0}^{h-1} \|g_{m,r,k}\| \right] + \|x_r - x_*\| \\
&\leq \left[ \frac{\eta}{M} \sum_{m=1}^{M} \sum_{k=0}^{H-1} \|g_{m,r,k}\| \right] + \|x_r - x_*\| .
\end{aligned}
\tag{60}
$$

Using Equation (60) in Equation (59) we get

$$
\begin{aligned}
\bar{d}_R^2 &\leq d_0^2 + \Omega + 32\gamma\eta\iota\sigma\sqrt{RH}\,\bar{\nu}_{R,H} \\
&\leq d_0^2 + \Omega + 32\gamma\eta\iota\sigma\sqrt{RH} \left[ \frac{\eta}{M} \sum_{m,h} \|g_{m,r,h}\| \right] + 32\gamma\eta\iota\sigma\sqrt{RH}\,\bar{d}_R \\
&\leq d_0^2 + \Omega + 2\left(32\gamma\eta\iota\sigma\sqrt{RH}\right)^2 + \eta^2 \left( \frac{1}{M} \sum_{m,h} \|g_{m,r,h}\| \right)^2 + \frac{\bar{d}_R^2}{2}.
\end{aligned}
$$

Therefore

$$
\bar{d}_R^2 \leq 2d_0^2 + 2\Omega + 4096\gamma^2\eta^2\iota^2\sigma^2 RH + 2\eta^2 \left( \frac{1}{M} \max_r \sum_{m,h} \|g_{m,r,h}\| \right)^2 .
\tag{61}
$$

By the triangle inequality applied twice and the definition of $\bar{d}_R$,

$$
\begin{aligned}
\|y_{m,r,s} - x_*\| &\leq \|y_{m,r,0} - y_{m,r,s}\| + \|y_{m,r,0} - x_*\| \\
&= \eta \left\| \sum_{h=0}^{s-1} g_{m,r,h} \right\| + \|y_{m,r,0} - x_*\| \\
&\leq \eta \sum_{h=0}^{s-1} \|g_{m,r,h}\| + \|y_{m,r,0} - x_*\| \\
&\leq \eta \sum_{h=0}^{s-1} \|g_{m,r,h}\| + \bar{d}_R \\
&\leq \eta \sum_{h=0}^{H-1} \|g_{m,r,h}\| + \bar{d}_R.
\end{aligned}
$$

Therefore

$$
\frac{1}{M} \sum_{m=1}^{M} \|y_{m,r,s} - x_*\| \leq \eta \left( \frac{1}{M} \sum_{m=1}^{M} \sum_{h=0}^{H-1} \|g_{m,r,h}\| \right) + \bar{d}_R
$$

We now use the inequality $(a + b)^2 \leq 2a^2 + 2b^2$ to get

$$\nu_{r,s}^2 = \left( \frac{1}{M} \sum_{m=1}^{M} \|y_{m,r,s} - x_*\| \right)^2$$

$$\leq 2 \left( \eta \left( \frac{1}{M} \sum_{m=1}^{M} \sum_{h=0}^{H-1} \|g_{m,r,h}\| \right) \right)^2 + 2\bar{d}_R^2$$

$$= 2\eta^2 \left( \frac{1}{M} \sum_{m,h} \|g_{m,r,h}\| \right)^2 + 2\bar{d}_R^2.$$

Finally, using our bound on $\bar{d}_R^2$ given by equation (61)

$$\nu_{r,s}^2 \leq 4d_0^2 + 4\Omega + 8192\gamma^2\eta^2\iota^2\sigma^2 RH + 6\eta^2 \left( \frac{1}{M} \sum_{m,h} \|g_{m,r,h}\| \right)^2,$$

Therefore

$$\bar{\nu}_{R,H}^2 = \max_{r,s} \nu_{r,s}^2$$

$$\leq 4d_0^2 + 4\Omega + 8192\gamma^2\eta^2\iota^2\sigma^2 RH + 6\eta^2 \left( \frac{1}{M} \max_r \sum_{m,h} \|g_{m,r,h}\| \right)^2.$$

By Equations (57) and (58) and the last equation,

$$\frac{2\gamma\eta}{M} \sum_{m,r,h} \langle y_{m,r,h} - x_*, \nabla f(y_{m,r,h}) \rangle \leq \|x_0 - x_*\|^2 - \|x_R - x_*\|^2 + \Omega + 2\gamma\eta \left[ 16\bar{\nu}_{R,H}\iota\sigma\sqrt{RH} \right]$$

$$\leq d_0^2 - d_R^2 + \Omega + \frac{(32\gamma\eta\iota\sigma\sqrt{RH})^2}{2} + 4 \left[ d_0^2 + \Omega + 2048\gamma^2\eta^2\iota^2\sigma^2 RH \right] + 6\eta^2 R \left( \frac{1}{M} \max_r \sum_{m,h} \|g_{m,r,h}\| \right)^2$$

$$= d_0^2 - d_R^2 + \Omega + \frac{(32\gamma\eta\iota\sigma\sqrt{RH})^2}{2} + 4 \left[ d_0^2 + \Omega + 2048\gamma^2\eta^2\iota^2\sigma^2 RH \right] + 6\eta^2 R \left( \frac{1}{M} \max_r \sum_{m,h} \|g_{m,r,h}\| \right)^2$$

$$\leq d_0^2 - d_R^2 + 6\Omega + 8704\gamma^2\eta^2\iota^2\sigma^2 RH + 4d_0^2 + 6\eta^2 R \left( \frac{1}{M} \max_r \sum_{m,h} \|g_{m,r,h}\| \right)^2. \qquad (62)$$

Dropping the $-d_R^2$ term, we get

$$\frac{2\gamma\eta}{M} \sum_{m,r,h} \langle y_{m,r,h} - x_*, \nabla f(y_{m,r,h}) \rangle \leq 5d_0^2 + 6\Omega + 8704\gamma^2\eta^2\iota^2\sigma^2 RH + 6\eta^2 R \left( \frac{1}{M} \max_r \sum_{m,h} \|g_{m,r,h}\| \right)^2$$

$$\leq 5d_0^2 + 6\gamma^2\eta^2 \sum_{r,h} \|g_{r,h}\|^2 + 12\gamma |1 - \gamma| \eta^2 \sum_{r=0}^{R-1} \left( \sum_{h=0}^{H-1} \|g_{r,h}\| \right)^2 + RH \frac{24624\gamma\eta^2\sigma^2 H\iota}{\alpha}$$

$$+ \frac{3\alpha\gamma\eta^2}{M} \sum_{m,r,h} \|g_{m,r,h}\|^2 + 8704\gamma^2\eta^2\iota^2\sigma^2 RH + 6\eta^2 R \left( \frac{1}{M} \max_r \sum_{m,h} \|g_{m,r,h}\| \right)^2.$$

Dividing both sides by $2\gamma\eta RH$ gives

$$\frac{1}{MRH}\sum_{m,r,h}\langle y_{m,r,h}-x_*,\nabla f(y_{m,r,h})\rangle \leq \frac{5d_0^2}{2\gamma\eta RH}+\frac{3\gamma\eta}{RH}\sum_{r,h}\|g_{r,h}\|^2$$

$$+\frac{6\,|1-\gamma|\,\eta}{RH}\sum_{r=0}^{R-1}\left(\sum_{h=0}^{H-1}\|g_{r,h}\|\right)^2+\frac{24624\eta\sigma^2 H\iota}{\alpha} \tag{63}$$

$$+\frac{3\alpha\eta}{MRH}\sum_{m,r,h}\|g_{m,r,h}\|^2+8704\gamma\eta\iota^2\sigma^2+\frac{6\eta}{\gamma H}\left(\frac{1}{M}\max_r\sum_{m,h}\|g_{m,r,h}\|\right)^2.$$

Observe that by optimizing over $\alpha$ we have

$$\frac{24624\eta\sigma^2 H\iota}{\alpha}+\frac{3\alpha\eta}{MRH}\sum_{m,r,h}\|g_{m,r,h}\|^2 \leq 2\sqrt{(24624\eta\sigma^2 H\iota)\left(\frac{3\eta}{MRH}\sum_{m,r,h}\|g_{m,r,h}\|^2\right)}$$

$$\leq 544\eta\sigma\iota\sqrt{\frac{1}{MR}\sum_{m,r,h}\|g_{m,r,h}\|^2}.$$

Using this in Equation (63) followed by convexity completes the proof. $\qquad\square$

