# OpenReview forum: "Understanding Outer Optimizers in Local SGD: Learning Rates, Momentum, and Acceleration"
_NeurIPS.cc/2025/Conference — NeurIPS 2025 poster_

### Official Review · Reviewer_GcXM · 2025-06-14

**Clarity:** 4
**Significance:** 3
**Originality:** 3
**Rating:** 5
**Confidence:** 3

**Summary:**

This work studies the impact of outer learning rate in Local SGD through a theoretical lens of convergence bounds. The theorems show the inverse relation between optimization error and gradient variance with different regimes of outer learning rates. The analysis is also extended to the momentum-based outer updates.

**Questions:**

a. What changes would be needed to make for a non-convex analysis on outer learning rates?

b. Are there empirical results on end perplexity for a grid of local and global learning rates? I wanted to see how the relation between the local and the global learning rates (of theorem 3.3) translate to convex and non-convex objectives in practice.

c. For clarification, does theorem 3.3 imply that larger $\eta$ (local learning rates) might result in an accumulated gradient with large variance, which needs to be counterbalanced by a smaller $\gamma$? Would we achieve the same effect with a larger $H$ (local steps)?

d. A follow up to the above question would be, under what setting of local and global learning rates we can minimize the FLOPs?

e. For Figure 9 in the Appendix, why are we seeing high gradient similarity near the end of the training for the left plots and why are we seeing low gradient similarity for the right plots? Furthermore, line 283 states "... noise-dominated regime, which we may expect to observe towards the end of the training process." Why is that so?

f. For Figure 1b, why does the loss start high for low variance setting of $\sigma = 0.001$?

g. For Table 1, SF-SGD case, why are outer learning rates with magnitudes >2 detrimental?

**Ethical Concerns:**

["NO or VERY MINOR ethics concerns only"]

**Final Justification:**

The authors have addressed my questions about non-convex analysis satisfactorily and clarified the empirical results, hence I would like to keep my score to "Accept".

**Limitations:**

Yes

**Quality:**

4

**Strengths And Weaknesses:**

**Strengths**

a. The work is well-motivated and the manuscript is well-written. The objective of the paper and the theoretical treatment were easy to follow.

b. The authors have done a good job of building the related work section, it would be extremely helpful for the beginners in this area.

c. The theorems are followed by insightful discussion which captures the key points well.

___
**Weaknesses**

a. The theoretical analysis is limited to the convex objective functions.

b. There's a lack of clarity in the empirical results in terms of the relation between local and global learning rates. E.g., For Section 4.1, is the local learning rate $\eta$ supposed to be in the "higher" or "lower" ranges according to the upper bounds set through Theorem 3.3. Some more questions are detailed below.

---

> ### Author Rebuttal · Authors · 2025-07-30
>
> Thank you so much for your review and positive evaluation of our work.
>
> 1. (Non-convex analysis). Our theorem mainly relies on the two lemmas Lemma B.6 and Lemma B.7; Lemma B.6 holds regardless of convexity, but Lemma B.7. does not. We would need to relax Lemma B.7. to handle non-convex objectives, but this lemma in turn relies on the fact that gradient descent is contractive (Lemma B.1.), which is not true for arbitrary non-convex functions. We believe it is possible to use similar tools as in [10, Theorem 4.1], but this requires adding the bounded gradients assumption and using a different approach for the theorem derivation that starts with the smoothness inequality. Alternatively, we can use the approach of [12] of bounding this sequence by a sum of gradients at the cost of an extra $H$ factor in the Local SGD variance term.
>
> 2. (LR grids) The local learning rate is always fixed in our experiments is fixed to $\frac{\eta_{\mathrm{baseline}}}{\sqrt{M}}$ where $\eta_{\mathrm{baseline}}$ is the local learning rate that was tuned on the minibatch SGD baseline for each model scale and $M$ is the number of nodes (see Table 1 in the main paper). We only varied the outer learning rate otherwise, since what is important for our analysis is the ratio of outer to local learning rate. We have included a table with all the experiments we’ve conducted varying the outer learning rate in our response to Reviewer advP.
>
> 3. (Implications of Thm. 3.3) Theorem 3.3 shows there are two sources of noise for Local SGD: the second term in equation (3) scales with $\eta$ and $\max(0, \gamma-1)$ but not $H$, and the third term scales with $\eta^2$ and also $H$. In addition, we also have to satisfy the constraint $\eta L (1 + (\gamma-1)_+ H) \leq 1$. For tuning the outer learning rate, if either $\eta$ or $H$ are larger, we are forced to use smaller $\gamma$ in the same way. However, the convergence of Local SGD will suffer more under larger $\eta$ than larger $H$, all else being equal, because larger $\eta$ will increase both noise terms while larger $H$ only increases the latter.
>
> 4. (Minimizing FLOPs) To minimize iteration complexity given a fixed number of clients $M$ and local steps $H$, we ought to choose the outer learning rate to make up for a small inner learning rate and to vary with the number of local steps. Depending on the magnitude of the gradient variance, choosing an outer learning rate larger than 1 can also be useful. We found that outer learning rates can be tuned at a smaller scale and transfer as-is to larger models. For example, with the 150M model at H=50, M=4, it achieves a perplexity of 17.75, while Nesterov with lr=0.7 reaches 17.25 and SF-SGD with lr=2.0 achieves 16.88. These represent improvements of 2.8% and 4.9% respectively compared to the baseline with no outer learning rate. The performance gains transfer and become even more pronounced at larger model scales: for the 1B model, vanilla SGD achieves 13.67 perplexity while Nesterov (lr=0.7) and SF-SGD (lr=2.0) reach 12.51 and 12.40 respectively, representing improvements of 8.5% and 9.3%. Of course, in practice, we control the number of clients $M$ and also have to consider FLOP efficiency from a systems perspective, e.g. normally increasing $M$ comes at the cost of decreasing the local batch size or alternatively increasing $\sigma^2$. This complicates the trade-off; Figure 4 shows the result of varying the number of clients/replicas in practice on FLOP efficiency. A study of FLOP efficiency would have to take into account the specific hardware we have available, and what the minimum batch size needed to saturate the GPU computation is.
>
> 5. “For Figure 9 in the Appendix, why are we seeing high gradient similarity near the end of the training for the left plots and why are we seeing low gradient similarity for the right plots?” For all of them except SF-SGD in Figure 9 (c), we do observe low similarity towards the end of training (lower than 0.1). For SF-SGD in Figure 9 (c), we do observe a spike in similarity towards the end but are not sure why this is the case.
>
> 3. “Furthermore, line 283 states "... noise-dominated regime, which we may expect to observe towards the end of the training process." Why is that so?” This is an empirical observation made in prior work [11], and the intuition is that towards the end of training the gradient norms are smaller so the relative magnitude of the noise is higher compared to the gradient norms.
>
> 4. “f. For Figure 1b, why does the loss start high for low variance setting of $\sigma = 0.001$?” All of the losses are reported after one communication round– the figure should start at r=1 not at r=0. They would all start at the same value if we started at r=0. Since we cannot upload PDFs or figures, here are the trajectories in tabular form:
>
>
> ### Figure 1 (b) tabular
> | σ     | Round 1      | Round 2      | Round 3      | Round 4      | Round 5      |
> |-------|--------------|--------------|--------------|--------------|--------------|
> | 0.001 | 1.421e+07    | 3.965e-02    | 3.050e-03    | 7.399e-04    | 5.203e-04    |
> | 0.01  | 1.421e+07    | 6.859e-02    | 6.751e-03    | 1.370e-03    | 7.368e-04    |
> | 0.1   | 1.421e+07    | 1.919e-01    | 2.858e-02    | 1.052e-02    | 6.916e-03    |
> | 0.5   | 1.421e+07    | 1.051e+00    | 2.031e-01    | 1.059e-01    | 7.660e-02    |
> | 1.0   | 1.421e+07    | 1.220e+00    | 3.663e-01    | 2.674e-01    | 2.948e-01    |
> | 10.0  | 1.421e+07    | 2.333e+01    | 8.727e+00    | 6.208e+00    | 5.305e+00    |
>
> 5. “g. For Table 1, SF-SGD case, why are outer learning rates with magnitudes >2 detrimental?” They weren’t really detrimental, they just did not help very much. Here are the results of our grid search on a 150M parameter model with $H=50$ and $M=4$.  Very small outer learning rates were a lot more detrimental, as they did not allow for enough progress.
>
> ## Additional Learning Rate Sweeps (150M, H=50, M=4)
>
> | Algorithm | Learning Rate | Perplexity |
> |-----------|---------------|------------|
> | SF-SGD | 0.1 | 30 |
> | SF-SGD | 0.5 | 22.89 |
> | SF-SGD | 1.0 | 19.42 |
> | SF-SGD | 1.5 | 18.32 |
> | SF-SGD | 2.0 | 17.98 |
> | SF-SGD | 3.0 | 17.96 |
> | SF-SGD | 4.0 | 18.09 |
> | SF-SGD | 5.0 | 17.51 |
>
>
> [10] Glasgow, M. R., Yuan, H., & Ma, T. (2022, May). Sharp bounds for federated averaging (local sgd) and continuous perspective. In International Conference on Artificial Intelligence and Statistics (pp. 9050-9090). PMLR.
> [11] Faghri, F., Duvenaud, D., Fleet, D. J., & Ba, J. (2020). A study of gradient variance in deep learning. arXiv preprint arXiv:2007.04532.

---

> > ### Comment · Reviewer_GcXM · 2025-08-05
> >
> > Apologies for having missed replying to the authors.
> > The authors have addressed my questions about non-convex analysis satisfactorily and clarified the empirical results. I would like to keep my score to "Accept".

---

### Official Review · Reviewer_advP · 2025-06-25

**Clarity:** 4
**Significance:** 3
**Originality:** 3
**Rating:** 5
**Confidence:** 4

**Summary:**

The paper seeks to better understand the role of the outer optimizer and its hyperparameters in the i.i.d Local SGD setting. Specifically, the main results of the paper are (1) a convergence theorem for generalized Local SGD with arbitrary inner and outer learning rates with SGD being the inner and outer optimizer, (2) an extension of the first theorem but using SGD with momentum as the outer optimizer, and (3) a high probability guarantee for the iterates of generalized local SGD. The authors provide analysis of each result, illustrate how their high-probability theory is borne out in practice, and position their results within the existing literature. Finally, the authors provide empirical results on a toy convex problem that is illustrative of their theoretical analysis and provide empirical results on language model pre-training of decoder-only transformers following the DiLoCo setting.

**Questions:**

- Is the use of a schedule-free outer optimizer instead of SGD with a schedule is due to the theory not allowing for scheduled outer learning rates? If so, mentioning this could be helpful for the reader.
- How is perplexity calculated in Figure 2? (e.g., what data, what batch size, etc)
- Why does Theorem 3.5 not use an EMA for the momentum?
- For LLM experiments, were the hyperparameters swept at each scale or were they swept only at a smaller scale?

**Ethical Concerns:**

["NO or VERY MINOR ethics concerns only"]

**Final Justification:**

I have increased my score. I recommend accepting the paper.

**Issues resolved:**
- Missing hyperparameters and final loss values for each.
- Misunderstanding of the momentum used in the proofs.
- Clarification of scheduled learning rates for the outer optimizer.
- Clarification of the use of a schedule-free outer-optimizer.

**Limitations:**

Yes

**Quality:**

3

**Strengths And Weaknesses:**

Strengths:
- The paper is well written, the work is effectively placed within the context of the existing literature, and the connection to and demarcation between new and existing results is clearly stated.
- To the extend of my limited understanding of convergence analysis proofs (I am not intimately familiar with the proofs from existing work, but I did read through the math in the main text), the authors theorems seem sound and to the best of my knowledge, they support what is mentioned in the implications sections/contributions.
- The empirical evaluations correspond with the theory, and the LLM experiments are conducted at a reasonably large scale.

Weaknesses:
- **Reproducibility** Having a single table with all hyperparameters swept would be beneficial for reproducibility. Currently, the ranges presented in Table 1 leave some guesswork to the reader, and I’m not sure the actual values used can be found from an attentive read of the appendix. Moreover, it would be beneficial to provide the final performance of all different hyperparameters swept in the appendix. This will help further improve the reproducibility of the paper and will be of use to practitioners reading the paper and wondering about the performance of different outer learning rates in the appendix. I will raise my score if the above concerns are adequately addressed (e.g., provide in line the tables of values you will report in the appendix and the final performance of HPs swept).

- **Impact** The practical guidance that can be gleaned from the current theory seems limited. I think the theoretical results are interesting and will certainly help improve our current understanding of the outer learning rate’s role in Local SGD and could potentially lead to more practical improvements down the line (why I recommend accepting). However, from my understanding, the current theory does not suggest practical improvements to the Local SGD algorithm beyond: 1) tuning the outer learning rate is important and 2) noisy outer gradients require a lower outer learning rate. Could the authors comment on the above?

- **presentation** Most of the captions could be more self-contained.

---

> ### Author Rebuttal · Authors · 2025-07-28
>
> Thank you so much for your review and positive evaluation of our work.
>
> 1. (Reproducibility) We did not report all of the hyperparameter sweep results in line with reporting in prior work which reported the best hyperparameters only. We are happy to report all the experimental results we have done. Here are the results of the hyperparameter sweeps we will include in the appendix:
>
> | H | M | Algorithm | Learning Rate | Perplexity | Model Size |
> |---|---|-----------|---------------|------------|------------|
> | - | 1 | Data-Parallel | - | 18.07 | 150M |
> | - | 1 | Data-Parallel | 4x BS | 16.89 | 150M |
> | - | 1 | Data-Parallel | - | 15.28 | 400M |
> | - | 1 | Data-Parallel | 4x BS | 13.21 | 400M |
> | - | 1 | Data-Parallel | - | 13.38 | 1B |
> | - | 1 | Data-Parallel | 4x BS | 11.34 | 1B |
> | 50 | 4 | SGD | 1.0 | 17.75 | 150M |
> | 50 | 4 | Nesterov | 0.7 | 17.25 | 150M |
> | 50 | 4 | Nesterov | 1.0 | 16.38 | 150M |
> | 50 | 4 | SF-SGD | 2.0 (b=0.2) | 16.88 | 150M |
> | 50 | 4 | SGD | 1.0 | 14.90 | 400M |
> | 50 | 4 | Nesterov | 0.7 | 13.71 | 400M |
> | 50 | 4 | Nesterov | 1.0 | 30 | 400M |
> | 50 | 4 | SF-SGD | 2.0 (b=0.2) | 13.95 | 400M |
> | 50 | 4 | SGD | 1.0 | 13.67 | 1B |
> | 50 | 4 | Nesterov | 0.7 | 12.51 | 1B |
> | 50 | 4 | SF-SGD | 2.0 (b=0.2) | 12.40 | 1B |
> | 150 | 4 | SGD | 1.0 | 17.58 | 150M |
> | 150 | 4 | Nesterov | 0.7 | 17.90 | 150M |
> | 150 | 4 | Nesterov | 1.0 | 16.79 | 150M |
> | 150 | 4 | SF-SGD | 2.0 (b=0.2) | 16.96 | 150M |
> | 250 | 4 | SGD | 1.0 | 18.20 | 150M |
> | 250 | 4 | Nesterov | 0.7 | 18.09 | 150M |
> | 250 | 4 | Nesterov | 1.0 | 17.12 | 150M |
> | 250 | 4 | SF-SGD | 2.0 (b=0.2) | 16.97 | 150M |
> | 500 | 4 | SGD | 1.0 | 18.44 | 150M |
> | 500 | 4 | Nesterov | 0.7 | 17.95 | 150M |
> | 500 | 4 | Nesterov | 1.0 | 18.15 | 150M |
> | 500 | 4 | SF-SGD | 2.0 (b=0.2) | 17.18 | 150M |
> | 1000 | 4 | SGD | 1.0 | 18.18 | 150M |
> | 1000 | 4 | Nesterov | 0.7 | 18.16 | 150M |
> | 1000 | 4 | Nesterov | 1.0 | 18.75 | 150M |
> | 1000 | 4 | SF-SGD | 2.0 (b=0.2) | 17.29 | 150M |
> | 2000 | 4 | SGD | 1.0 | 18.11 | 150M |
> | 2000 | 4 | Nesterov | 0.7 | 18.40 | 150M |
> | 2000 | 4 | Nesterov | 1.0 | 18.36 | 150M |
> | 2000 | 4 | SF-SGD | 2.0 (b=0.2) | 17.59 | 150M |
> | 50 | 2 | SGD | 1.0 | 18.64 | 150M |
> | 50 | 2 | Nesterov | 1.0 | 16.81 | 150M |
> | 50 | 2 | SF-SGD | 2.0 (b=0.2) | 17.13 | 150M |
> | 50 | 8 | SGD | 1.0 | 18.38 | 150M |
> | 50 | 8 | Nesterov | 1.0 | 16.27 | 150M |
> | 50 | 8 | SF-SGD | 2.0 (b=0.2) | 16.92 | 150M |
> | 50 | 16 | SGD | 1.0 | 19.86 | 150M |
> | 50 | 16 | Nesterov | 1.0 | 16.25 | 150M |
> | 50 | 16 | SF-SGD | 2.0 (b=0.2) | 16.75 | 150M |
>
> ## Additional Learning Rate Sweeps (150M, H=50, M=4)
>
> | Algorithm | Learning Rate | Perplexity |
> |-----------|---------------|------------|
> | SF-SGD | 0.1 | 30 |
> | SF-SGD | 0.5 | 22.89 |
> | SF-SGD | 1.0 | 19.42 |
> | SF-SGD | 1.5 | 18.32 |
> | SF-SGD | 2.0 | 17.98 |
> | SF-SGD | 3.0 | 17.96 |
> | SF-SGD | 4.0 | 18.09 |
> | SF-SGD | 5.0 | 17.51 |
> | Nesterov | 0.3 (cosine) | 17.16 |
> | Nesterov | 0.5 (cosine) | 17.06 |
> | Nesterov | 0.7 (cosine) | 16.93 |
> | Nesterov | 0.9 (cosine) | 17.19 |
> | Nesterov | 1.1 (cosine) | 17.56 |
> | SGD | 0.3 (fixed) | 21.04 |
> | SGD | 0.3 (cosine) | 17.68 |
> | SGD | 0.5 (cosine) | 16.63 |
> | SGD | 0.7 (cosine) | 18.84 |
> | SGD | 1.0 (cosine) | 19.21 |
>
> ## SF-SGD Beta (b) Parameter Sweep (150M, H=50, M=4, lr=2.0)
>
> | b value | Perplexity |
> |---------|------------|
> | 0.0 | 30 |
> | 0.05 | 16.88 |
> | 0.1 | 16.78 |
> | 0.2 | 16.89 |
> | 0.4 | 17.15 |
> | 0.5 | 17.35 |
> | 0.7 | 17.93 |
> | 0.9 | 19.07 |
> | 0.95 | 19.65 |
> | 0.99 | 20.51 |
>
> 2. (Impact) Yes, it is true that our theory as it is provides limited practical prescriptions. We are primarily concerned with understanding the role of the outer learning rate and how it should be tuned, but have not directly provided a theory-based schedule for it; this task we leave to future work. One crucial takeaway from our theory, which is not common in practice, is that it sometimes pays off to try outer learning rates greater than 1.0. To the best of our knowledge, the current literature (e.g. [7, 8]) did not sweep for such learning rates, going up to 1.0 and no higher. Our work shows that for Schedule-Free SGD as the outer optimizer, a learning rate of 2.0 performed best. We will highlight this recommendation further.
>
> 3. (Captions) We will expand the captions to be more self-contained in the revised manuscript. Thank you for pointing this out.
>
> 4. "Is the use of a schedule-free outer optimizer instead of SGD with a schedule is due to the theory not allowing for scheduled outer learning rates? If so, mentioning this could be helpful for the reader." The main reason we used it was to remove the variable of (outer) learning rate scheduling. The theory can be extended to scheduled outer learning rates, but in its current form  it does not allow for them. We will mention this.
>
> 5. "How is perplexity calculated in Figure 2? (e.g., what data, what batch size, etc)" Perplexity is calculated on the C4 validation set, with sequence length 1024, and batch size 512 for all model scales. The tokenizer is SentencePiece.
>
> 6. "Why does Theorem 3.5 not use an EMA for the momentum?" Our way of writing the momentum update is equivalent to using EMA for momentum. [9, Lemma 7.2] shows it is equivalent to the update $m_{t} = \mu m_{t-1} + g_t$ and $x_{t+1} = x_t - \eta m_t$. By unrolling, this corresponds to $x_{t+1} = x_t - \eta \sum_{k=0}^t \mu^k g_{t-k}$. For classical EMA written as $m_t = (1-\alpha) m_{t-1} + \alpha g_t$ and $x_{t+1} = x_t - \gamma m_t$, unrolling we have $x_{t+1} = x_t - \gamma \alpha \sum_{k=0}^t (1-\alpha)^k g_{t-k}$. Thus putting $\mu = 1-\alpha$ and $\gamma (1-\mu) = \eta$ we can see that they are equivalent up to a rescaling of the learning rate.
>
> 7. "For LLM experiments, were the hyperparameters swept at each scale or were they swept only at a smaller scale?" We swept them at the 150M scale, with a small number of additional trials at higher scale.
>
> Thank you so much for your detailed comments.
>
> [7] Douillard, Arthur, et al. "Diloco: Distributed low-communication training of language models." arXiv preprint arXiv:2311.08105 (2023).
> [8] Liu, B., Chhaparia, R., Douillard, A., Kale, S., Rusu, A. A., Shen, J., ... & Ranzato, M. A. (2024). Asynchronous local-sgd training for language modeling. arXiv preprint arXiv:2401.09135.
> [9] Garrigos, G., & Gower, R. M. (2023). Handbook of convergence theorems for (stochastic) gradient methods. arXiv preprint arXiv:2301.11235.

---

> > ### Comment · Reviewer_advP · 2025-08-04
> >
> > Thank you for your reply. I have no further questions.

---

### Official Review · Reviewer_DffW · 2025-06-28

**Clarity:** 2
**Significance:** 3
**Originality:** 2
**Rating:** 4
**Confidence:** 3

**Summary:**

This paper analyzes the role of the outer learning rate in Local SGD, proving its impact on convergence and showing it can compensate for suboptimal inner learning rates, sometimes requiring values exceeding 1. The theoretical framework includes extensions to momentum-based outer optimizers and introduces an analysis for tuning insights. Experiments on language models validate these findings.

**Questions:**

Please see Strengths And Weaknesses

**Ethical Concerns:**

["NO or VERY MINOR ethics concerns only"]

**Final Justification:**

Given that my concerns are addressed, I'd like to raise my score.

**Limitations:**

Please see Strengths And Weaknesses

**Paper Formatting Concerns:**

Please see Strengths And Weaknesses

**Quality:**

3

**Strengths And Weaknesses:**

1.	The paper analyzes local SGD methods in the homogeneous setting. However, the heterogeneous setting is more common in convergence analyses for similar local SGD methods (e.g., [1]). Could the authors discuss the challenges of extending their analysis to the heterogeneous setting?

2.	Since this work does not propose a new method, I would like to assess its contributions along the following dimensions:

a)	Theoretical Novelty:

Are there any new frameworks or techniques in the theoretical analysis compared to prior works like [1][2]? If so, the authors should highlight these differences more explicitly.

b)	Practical Impact:

For optimization methods, it’s not recommended to introduce new hyperparameters if not necessary, due to the cost of hyperparameter tuning. Gen-loc-SGD has more learning rates that need to be tuned compared with vanilla local SGD. Are there some dramatic improvements due to the introduction of the outer learning rates?

3.	Some suggestions for writing:

a)	Clarity of Notation (Lines 54–55 & Algorithm 1):
The phrase “for what the ideal learning rate pair ($\eta$, $\gamma$) should be” is unclear. The roles of $\eta$ and $\gamma$ should be defined earlier, ideally in Algorithm 1, rather than later in Section 3 (Lines 116–118).

b)	Mathematical Notation:
Vectors (e.g., parameters x and gradients g) should be consistently boldfaced for better readability.

[1] Hao Yu, et al. On the Linear Speedup Analysis of Communication Efficient Momentum SGD for Distributed Non-Convex Optimization. ICML 2019.
[2] Stich, SebastianU, et al. Local SGD Converges Fast and Communicates Little. ICLR 2019.

---

> ### Author Rebuttal · Authors · 2025-07-27
>
> Thank you so much for your review.
>
> 1. "Could the authors discuss the challenges of extending their analysis to the heterogeneous setting?" Our primary setting is large language model training, as in DiLoCo and similar works, and there the training data is i.i.d. Additionally, we would like to point out that in the heterogeneous setting, the guarantees for Local SGD are very pessimistic; that is, small degrees of heterogeneity result in much worse theoretical guarantees for Local SGD compared to Minibatch SGD, as discussed in [3, p. 5-6], and it is not clear under which assumptions the algorithm should be analyzed (see [12]). Because we wanted to focus on when Local SGD is superior to the alternatives, this is one more reason we focused on the i.i.d. setting.
>
> **That said, we believe extending our analysis to the heterogeneous setting is possible.** Lemma B.6. still holds, but Lemma B.7. should be adjusted to handle function heterogeneity. If we make the same assumption as [1, Assumption 1.2 (3)], then this is possible at the cost of an extra $H$ factor. Modifying the rest of the proof of Theorem 3.3 with the revised Lemma B.7. is straightforward. The biggest drawback is that the conclusion will be too pessimistic compared to Minibatch SGD. We believe it might be possible to use assumptions such as second-order similarity and second-order smoothness, as in [4], to get around this, but this requires significant modifications not just to Lemma B.7 but to the main proof strategy of Theorem 3.3, and we leave it to future work.
>
> 2. "Are there any new frameworks or techniques in the theoretical analysis compared to prior works like [1][2]? If so, the authors should highlight these differences more explicitly." Yes, there are. First, our algorithmic framework is significantly different; Neither [1] nor [2] consider the use of outer learning rates or outer momentum, with both using simple averaging (equivalent to fixing the outer learning rate at $1$) and [1] considering *local* momentum. Our framework enables us to study the conditions under which using *outer* learning rates and momentum is useful. This difference in framework is the primary reason why we compare against [5, 6] in the discussion after Theorem 3.3 and not [1, 2]. Second, the analysis of Local SGD always involves bounding the so-called "client drift", or how far the local iterates stray from each other between synchronizations. In [1] this is controlled in Lemma 4 and in [2] this is controlled in Lemma 3.3. Both of these control the drift by bounding the maximum squared norm. We control the drift using a combination of the local regret against the starting point (Lemma B.6) and the maximum squared norm (Lemma B.7). This more fine-grained control allows us to show the benefit of using a large outer learning rate. To the best of our knowledge, a regret guarantee like Lemma B.6. has not been used in the literature before. In order to use this lemma, we also have to conduct the analysis *per communication round* rather than per training step as in [1, 2]. We will revise the next version of the manuscript to highlight these novel contributions.
>
> 3. "For optimization methods, it’s not recommended to introduce new hyperparameters if not necessary, due to the cost of hyperparameter tuning. Gen-loc-SGD has more learning rates that need to be tuned compared with vanilla local SGD. Are there some dramatic improvements due to the introduction of the outer learning rates?" Yes, the performance degradation when not using an outer learning rate is significant. In our experiments, vanilla Local SGD consistently underperforms compared to using outer optimizers. For example, with the 150M model at H=50, M=4, it achieves a perplexity of 17.75, while Nesterov with lr=0.7 reaches 17.25 and SF-SGD with lr=2.0 achieves 16.88. These represent improvements of 2.8% and 4.9% respectively. The performance gains become even more pronounced at larger model scales: for the 1B model, vanilla SGD achieves 13.67 perplexity while Nesterov (lr=0.7) and SF-SGD (lr=2.0) reach 12.51 and 12.40 respectively, representing improvements of 8.5% and 9.3%. Because we can tune the outer learning rate at a smaller scale (this is what we did here) and therefore at a much smaller cost, we believe the additional hyperparameter is justified. There are very few reliable ways of squeezing out an additional 8%-10% performance at the 1B scale at the cost of one additional hyperparameter that can be tuned at a smaller scale.
>
> 4. Thank you for your comments on improving clarity. We will define the learning rates $\gamma$ and $\eta$ in Algorithm 1 and use bold faces for the vectors.
>
> We thank you for your review and hope our discussion above sufficiently addresses your concerns.
>
> [1] Hao Yu, et al. On the Linear Speedup Analysis of Communication Efficient Momentum SGD for Distributed Non-Convex Optimization. ICML 2019.
> [2] Stich, Sebastian U. Local SGD Converges Fast and Communicates Little. ICLR 2019.
> [3] Woodworth, B. E., Patel, K. K., & Srebro, N. (2020). Minibatch vs local sgd for heterogeneous distributed learning. Advances in Neural Information Processing Systems, 33, 6281-6292.
> [4] Zindari, A., Luo, R., & Stich, S. U. (2023). On the convergence of local sgd under third-order smoothness and hessian similarity. In OPT 2023: Optimization for Machine Learning.
> [5] Karimireddy, S. P., Kale, S., Mohri, M., Reddi, S., Stich, S., & Suresh, A. T. (2020, November). Scaffold: Stochastic controlled averaging for federated learning. In International conference on machine learning (pp. 5132-5143). PMLR.
> [6] Jhunjhunwala, D., Wang, S., & Joshi, G. (2023). Fedexp: Speeding up federated averaging via extrapolation. arXiv preprint arXiv:2301.09604.
> [12] Patel, Kumar Kshitij, et al. "The limits and potentials of local sgd for distributed heterogeneous learning with intermittent communication." The Thirty Seventh Annual Conference on Learning Theory. PMLR, 2024.

---

> > ### Comment · Reviewer_DffW · 2025-08-05
> >
> > Thanks for your response. I have no further questions currently. Given that my concerns are addressed, I'd like to raise my score.

---

### Official Review · Reviewer_GxxH · 2025-07-01

**Clarity:** 2
**Significance:** 3
**Originality:** 3
**Rating:** 5
**Confidence:** 4

**Summary:**

This paper presents a tighter convergence analysis of Generalized Local SGD in convex settings, highlighting the critical role of the outer learning rate. In particular, the authors show that appropriately setting the outer learning rate can: (1) interpolate between vanilla Local SGD and standard SGD, achieving the better of the two convergence rates; and (2) compensate for an overly small inner learning rate. Additionally, the paper provides (a) convergence guarantees when the outer optimizer uses momentum, and (b) the first high-probability convergence bound for Generalized Local SGD. Finally, the authors validate their theoretical findings through experiments on both convex optimization tasks and transformer models.

**Questions:**

Why is the high-probability bound in Section 3.3 referred to as an "adaptive convergence result"? The term "adaptive" might be somewhat misleading, as it typically reminds people of adaptive gradient methods such as Adam

**Ethical Concerns:**

["NO or VERY MINOR ethics concerns only"]

**Final Justification:**

All my concerns and questions have been addressed. I would like to keep my positive rating.

**Limitations:**

Yes.

**Quality:**

3

**Strengths And Weaknesses:**

**Strengths**:

1. This is a nice theory paper, with solid technical contributions.

2. The theoretical insights are valuable and provide practical guidance for setting the outer learning rate.

3. The experiments align with the theoretical findings well. In particular, when properly tuned, Generalized Local SGD outperforms vanilla parallel AdamW within a fixed number of training steps.

**Weaknesses**:

1. The writing could be improved. In particular, the presentation of Theorems 3.5 and 3.6 is overly technical. The authors should better highlight the key takeaways and practical implications.

2. The experimental section lacks clarity. For example, in Figure 2, it is unclear what the numbers in the legend refer to. Do they refer to the wall-clock time? I could not find an explanation in the main text or figure caption. Additionally, the authors should clearly introduce all algorithm abbreviations used in the figures in the captions.

---

> ### Author Rebuttal · Authors · 2025-07-27
>
> Thank you so much for your review and positive evaluation of our work.
>
> 1. We agree that our use of "adaptive" in Section 3.3, as we do not mean adaptivity in the same sense as Adam or AdaGrad (i.e. algorithmic adaptivity), but rather in the sense of obtaining data-dependent guarantees. We will modify the term "adaptive" to instead be "data-dependent" and thank the reviewer for pointing this out.
> 2. We will reorganize the presentation of Theorems 3.5 and 3.6 to highlight the key takeaways and defer the more technical discussions. The core observation from  Theorem 3.5,  is that momentum allows us to use an effectively larger stepsize but does not alter the outer stepsize tradeoff shown by Theorem 3.3. Similarly, Theorem 3.6 shows that when noise magnitude dominates the optimization error (e.g. in later training stages), using $\gamma < 1$ helps maintain convergence. Conversely, when the optimization term is larger than noise, $\gamma > 1$ acts like momentum to accelerate convergence. When both terms are comparable, $\gamma = 1$ (simple averaging) is optimal.
> 3. We thank the reviewer for pointing this out. Figure 2 shows the total number of training steps (i.e. $R \times H$ where R is the number of communication rounds and $H$ is the number of local steps). We will clarify this and add a short table or glossary with all the abbreviations we use.

---

> > ### Comment · Reviewer_GxxH · 2025-08-03
> >
> > Thank you for your response. Replies 1 and 2 have addressed my concerns. Particularly, your explanation of the role of the outer learning rate in balancing the optimization and noise terms is both clear and intuitive.
> >
> > However, regarding Figure 2, my question was about the numbers shown in the legend (e.g., Data-parallel: 18.7), rather than the meaning of the x-axis. Could the authors clarify what these numbers represent? Do they indicate wall-clock time or something else?

---

> > > ### Author Response · Authors · 2025-08-03
> > >
> > > Thank you for your response and for pointing out the ambiguity in the figure. The numbers in the legend in Figure 2 represent the final perplexity reached by each method. We included them in the legend for ease of comparison; We will make it clear that they represent the perplexity in the caption.

---

> > > > ### Comment · Reviewer_GxxH · 2025-08-03
> > > >
> > > > Thank you for your response. All my concerns and questions have been addressed. I will keep my positive rating!

---

### Decision · Program_Chairs · 2025-09-17

**Decision:**

Accept (poster)

**Comment:**

This paper provides a clean theoretical result about the optimal "outer learning rate" of local SGD (a distributed version of SGD consisting of local gradient updates, aggregation and a outer update stages). The reviewer all agree that the contribution of the paper is solid and useful in optimization practice of distributed training.

The concerns seem to be minor, and the authors addressed them well during the rebuttal stage.